# Joint Hierarchical Representation Learning of Samples and Features via Informed Tree-Wasserstein Distance

**Ya-Wei Eileen Lin**[1]    **Ronald R. Coifman**[2]    **Gal Mishne**[3]    **Ronen Talmon**[1]

[1]Viterbi Faculty of Electrical and Computer Engineering, Technion
[2]Department of Mathematics, Yale University
[3]Halicioğlu Data Science Institute, University of California San Diego

## Abstract

High-dimensional data often exhibit hierarchical structures in both modes: samples and features. Yet, most existing approaches for hierarchical representation learning consider only one mode at a time. In this work, we propose an unsupervised method for jointly learning hierarchical representations of samples and features via Tree-Wasserstein Distance (TWD). Our method alternates between the two data modes. It first constructs a tree for one mode, then computes a TWD for the other mode based on that tree, and finally uses the resulting TWD to build the second mode's tree. By repeatedly alternating through these steps, the method gradually refines both trees and the corresponding TWDs, capturing meaningful hierarchical representations of the data. We provide a theoretical analysis showing that our method converges. We show that our method can be integrated into hyperbolic graph convolutional networks as a pre-processing technique, improving performance in link prediction and node classification tasks. In addition, our method outperforms baselines in sparse approximation and unsupervised Wasserstein distance learning tasks on word-document and single-cell RNA-sequencing datasets.

## 1 Introduction

High-dimensional data with hierarchical structures are prevalent in numerous fields, e.g., gene expression [1–3], image analysis [4–6], neuroscience [7, 8], and citation networks [9–11]. Therefore, finding meaningful hierarchical representations for these data is an important task that has attracted considerable attention [12–20]. While hierarchical structure is often assumed to exist in *either the samples or the features* [13, 21–23] (i.e., one of the data modes of a matrix), many real-world datasets exhibit hierarchical structure in both modes. For example, word-document data [24–26] commonly contain hierarchies within features (e.g., related keywords) and within samples (e.g., document topics). Another example is recommendation systems [27, 28], where items (features) could be arranged into taxonomies or product categories, and users (samples) can be represented by multiple levels of behavioral or demographic relations [29, 30].

An emerging approach to represent hierarchical data is based on embedding in hyperbolic space [13, 14, 21–23, 31]. However, most existing hyperbolic representation learning methods focus only on one mode of a data matrix [13, 21, 22], either the rows (samples) or the columns (features). A straightforward way to handle hierarchies in both modes is to learn a separate hierarchical representation for each. We postulate that a joint approach facilitates significant advantages and investigate how the hierarchical structure of features can improve the discovery of the sample hierarchy, and, conversely, how the sample hierarchy can improve the discovery of the feature hierarchy.

To this end, we propose to jointly learn the hierarchical representation of features and samples using Tree-Wasserstein Distance (TWD) [32] in an iterative manner. Our approach departs from

39th Conference on Neural Information Processing Systems (NeurIPS 2025).

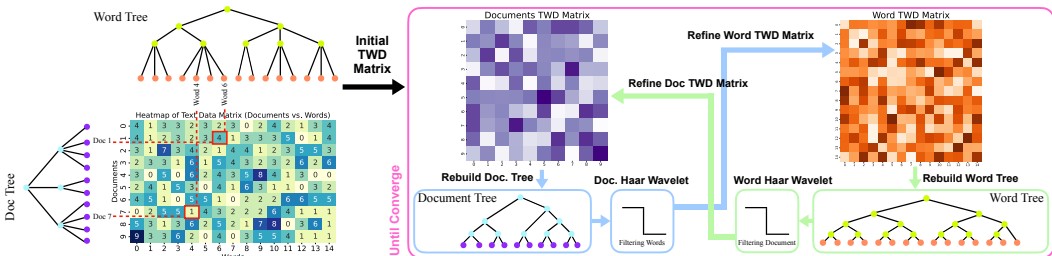

Figure 1: Overview of learning hierarchical representations across samples and features using TWD. Consider a word-document data matrix. We construct an initial tree for one data mode (e.g., words). This tree is then used to compute the TWD in the other mode (e.g., documents). The newly computed TWD informs a tree update of that mode, and the updated tree is subsequently used to compute the TWD in the cross-mode. This alternating procedure continues iteratively, refining both trees.

standard uses of TWD, where trees are typically built for computational efficiency in approximating Wasserstein distances with Euclidean ground metrics [32–36]. Instead, we use trees learned from hierarchical data through diffusion geometry and hyperbolic embedding [37], allowing the hierarchical structure of one mode to be incorporated in the computation of a distance of the other. We start by learning an initial tree of one mode (either features or samples). Then, this learned tree is incorporated into the computation of the TWD of the other mode. The computed, informed TWD is then used to infer a tree of that mode, i.e., sample TWD is used to update the sample tree and feature TWD is used to update the feature tree, as shown in Fig. 1. This procedure is repeated, alternating between the two modes, iteratively refining both trees and recomputing the corresponding TWDs. We provide theoretical guarantees that this iterative process converges and that a fixed point exists.

Based on the learned trees that represent the hierarchical structures of the data, we extend the iterative algorithm by incorporating an additional adaptive filtering step. This filtering is implemented using Haar bases [38–41], which are constructed based on the learned trees. More concretely, at each iteration, we view each sample (resp. feature) as a signal supported on the feature tree (resp. sample tree) [42]. We then apply Haar wavelet filters [38, 43] induced by the feature tree (resp. sample tree) to the samples (resp. features), as illustrated in Fig. 1. Assuming the trees reflect the intrinsic hierarchical structure, applying the resulting data-driven wavelet filters can remove noise and other nuisance components [44]. We show that integrating the filters into the iterative process also leads to convergence. Empirically, we demonstrate that this approach improves hierarchical representations in terms of sparse approximation, and the resulting TWD leads to superior performance in document and single-cell classification.

We further demonstrate the practical benefit of our hierarchical representations by using them to initialize hyperbolic graph convolutional networks (HGCNs) [21, 45]. By incorporating our data-driven hierarchical representation as a preprocessing step, we observe improved performance in link prediction and node classification tasks compared to standard HGCNs. This highlights both the compatibility and effectiveness of our approach for hyperbolic graph-based models.

**Our contribution.** (1) We present an iterative framework for jointly learning hierarchical representations of features and samples using TWD. (2) We further enhance the hierarchical representations using Haar wavelet filters constructed from the learned trees. (3) We show that our method achieves superior performance in sparse approximation, as well as document and single-cell (scRNA-seq) classification. (4) We also show that the proposed hierarchical representation can be used to initialize HGCNs, improving link prediction and node classification on hierarchical graph data.

## 2 Related work

**Tree-Wasserstein distance.** Tree-Wasserstein distance (TWD) [32–36] was introduced to mitigate the high computational cost of the Wasserstein distance [46], which is a powerful tool to compare sample distributions while taking into account feature relationship [47–49]. Most TWD methods involve two steps: (i) constructing a feature tree and (ii) using this tree to compute a sample TWD matrix. While the tree is typically built to approximate the *Euclidean* ground metric in TWD literature,

[37] recently proposed tree construction via hyperbolic embeddings [50] and diffusion geometry [51], enabling the tree metric to reflect geodesic distances on a latent unobserved tree underlying features.

**Co-Manifold learning.** Co-manifold learning aims to jointly recover the geometry of samples and features by treating their relationships as mutually informative, i.e., the feature manifold informs the sample manifold, and vice versa. It has been applied in joint embedding [52–55] and dimensionality reduction [7, 56, 57], assuming that both rows and columns of the data matrix lie on smooth, low-dimensional manifolds. While effective in capturing geometric structures, these methods often rely on Riemannian manifolds with non-negative curvature assumptions [55, 58], which may not be suited well with the negative curvature often associated with hierarchical data.

**Unsupervised ground metric learning.** Wasserstein singular vectors (WSV) [59] were introduced to jointly compute Wasserstein distances across samples and features, where the Wasserstein distance in one mode (e.g., samples) acts as the ground metric for the other (e.g., features). However, WSV is computationally expensive. To address this, Tree-WSV [60] was recently proposed, using tree-based approximations of the Wasserstein distance to reduce computational complexity. While these methods alternate between computing distances in a manner similar to ours, neither WSV nor Tree-WSV was designed to learn the hierarchical representations of the data. In contrast, our method explicitly learns the hierarchical representation of samples and features, further enhanced by wavelet filtering and its integration into HGCNs. These aspects were not explored in the WSV or Tree-WSV frameworks.

## 3 Background

**Tree construction using hyperbolic and diffusion geometry.** Consider a set of high-dimensional points $\mathcal{Z} = \{\mathbf{z}_j \in \mathbb{R}^n\}_{j=1}^m$, and let $\mathbf{M} \in \mathbb{R}^{m \times m}$ be a suitable pairwise distance matrix between these $m$ points. In a recent work [37], a binary tree was derived from a hyperbolic embedding obtained through multiscale diffusion densities built from $\mathbf{M}$ [50], employing the concept of lowest common ancestors [61] within this embedding space. The resulting tree, denoted $\mathcal{T}(\mathbf{M})$, has $m$ leaves corresponding to the $m$ points. Details on the embedding and tree construction are in App. A.

**Tree-Wasserstein distance.** Consider a tree $T = (V, E, \mathbf{A})$ with $N_{\text{leaf}}$ leaves, where $V$ is the vertex set with $N$ nodes, $E$ is the edge set, and $\mathbf{A} \in \mathbb{R}^{N \times N}$ is the edge weight matrix. The tree distance $d_T$ is the sum of weights of the edges on the shortest path between any two nodes on $T$. Let $\Upsilon_T(v)$ be the set of nodes in the subtree rooted at $v \in V$. Each $u \in V$ has a unique parent $v$, with edge weight $w_u = d_T(u, v)$ [62]. Given distributions $\boldsymbol{\rho}_1, \boldsymbol{\rho}_2 \in \mathbb{R}^{N_{\text{leaf}}}$ supported on $T$, the TWD [32] is defined by

$$\text{TW}(\boldsymbol{\rho}_1, \boldsymbol{\rho}_2, T) = \sum_{v \in V} w_v \left| \sum_{u \in \Upsilon_T(v)} (\boldsymbol{\rho}_1(u) - \boldsymbol{\rho}_2(u)) \right|. \tag{1}$$

**Haar wavelet.** Consider a complete binary tree $B$ with $m$ leaves. Let $\ell = 1, \ldots, L$ denote the levels in the tree, where $\ell = 1$ is the root level and $\ell = L$ is the leaf level. Let $\widehat{\Upsilon}(\ell, s)$ be the set of all leaves in the subtree, whose root is the $s$-th node of the tree at level $\ell$, where $s = 1, \ldots, N_\ell$ and $N_\ell$ is the number of nodes in the $\ell$-level. At level $\ell$, a subtree $\widehat{\Upsilon}(\ell, s)$ splits into two sub-subtrees $\widehat{\Upsilon}(\ell + 1, s_1)$ and $\widehat{\Upsilon}(\ell + 1, s_2)$. A zero-mean Haar wavelet $\boldsymbol{\beta}_{\ell,s} \in \mathbb{R}^m$ has non-zero values only at the indices corresponding to leaves in the sub-subtrees and is piecewise constant on each of them (see App. A for an illustration and further details) [40, 39, 43]. The set of these Haar wavelets, along with a constant vector, is complete and forms an orthonormal Haar basis, denoted by $\mathbf{B} \in \mathbb{R}^{m \times m}$ with each column corresponding to a Haar wavelet (basis vector). Any vector $\mathbf{a} \in \mathbb{R}^m$ can be expanded in this Haar basis as $\mathbf{a} = \sum_{\ell,\varepsilon} \alpha_{i,\ell,\varepsilon} \boldsymbol{\beta}_{\ell,\varepsilon}$, where $\alpha_{i,\ell,s} = \langle \mathbf{a}, \boldsymbol{\beta}_{\ell,s} \rangle$ is the expansion coefficient.

## 4 Proposed method

**Problem setting.** Given a data matrix $\mathbf{X} \in \mathbb{R}_+^{n \times m}$ with $n$ rows (samples) and $m$ columns (features), we denote $\mathbf{X}_{i,:}$ and $\mathbf{X}_{:,j}$ as the $i$-th sample and the $j$-th feature, respectively. Our goal is to learn hierarchical representations for both samples and features. We model the hierarchical structure of the data by constructing rooted weighted trees as follows: a sample tree $T_r$ with $n$ leaves, where each leaf represents one sample, and a feature tree $T_c$ with $m$ leaves, where each leaf represents one feature.

**Algorithm 1** Iterative Joint Hierarchical Representation Learning via Tree-Wasserstein Distance

---

**Input:** Data matrix $\mathbf{X} \in \mathbb{R}_+^{n \times m}$, $\mathbf{M}_c \in \mathbb{R}^{m \times m}$ and $\mathbf{M}_r \in \mathbb{R}^{n \times n}$, $\gamma_r, \gamma_c > 0$
**Output:** Trees $\mathcal{T}(\mathbf{W}_r^{(l)})$ and $\mathcal{T}(\mathbf{W}_c^{(l)})$, and TWDs $\mathbf{W}_r^{(l)}$ and $\mathbf{W}_c^{(l)}$

$l \leftarrow 0$, $\widehat{\mathbf{X}}_r \leftarrow \{\mathbf{r}_i = \mathbf{X}_{i,:}^\top / \|\mathbf{X}_{i,:}\|_1\}$, and $\widehat{\mathbf{X}}_c \leftarrow \{\mathbf{c}_j = \mathbf{X}_{:,j} / \|\mathbf{X}_{:,j}\|_1\}$ ▷ Initialization
$\mathbf{W}_r^{(0)} \leftarrow \Phi(\widehat{\mathbf{X}}_r; \mathcal{T}(\mathbf{M}_c))$     ▷ Construct initial feature tree and compute initial sample TWD
$\mathbf{W}_c^{(0)} \leftarrow \Phi(\widehat{\mathbf{X}}_c; \mathcal{T}(\mathbf{M}_r))$     ▷ Construct initial sample tree and compute initial feature TWD
**repeat**
    $\mathbf{W}_r^{(l+1)} \leftarrow \Phi(\widehat{\mathbf{X}}_r; \mathcal{T}(\mathbf{W}_c^{(l)}))$ and $\mathbf{W}_c^{(l+1)} \leftarrow \Phi(\widehat{\mathbf{X}}_c; \mathcal{T}(\mathbf{W}_r^{(l)}))$     ▷ Iterative update
    $l \leftarrow l + 1$
**until** convergence

---

Placing data points as leaves follows the established line of works in statistics [63–65], manifold learning [52–55, 7, 43], and TWD literature [32–36]. We propose an iterative scheme that alternates between learning the sample tree and learning the feature tree.

### 4.1 Iterative scheme for joint hierarchical representation learning

To jointly learn hierarchical representations for samples and features, we use TWD as a means to facilitate the relationships between samples and features. We begin by constructing a tree for one data mode; without loss of generality, we first construct an initial *feature* tree. Given a pairwise distance matrix $\mathbf{M}_c \in \mathbb{R}^{m \times m}$ over the $m$ features, we build a complete binary tree $\mathcal{T}(\mathbf{M}_c)$ with $m$ leaves [37], where each leaf corresponds to a feature (see App. A for details). This *feature* tree then serves as the hierarchical ground metric for computing TWD between the *samples*:

$$\mathbf{W}_r^{(0)}(i, i') = \text{TW}(\mathbf{r}_i, \mathbf{r}_{i'}, \mathcal{T}(\mathbf{M}_c)) + \gamma_r \zeta(\mathbf{r}_i - \mathbf{r}_{i'}), \tag{2}$$

where $\mathbf{r}_i = \mathbf{X}_{i,:}^\top / \|\mathbf{X}_{i,:}\|_1$ is the $i$-th normalized sample, $\zeta$ is a norm regularize based on the snowflake penalty [57], and $\gamma_r > 0$. For brevity, we write the resulting sample pairwise TWD matrix as

$$\mathbf{W}_r^{(0)} = \Phi(\widehat{\mathbf{X}}_r; \mathcal{T}(\mathbf{M}_c)) \in \mathbb{R}^{n \times n}, \tag{3}$$

where $\widehat{\mathbf{X}}_r = [\mathbf{r}_1, \ldots, \mathbf{r}_n]^\top \in \mathbb{R}^{n \times m}$, and $\Phi$ denotes a function that computes the pairwise TWD between rows of the matrix $\widehat{\mathbf{X}}_r$ using the tree $\mathcal{T}(\mathbf{M}_c)$ defined over features. A similar analogous construction is applied to the other mode (the samples). Given an initial pairwise distance matrix $\mathbf{M}_r \in \mathbb{R}^{n \times n}$ over the $n$ samples, we construct an initial *sample* tree $\mathcal{T}(\mathbf{M}_r)$ with $n$ leaves and use it to compute TWD between the *features*:

$$\mathbf{W}_c^{(0)}(j, j') = \text{TW}(\mathbf{c}_j, \mathbf{c}_{j'}, \mathcal{T}(\mathbf{M}_r)) + \gamma_c \zeta(\mathbf{c}_j - \mathbf{c}_{j'}), \tag{4}$$

where $\mathbf{c}_j = \mathbf{X}_{:,j} / \|\mathbf{X}_{:,j}\|_1$ and $\gamma_c > 0$. The resulting feature pairwise TWD is then written as

$$\mathbf{W}_c^{(0)} = \Phi(\widehat{\mathbf{X}}_c; \mathcal{T}(\mathbf{M}_r)) \in \mathbb{R}^{m \times m}, \tag{5}$$

where $\widehat{\mathbf{X}}_c = [\mathbf{c}_1, \ldots, \mathbf{c}_m]^\top \in \mathbb{R}^{m \times n}$. These matrices of the initial sample and feature pairwise TWDs in Eq. (3) and Eq. (5) serve as the starting point of the proposed iterative scheme.

Our iterative scheme alternates between the two data modes in a coordinate descent manner [66, 67]. At iteration $l = 0$, the initial pairwise distance matrices $\mathbf{M}_r$ and $\mathbf{M}_c$ are used to construct the corresponding sample and feature trees, respectively. These trees are then used to compute the initial TWDs in Eq. (3) and Eq. (5). For all subsequent iterations $l \geq 1$, the learned *feature* TWD from the previous step is used to construct a new *feature* tree, which is then used to compute the updated *sample* TWD. The same process is applied to the other mode: the *sample* TWD is used to construct a new *sample* tree, which in turn is used to compute the subsequent *feature* TWD. Formally, using the notation introduced above, the update steps at $l + 1$ iteration are given by

$$\mathbf{W}_r^{(l+1)} = \Phi(\widehat{\mathbf{X}}_r; \mathcal{T}(\mathbf{W}_c^{(l)})) \in \mathbb{R}^{n \times n}, \quad \mathbf{W}_c^{(l+1)} = \Phi(\widehat{\mathbf{X}}_c; \mathcal{T}(\mathbf{W}_r^{(l)})) \in \mathbb{R}^{m \times m}. \tag{6}$$

This alternating scheme, outlined in Alg. 1, allows the hierarchical representation in one mode to iteratively inform and refine the representation in the other mode.

**Theorem 1.** *The sequences* $\mathbf{W}_r^{(l)}$ *and* $\mathbf{W}_c^{(l)}$ *generated by Alg. 1 have at least one limit point, and all limit points are fixed points if* $\gamma_r, \gamma_c > 0$.

The proof is in App. C. Thm. 1 implies the existence of a limit point to which the proposed iterative scheme converges. Alg. 1 can be implemented in practice without regularization, i.e., $\gamma_r = \gamma_c = 0$. However, the conditions $\gamma_r, \gamma_c > 0$ are necessary for Thm. 1.

We remark that while other TWD methods [32–36] could in principle be incorporated into our iterative framework, we chose the tree construction method [37] for two main reasons. First, we empirically observe that computing sample and feature TWDs with these alternative TWD methods often fails to converge. Second, their use as cross-mode tree references would yield trees whose tree distances approximate the Wasserstein distance. However, the Wasserstein metric is not inherently a hierarchical metric. As a result, the trees derived from such approximations do not represent the hierarchy present in the data. In contrast, constructing a hierarchical representation using [37] yields a tree whose geodesic (shortest path) distances reflect the hierarchical relationship underlying the data [68, 51]. As shown empirically in Sec. 6, the trees obtained by Alg. 1 yield meaningful hierarchical representations of features and samples across benchmarks from multiple domains, leading to improved performance compared to using other TWD methods within our iterative scheme. In addition, we show that the trees and the corresponding TWDs are progressively refined throughout the iterations.

## 4.2 Haar wavelet filtering

To improve the refinement of trees across iterations, we apply a filtering step at every iteration. Specifically, our filters are constructed from Haar wavelets [38–41]. It was shown that Haar wavelets can be derived adaptively from trees [43]. Here, we propose to build the Haar wavelets from the trees inferred through the iterative process. Viewing each *sample* (resp. *feature*) as a signal supported on the *feature* tree (resp. *sample* tree) [42] allows us to apply data-driven Haar wavelet filters. Since by construction the filters reflect the intrinsic hierarchical structure of the data, they enhance this meaningful representation while suppressing noise and other nuisance components.

Given a feature tree $\mathcal{T}(\mathbf{M}_c)$, we construct a Haar basis $\mathbf{B}_c \in \mathbb{R}^{m \times m}$ associated with the tree (see Sec. 3 and App. A for details). Subsequently, each *sample* can be expanded in this Haar basis, and we denote $\boldsymbol{\alpha}_i = (\mathbf{X}_{i,:}\mathbf{B}_c)^\top \in \mathbb{R}^m$ as the vector of the expansion coefficients of the $i$-th sample. To define a wavelet filter, we select a subset of the Haar basis vectors in $\mathbf{B}_c$ as follows. For each coefficient index $j$, we compute the aggregate $L_1$ norm $\sum_{i=1}^n |\boldsymbol{\alpha}_i(j)|$ and sort these values in descending order. Then, we sequentially add the corresponding indices to $\Omega$ until the cumulative contribution $\eta_\Omega = \sum_{q \in \Omega} \sum_{i=1}^n |\boldsymbol{\alpha}_i(q)|$ exceeds a threshold $\vartheta_c > 0$. Let $\widehat{\mathbf{B}}_c \in \mathbb{R}^{m \times d}$ denote the matrix consisting of the $d$ basis vectors in $\Omega$. The filtering step, which yields the *filtered samples*, is defined as

$$\Psi(\mathbf{X}; \mathcal{T}(\mathbf{M}_c)) = (\mathbf{X}\widehat{\mathbf{B}}_c)\widehat{\mathbf{B}}_c^\top \in \mathbb{R}^{n \times m}, \tag{7}$$

where $\Psi$ denotes a wavelet filtering operator applied to the rows of the matrix $\mathbf{X}$, using the Haar wavelet induced by the tree $\mathcal{T}(\mathbf{M}_c)$ defined over features. An analogous wavelet filter can be constructed from a *sample* tree $\mathcal{T}(\mathbf{M}_r)$ and applied to the *features*:

$$\Psi(\mathbf{Z}; \mathcal{T}(\mathbf{M}_r)) = (\mathbf{Z}\widehat{\mathbf{B}}_r)\widehat{\mathbf{B}}_r^\top \in \mathbb{R}^{m \times n}, \tag{8}$$

where $\mathbf{Z} = \mathbf{X}^\top$, and $\widehat{\mathbf{B}}_r \in \mathbb{R}^{n \times d'}$ consists of the top $d'$ basis vectors selected using a threshold $\vartheta_r > 0$ on cumulative coefficient magnitude.

We can apply this *Tree Haar Wavelet filtering* in our joint iterative scheme to update the trees as follows. Given initial inputs $\mathbf{X}^{(0)} = \mathbf{X}, \mathbf{Z}^{(0)} = \mathbf{X}^\top$ and using the notation introduced above, at each iteration $l \geq 0$, we filter the data using the Haar basis vectors derived from the trees:

$$\mathbf{X}^{(l+1)} = \Psi(\mathbf{X}^{(l)}; \mathcal{T}(\widehat{\mathbf{W}}_c^{(l)})) \in \mathbb{R}^{n \times m}, \quad \mathbf{Z}^{(l+1)} = \Psi(\mathbf{Z}^{(l)}; \mathcal{T}(\widehat{\mathbf{W}}_r^{(l)})) \in \mathbb{R}^{m \times n}, \tag{9}$$

where $\widehat{\mathbf{W}}_r^{(0)} = \mathbf{M}_r$ and $\widehat{\mathbf{W}}_c^{(0)} = \mathbf{M}_c$ for iteration $l = 0$. The filtered data is normalized into discrete histograms[1]. Then we refine the trees and TWDs based on the normalized Haar-filtered data:

$$\widehat{\mathbf{W}}_r^{(l+1)} = \Phi(\widehat{\mathbf{X}}_r^{(l+1)}; \mathcal{T}(\widehat{\mathbf{W}}_c^{(l)})) \in \mathbb{R}^{n \times n}, \quad \widehat{\mathbf{W}}_c^{(l+1)} = \Phi(\widehat{\mathbf{X}}_c^{(l+1)}; \mathcal{T}(\widehat{\mathbf{W}}_r^{(l)})) \in \mathbb{R}^{m \times m}, \tag{10}$$

where $\widehat{\mathbf{X}}_r^{(l+1)}$ and $\widehat{\mathbf{X}}_c^{(l+1)}$ are the column-normalized matrices of $\mathbf{X}^{(l+1)}$ and $\mathbf{Z}^{(l+1)}$, respectively. We summarize this iterative learning scheme with the Haar wavelet filters in Alg. 2.

---

[1]The resulting filtered data may contain negative values. To represent it as a histogram, we subtract the vector with its minimum value.

---

**Algorithm 2** Haar Filtering over Alternating Tree Refinement

---

**Input:** Data matrix $\mathbf{X} \in \mathbb{R}_+^{n \times m}$, $\mathbf{M}_c \in \mathbb{R}^{m \times m}$ and $\mathbf{M}_r \in \mathbb{R}^{n \times n}$, $\gamma_r$ and $\gamma_c$, thresholds $\vartheta_c$ and $\vartheta_r$

**Output:** Trees $\mathcal{T}(\widehat{\mathbf{W}}_r^{(l)})$ and $\mathcal{T}(\widehat{\mathbf{W}}_c^{(l)})$, and TWDs $\widehat{\mathbf{W}}_r^{(l)}$ and $\widehat{\mathbf{W}}_c^{(l)}$

$l \leftarrow 0, \mathbf{X}^{(0)} \leftarrow \mathbf{X}, \mathbf{Z}^{(0)} \leftarrow \mathbf{X}^\top, \widehat{\mathbf{W}}_r^{(0)} \leftarrow \mathbf{M}_r$ and $\widehat{\mathbf{W}}_c^{(0)} \leftarrow \mathbf{M}_c$        ▷ Initialization

**repeat**

    $\mathbf{X}^{(l+1)} \leftarrow \Psi(\mathbf{X}^{(l)}; \mathcal{T}(\widehat{\mathbf{W}}_c^{(l)}))$ and $\mathbf{Z}^{(l+1)} \leftarrow \Psi(\mathbf{Z}^{(l)}; \mathcal{T}(\widehat{\mathbf{W}}_r^{(l)}))$   ▷ Tree haar wavelet filtering

    $\widehat{\mathbf{X}}_r^{(l+1)} \leftarrow \left\{ \mathbf{r}_i^{(l+1)} = \left( \mathbf{X}_{i,:}^{(l+1)} \right)^\top / \|\mathbf{X}_{i,:}^{(l+1)}\|_1 \right\}$

    $\widehat{\mathbf{X}}_c^{(l+1)} \leftarrow \left\{ \mathbf{c}_j^{(l+1)} = \left( \mathbf{Z}_{j,:}^{(l+1)} \right)^\top / \|\mathbf{Z}_{j,:}^{(l+1)}\|_1 \right\}$

    $\widehat{\mathbf{W}}_r^{(l+1)} \leftarrow \Phi(\widehat{\mathbf{X}}_r^{(l+1)}; \mathcal{T}(\widehat{\mathbf{W}}_c^{(l)}))$ and $\widehat{\mathbf{W}}_c^{(l+1)} \leftarrow \Phi(\widehat{\mathbf{X}}_c^{(l+1)}; \mathcal{T}(\widehat{\mathbf{W}}_r^{(l)}))$     ▷ Iterative update

    $l \leftarrow l + 1$

**until** convergence

---

**Theorem 2.** *The sequences $\widehat{\mathbf{W}}_r^{(l)}$ and $\widehat{\mathbf{W}}_c^{(l)}$ generated by Alg. 2 have at least one limit point, and all limit points are fixed points if $\gamma_r, \gamma_c > 0$.*

The proof of Thm. 2 is in App. C. Same as Thm. 1, Thm. 2 implies the existence of a limit point to which the proposed iterative scheme with the addition of the Haar wavelet filters converges. We argue that the resulting data-driven wavelet filters attenuate noise and other nuisance components [43], and therefore improve the quality of the learned trees. As shown in Sec. 6, this filtering step contributes to more informative hierarchical representations underlying the high-dimensional data, resulting in superior performance on various tasks. We note that while constructing the filters using $L_1$-based selection criterion [69] is effective, alternative filtering strategies [70–75] can be considered, depending on the downstream tasks. We leave this extension for future work.

We conclude this section with a few remarks. First, while our iterative procedure is broadly applicable to various TWD methods [32–36], the theoretical results and the ability to obtain meaningful hierarchical representations rely on our choice of the specific tree construction method [37]. Second, in each iteration, our method can be computed in $O(n^{1.2} + m^{1.2})$ [37, 76] (see App. E). In contrast, a naïve computation requires $O(mn^3 + m^3 \log m + nm^3 + n^3 \log n)$, making our method more efficient. Third, similar to WSV [59] and Tree-WSV [60], our approach can be viewed as an unsupervised ground metric learning technique [77]. However, its specific focus on learning hierarchical representations for both samples and features distinguishes it from these methods and leads to superior empirical performance for hierarchical data (see Sec. 6). For further comparison with Tree-WSV, we refer to App. B and App. G.3. Finally, we remark that either the sample or the feature hierarchical structure can be provided as a prior and used for initialization of our method (see Sec. 5).

## 5   Incorporating learned hierarchical representation within HGCNs

In Sec. 4, we introduced a joint hierarchical representation learning framework using TWD. Typically, the initial pairwise distance matrices $\mathbf{M}_r$ and $\mathbf{M}_c$ are data-driven, e.g., using standard metrics such as the Euclidean distance or the cosine similarity. An advantage of our method is that it can incorporate prior knowledge when a hierarchical structure is available for one of the modes. Without loss of generality, we consider scenarios where a hierarchical structure over the $n$ samples is known and represented by a graph $H = ([n], E, \mathbf{A})$, with $[n] = \{1, \ldots, n\}$. To integrate this prior, we initialize $\mathbf{M}_r$ using the shortest-path distances $d_H$ induced by $H$. This initialization introduces a structured prior in one mode while allowing the hierarchy in the other mode to be learned jointly via TWD.

Incorporating such a prior is particularly relevant for hierarchical graph data [9, 21, 78], where the node hierarchy is provided, while the feature structure remains implicit. We demonstrate the compatibility of our approach with hyperbolic graph convolutional networks (HGCNs) [21, 45]. Specifically, during the neighborhood aggregation step in HGCNs, we replace the predefined hierarchical graph with the sample tree inferred by our method after convergence. This learned tree guides the aggregation process, enabling hierarchy-aware message passing that reflects structured relations among samples and across features. As shown in Sec. 6, incorporating our method as a pre-processing

Table 1: The $L_1$ norm of the Haar expansion coefficients. Values are reported in the format samples / features (the lowest in bold and the second lowest underlined).

| | BBCSPORT | TWITTER | CLASSIC | AMAZON | ZEISEL | CBMC |
|---|---|---|---|---|---|---|
| Co-Quadtree | 26.6 / 27.8 | 59.5 / 24.9 | 64.3 / 108.7 | 87.3 / 102.7 | 157.0 / 242.4 | 1616.6 / 77.3 |
| Co-Flowtree | 27.4 / 31.5 | 69.1 / 20.9 | 73.7 / 97.5 | 88.1 / 110.1 | 173.7 / 240.0 | 1688.8 / 81.7 |
| Co-WCTWD | 26.5 / 26.7 | 57.6 / 17.1 | 63.3 / 81.7 | 77.4 / 96.4 | 136.4 / 202.9 | 1308.1 / 68.4 |
| Co-WQTWD | 25.2 / 32.1 | 56.9 / 30.2 | 61.3 / 100.1 | 74.7 / 111.0 | 135.0 / 224.8 | 1324.8 / 67.4 |
| Co-UltraTree | 37.6 / 32.1 | 69.8 / 18.8 | 76.0 / 125.5 | 86.3 / 133.3 | 155.4 / 226.6 | 1450.0 / 82.2 |
| Co-TSWD-1 | 26.0 / 29.6 | 70.1 / 15.0 | 69.8 / 91.6 | 100.4 / 111.2 | 179.5 / 211.5 | 1716.1 / 79.5 |
| Co-TSWD-5 | 30.5 / 24.7 | 58.9 / 16.0 | 65.7 / 98.0 | 83.3 / 102.3 | 170.2 / 234.7 | 1560.2 / 70.7 |
| Co-TSWD-10 | 21.1 / 34.5 | 52.6 / 23.6 | 59.3 / 140.8 | 74.5 / 135.8 | 141.4 / 229.7 | 1373.3 / 81.2 |
| Co-SWCTWD | 35.0 / 29.5 | 69.0 / 24.7 | 79.8 / 93.6 | 95.9 / 104.6 | 170.4 / 215.9 | 1595.0 / 89.7 |
| Co-SWQTWD | 33.5 / 25.6 | 57.6 / 13.9 | 61.3 / 86.0 | 77.0 / 98.9 | 141.8 / 209.6 | 1369.8 / 68.0 |
| Co-MST-TWD | 30.4 / 34.1 | 69.1 / 22.1 | 77.5 / 109.3 | 97.0 / 126.4 | 179.1 / 254.0 | 1755.3 / 83.7 |
| Co-TR-TWD | 36.8 / 24.5 | 59.1 / 15.2 | 66.1 / 94.7 | 82.2 / 102.3 | 124.6 / 238.6 | 926.8 / 68.8 |
| Co-HHC-TWD | 22.5 / 22.7 | 51.5 / 13.6 | 58.7 / 93.3 | 74.1 / 110.1 | 192.1 / 215.5 | 1148.8 / 79.0 |
| Co-gHHC-TWD | 27.9 / 10.6 | 65.0 / 16.7 | 72.8 / 112.9 | 88.5 / 115.5 | 162.7 / 240.7 | 1612.8 / 70.8 |
| Co-UltraFit-TWD | 22.5 / 22.1 | 51.9 / 13.4 | 58.4 / 81.3 | 73.1 / 92.8 | 133.7 / 202.0 | 1294.6 / 78.1 |
| QUE | 22.8 / 19.8 | 57.8 / 10.4 | 71.6 / 67.6 | 88.8 / 72.0 | 93.1 / 173.3 | 906.4 / 58.5 |
| Tree-WSV | 23.6 / 24.9 | 54.3 / 18.2 | 65.4 / 99.2 | 84.2 / 106.3 | 139.7 / 201.9 | 1637.4 / 73.2 |
| Alg. 1 | 12.9 / 7.0 | 25.4 / 3.7 | 40.4 / 24.6 | 51.0 / 30.1 | 64.4 / 110.8 | 511.3 / 50.1 |
| Alg. 2 | **10.1 / 4.8** | **22.4 / 3.4** | **37.2 / 19.4** | **46.6 / 26.6** | **50.9 / 93.7** | **489.4 / 45.6** |

step improves performance on link prediction (LP) and node classification (NC) for hierarchical graph datasets. While our focus here is on HGCNs due to their improved performance on NC and LP, the same initialization strategy can be adopted in other hyperbolic architectures, such as hyperbolic neural networks (HNNs) [79, 80]. Further technical details on this integration are provided in App. D.

# 6 Experimental results

We evaluate our methods on sparse approximation and unsupervised Wasserstein distance learning with word-document and scRNA-seq benchmarks. Additionally, we examine the integration of our methods into HGCNs for hierarchical graph data on LP and NC tasks. The implementation details, including hyperparameters, are reported in App. E. Additional experiments, e.g., empirical convergence, ablation study, runtime analysis, and co-clustering performance, are presented in App. F.

## 6.1 Evaluating hierarchical representations via sparse approximation

We first demonstrate the advantages of the learned trees from Alg. 1 and Alg. 2 for sparse approximation tasks on the data matrix. The quality of the feature tree (and similarly, the sample tree) is evaluated by the $L_1$ norm of their expansion coefficients across all samples (and the features, respectively) [81–83]. A lower $L_1$ norm indicates a more efficient (sparser) representation of the data using fewer significant Haar coefficients, thus indicating the learned tree structures better reflect the hierarchical information of the data [43, 53]. We test four word-document datasets [25]: BBCSPORT, TWITTER, CLASSIC, and AMAZON, and two scRNA-seq datasets [84]: ZEISEL and CBMC. Both types of data exhibit hierarchical structures in their features and samples [85, 86]. In document data, words (features) form semantic hierarchies (e.g., animal → mammal → dog), while documents (samples) follow topic–subtopic structures (e.g., science → biology → genomics). In scRNA-seq data, genes (features) are organized by functional relationships such as biological pathways and gene ontologies (e.g., immune response genes → cytokine genes → specific interleukins), and cells (samples) follow developmental or taxonomic hierarchies (e.g., hematopoietic stem cell → progenitor cell → mature blood cell types), which are widely modeled as hierarchical in computational biology [87]. Additional information about these datasets can be found in App. E.

We propose to learn hierarchical representations for both samples and features, where each informs the other through TWD. To the best of our knowledge, this approach to hierarchical representation learning has not been previously explored. To demonstrate its effectiveness, we compare Alg. 1 and Alg. 2 against existing TWD-based methods as follows. For a fair comparison, we adapt each baseline TWD method to our iterative setting. Specifically, we begin by using the competing method to compute the sample TWD matrix, from which the sample tree is constructed to approximate the corresponding

Table 2: Document and single-cell classification accuracy.

| | BBCSPORT | TWITTER | CLASSIC | AMAZON | ZEISEL | CBMC |
|---|---|---|---|---|---|---|
| Co-Quadtree | 96.2±0.4 | 69.6±0.3 | 95.9±0.2 | 89.4±0.2 | 81.7±1.0 | 80.7±0.3 |
| Co-Flowtree | 95.7±0.9 | 71.5±0.7 | 95.6±0.5 | 91.4±0.4 | 84.3±0.7 | 83.0±1.2 |
| Co-WCTWD | 93.2±1.2 | 70.2±2.1 | 94.7±2.6 | 87.4±1.0 | 82.5±2.9 | 79.4±2.1 |
| Co-WQTWD | 95.7±1.8 | 70.7±2.2 | 95.5±1.3 | 88.2±2.1 | 82.3±3.1 | 80.5±2.8 |
| Co-UltraTree | 95.3±1.4 | 70.1±2.8 | 93.6±2.0 | 86.5±2.8 | 85.8±1.1 | 84.6±1.3 |
| Co-TSWD-1 | 88.2±1.4 | 70.4±1.2 | 94.7±0.9 | 86.1±0.5 | 80.2±1.4 | 73.2±1.0 |
| Co-TSWD-5 | 88.7±1.7 | 71.0±1.5 | 96.7±0.8 | 91.5±0.4 | 82.0±0.9 | 75.4±0.7 |
| Co-TSWD-10 | 89.2±1.1 | 71.4±1.8 | 95.5±0.2 | 91.8±0.7 | 83.8±0.5 | 77.2±0.9 |
| Co-SWCTWD | 93.5±2.4 | 70.5±1.0 | 94.4±1.3 | 90.7±1.5 | 82.7±1.7 | 79.0±0.9 |
| Co-SWQTWD | 96.2±1.2 | 72.4±2.1 | 96.0±1.1 | 90.6±2.3 | 82.4±1.4 | 81.3±1.1 |
| Co-MST-TWD | 88.7±2.4 | 68.4±3.3 | 91.3±2.9 | 87.1±1.4 | 80.1±2.8 | 76.5±1.3 |
| Co-TR-TWD | 89.5±1.2 | 70.9±1.7 | 93.4±2.2 | 89.5±1.4 | 80.7±0.8 | 78.5±0.9 |
| Co-HHC-TWD | 86.1±2.1 | 70.1±1.3 | 93.6±1.5 | 88.5±0.5 | 83.2±1.4 | 77.6±0.8 |
| Co-gHHC-TWD | 84.0±2.0 | 70.4±1.6 | 90.7±1.7 | 87.2±1.9 | 79.9±1.4 | 84.2±1.2 |
| Co-UltraFit-TWD | 86.8±0.9 | 70.9±1.1 | 91.9±1.0 | 89.9±2.0 | 83.7±2.9 | 79.1±1.8 |
| QUE | 84.7±0.5 | 72.4±0.6 | 91.9±0.5 | 91.6±0.9 | 83.6±1.4 | 82.5±1.9 |
| WSV | 85.9±1.0 | 71.4±1.3 | 92.6±0.7 | 89.0±1.5 | 81.6±2.4 | 77.5±1.7 |
| Tree-WSV | 86.3±1.5 | 71.2±1.9 | 92.4±1.0 | 88.7±1.9 | 82.0±2.9 | 76.4±2.4 |
| Alg. 1 | 96.7±0.3 | 74.1±0.5 | 97.3±0.2 | 94.0±0.4 | 90.1±0.4 | 86.7±0.5 |
| Alg. 2 | **97.3**±0.5 | **76.7**±0.7 | **97.6**±0.1 | **94.2**±0.2 | **94.0**±0.6 | **93.3**±0.7 |

tree-based Wasserstein ground metric. This sample tree is then used to compute the feature TWD matrix, which in turn defines the feature tree. At each iteration, the same baseline method is used to compute the TWDs for samples and features. The competing TWD methods include: Quadtree [32], Flowtree [88], TSWD [35], UltraTree [36], weighted cluster TWD (WCTWD), weighted Quadtree TWD (WQTWD) [34], their sliced variants SWCTWD and SWQTWD, MST [89], Tree Representation (TR) [22], gradient-based hierarchical clustering (HC) in hyperbolic space (gHHC) [90], gradient-based Ultrametric Fitting (UltraFit) [91], and HC by hyperbolic Dasgupta's cost (HHC) [92]. See App. B for details on these methods. We use the prefix "co-" to denote the adaptation of each method to our iterative framework. In addition, we include comparisons with co-manifold learning that involves trees induced by a diffusion embedding (QUE) [52], and Tree-WSV [60], which learns unsupervised ground metrics based on tree approximation of WSV [59].

Tab. 1 reports the $L_1$ norm of the Haar coefficients across all samples and all features, respectively. We report the value after convergence for our methods, and for baselines, we either report it after convergence or, if convergence is not achieved, the obtained value after 25 iterations. We see that our methods provide a more efficient representation, showing that the sparsity and quality of the trees produced by our methods are superior and outperform baselines. Fig. 2 shows the $L_1$ norm of the Haar expansion coefficients obtained by the proposed methods across iterations on ZEISEL dataset, where we observe that the $L_1$ norm is iteratively reduced and reaches convergence. Note that this is not the objective we are minimizing but a conse-

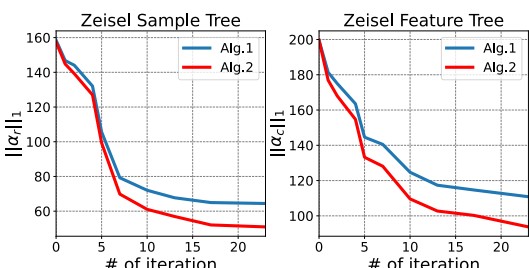

Figure 2: The $L_1$ norm of the Haar coefficients from the sample tree (left) and the feature tree (right) during the sparse approximation task across iterations on the ZEISEL dataset.

quence of our method of learning well hierarchical representations of the data. Notably, Alg. 2 consistently achieves better sparse approximation performance than Alg. 1. We attribute this improvement to the wavelet filtering step, which jointly considers the data and the structure and further improves the quality of the learned hierarchies at each iteration. One might ask whether wavelet filters could similarly benefit other TWD-based baselines. In App. F, we test this by applying wavelets using trees constructed from various TWD baselines. The results show that it does not consistently improve the quality of the tree representations. We argue that this is because the trees in these methods are primarily designed to approximate the Wasserstein distance as the ground metric, rather than to represent the hierarchical structure of the data. Therefore, these trees are not faithful hierarchical

Table 3: ROC AUC for LP, F1 score for the DISEASE dataset, and accuracy for the AIRPORT, PUBMED, and CORA datasets for NC tasks (the highest in bold and the second highest underlined).

| | DISEASE | | AIRPORT | | PUBMED | | CORA | |
|---|---|---|---|---|---|---|---|---|
| | LP | NC | LP | NC | LP | NC | LP | NC |
| EUC | 59.8±2.0 | 32.5±1.1 | 92.0±0.0 | 60.9±3.4 | 83.3±0.1 | 48.2±0.7 | 82.5±0.3 | 23.8±0.7 |
| HYP | 63.5±0.6 | 45.5±3.3 | 94.5±0.0 | 70.2±0.1 | 87.5±0.1 | 68.5±0.3 | 87.6±0.2 | 22.0±1.5 |
| EUC-MIXED | 49.6±1.1 | 35.2±3.4 | 91.5±0.1 | 68.3±2.3 | 86.0±1.3 | 63.0±0.3 | 84.4±0.2 | 46.1±0.4 |
| HYP-MIXED | 55.1±1.3 | 56.9±1.5 | 93.3±0.0 | 69.6±0.1 | 83.8±0.3 | 73.9±0.2 | 85.6±0.5 | 45.9±0.3 |
| MLP | 72.6±0.6 | 28.8±2.5 | 89.8±0.5 | 68.6±0.6 | 84.1±0.9 | 72.4±0.2 | 83.1±0.5 | 51.5±1.0 |
| HNNs | 75.1±0.3 | 41.0±1.8 | 90.8±0.2 | 80.5±0.5 | 94.9±0.1 | 69.8±0.4 | 89.0±0.1 | 54.6±0.4 |
| GCN | 64.7±0.5 | 69.7±0.4 | 89.3±0.4 | 81.4±0.6 | 91.1±0.5 | 78.1±0.2 | 90.4±0.2 | 81.3±0.3 |
| GAT | 69.8±0.3 | 70.4±0.4 | 90.5±0.3 | 81.5±0.3 | 91.2±0.1 | 79.0±0.3 | 93.7±0.1 | 83.0±0.7 |
| GRAPHSAGE | 65.9±0.3 | 69.1±0.6 | 90.4±0.5 | 82.1±0.5 | 86.2±1.0 | 77.4±2.2 | 85.5±0.6 | 77.9±2.4 |
| SGC | 65.1±0.2 | 69.5±0.2 | 89.8±0.3 | 80.6±0.1 | 94.1±0.0 | 78.9±0.0 | 91.5±0.1 | 81.0±0.1 |
| HGCNs | 90.8±0.3 | 74.5±0.9 | 96.4±0.1 | 90.6±0.2 | 96.3±0.0 | 80.3±0.3 | 92.9±0.1 | 79.9±0.2 |
| H2H-GCN | 97.0±0.3 | 88.6±1.7 | 96.4±0.1 | 89.3±0.5 | 96.9±0.0 | 79.9±0.5 | 95.0±0.0 | 82.8±0.4 |
| HGCN-Alg. 1 | 93.2±0.6 | 87.9±0.7 | 93.7±0.2 | 89.9±0.4 | 94.1±0.7 | 81.7±0.2 | 93.1±0.1 | 82.9±0.3 |
| HGCN-Alg. 2 | **98.4**±0.4 | **89.4**±0.3 | **97.2**±0.1 | **92.1**±0.3 | **97.2**±0.2 | **83.6**±0.4 | **96.9**±0.3 | **83.9**±0.2 |

representations of the data, and thus, applying wavelet filters that depend on both data and tree structure fails to improve hierarchical representation learning.

## 6.2 Document and single-cell classification using learned TWD

We further demonstrate the effectiveness of the TWDs obtained by our methods through document and cell classification tasks. We compare our results with the same competing methods used in Sec. 6.1, and additionally include WSV [59] that learns unsupervised ground metrics as a baseline. Classification is performed using $k$NN based on the obtained distances, with cross-validation over five trials. Each trial randomly splits the dataset into 70% training and 30% testing sets.

Tab. 2 shows the document and single-cell classification accuracy. The accuracy of our methods is based on the TWDs after convergence. For the baselines, the accuracy is reported either based on the distance obtained after convergence or the distance after 25 iterations if convergence is not achieved. We see that our methods outperform the baselines by a large margin. This indicates that the TWDs, learned through our iterative scheme, effectively capture the interplay of the hierarchical structures between rows and columns. In addition, we observe that the classification accuracy improves with each iteration of our methods (see Fig. 9 in App. F). While the competing methods also show marginal improvement through the iterative procedure, our methods consistently achieve better performance. Our approaches exhibit fast convergence, typically within 10-14 iterations in all the tested datasets.

## 6.3 Link prediction and node classification for hierarchical graph data

Finally, we show the utility of our methods as a pre-processing step for HGCNs [21, 45], evaluated on LP and NC tasks. We adhere to the experimental setups and baselines used in these works to maintain consistency. For LP task, we use a Fermi-Dirac decoder [93, 13] to compute probability scores for edges. Then, the networks are trained by minimizing cross-entropy loss with negative sampling. The performance of LP is assessed by measuring the area under the ROC curve (AUC). For NC task, we employ a centroid-based classification method [94], where softmax classifiers and cross-entropy loss functions are utilized. Additionally, an LP regularization objective is integrated into the NC task [21, 45]. The NC task is evaluated using the F1 score for binary-class datasets and accuracy for multi-class datasets. We test four datasets [9, 95, 21], including CORA, PUBMED, DISEASE and AIRPOT. Descriptions of these datasets and their splits are included in App. E. We compare our methods with two shallow methods: Euclidean embedding (EUC) and Poincaré embedding (HYP) [13]. We also include comparisons with the concatenation of shallow embeddings and node features, denoted as EUC-MIXED and HYP-MIXED. Furthermore, we include multi-layer perceptron (MLP) and its hyperbolic extension, HNNs [79], as well as four GNNs: GCN [96], GAT [97], GRAPHSAGE [98], and SGC [99]. Lastly, we include HGCNs [21] and H2H-GCN [45].

Tab. 3 shows the performance of integrating our methods with HGCNs on the LP and NC tasks compared to the competing methods. We repeat the random split process 10 times and report the average performance and standard deviation. Our methods consistently outperform the competing baselines across both tasks. Similar to Tab. 1 and Tab. 2, we observe in Tab. 3 that using wavelet filters demonstrates superior performance by a significant margin. This indicates that it effectively represents the hierarchical structure of the graph data, improving the expressiveness of both GNNs and hyperbolic embeddings. A natural question is whether wavelet filters alone (i.e., without our iterative scheme) could benefit HGCNs. We investigate it in App. F and find that such integration does not yield comparable outcomes. To our knowledge, our work is the first to incorporate wavelets with TWD and apply them to HGCNs. We note that our methods could have been integrated within HGCNs in other ways. However, as shown in Tab. 3, the straightforward integration already delivers favorable results. Thus, we opt to explore more complex integration techniques for future work.

## 7 Conclusions

This work introduces an iterative framework for jointly learning hierarchical representations of both samples and features using TWD. The proposed method begins by constructing a tree for one mode (either features or samples), which is then used to compute TWD and infer the tree construction for the other mode. The process alternates between modes, with each tree informing the inference of the other through pairwise TWD computations. To further improve the quality of the tree representations, we apply wavelet filters derived from the learned trees to the data at each iteration, which effectively suppress noise and filter out nuisance components. We show theoretically that the procedure converges and empirically that the trees and TWDs are refined across the iterations. Specifically, empirical evaluations on word-document and scRNA-seq datasets show that the resulting tree representations and TWDs lead to meaningful hierarchical representations. We further demonstrate that the proposed method can serve as a preprocessing step for HGCNs applied to hierarchical graph data, improving performance in hierarchical graph-based learning problems on link prediction and node classification.

**Limitations and future work.** One limitation of our approach lies in the use of trees to represent data geometry. While this representation aligns with our assumption that the data exhibits underlying hierarchical structures, it may not generalize well to data supported on more complex geometries, e.g., spherical manifolds [100], spaces with mixed curvatures [101], asymmetric data [102–104], or general graphs [105]. In future work, we plan to explore more flexible geometric representation methods that can accommodate a broader class of data geometries, while utilizing the proposed iterative procedure between samples and features. We also plan to explore a differentiable variant of TWD, such as the soft TWD [106], which could enable integration of our iterative process into neural architectures. We will further incorporate supervision or task-specific signals into the learning process. Finally, we plan to extend the approach to multi-way data (e.g., tensor-valued inputs).

## Acknowledgments

We thank the anonymous reviewers for their insightful feedback. We thank Gal Maman for proofreading this manuscript. The work of YEL and RT was supported by the European Union's Horizon 2020 research and innovation programme under grant agreement No. 802735-ERC-DIFFOP. The work of GM was supported by NSF CCF-2217058 and CCF-2403452. The work of RC was supported by AFOSR MURI 052721 and DARPA-PA-23-03.

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

# Supplementary Material

This supplemental material is organized as follows:

- **Appendix** A contains the additional background to the proposed method. We begin by reviewing the fundamentals of tree structures. Next, we present key concepts from hyperbolic and diffusion geometry, which are used to construct trees from high-dimensional data. We then describe the construction of a Haar wavelet basis on a given tree and introduce the Wasserstein distance and tree-Wasserstein distance. Lastly, we briefly discuss graph convolutional networks and their hyperbolic counterpart.

- **Appendix** B provides additional related work relevant to our method. It includes manifold learning based on the Wasserstein distance metric, further discussion of existing TWD methods, the sliced-Wasserstein distance, and additional background on co-manifold learning and unsupervised ground metric learning.

- **Appendix** C contains the proofs of Thm. 1 and Thm. 2, along with the supporting lemmas.

- **Appendix** D describes how the learned hierarchical representations obtained by our method can be incorporated into hyperbolic graph convolutional networks as an initialization step.

- **Appendix** E provides details about the experiments presented in Sec. 6. We first describe the datasets used and their statistics. We then report the initial pairwise distances, scaling factors, Haar wavelet thresholds, and other hyperparameters used in our experiments. The norm regularization terms applied in the iterative learning scheme are also specified. We explain how our method is scaled to handle large datasets. We provide details of the experimental setups for document classification and cell-type classification, as well as for link prediction and node classification on hierarchical graph data.

- **Appendix** F presents additional experimental results supporting our method. We first show a synthetic toy example simulating a video recommendation system for visualization. We then demonstrate the empirical uniqueness of the learned hierarchical representations under strong regularization. An ablation study evaluates the role of the iterative joint learning scheme in classification performance and contrasts it with applying wavelet filtering only after convergence. We further assess the effect of integrating wavelet filtering with alternative TWD variants, as well as with HGCNs without the iterative updates. We provide details on the sparse approximation analysis using Haar coefficients across iterations, as well as report the classification performance over the iterative learning scheme. Additionally, we include runtime analysis and explore the use of alternative regularizers within the iterative learning scheme. We include additional experiments on co-clustering tasks and incorporating fixed hierarchical distances in the iterative learning scheme for HGCNs. We evaluate the effectiveness of our method as a preprocessing step across different neural network backbones. We further include a discussion and comparison with other tree distance metrics. Finally, we present the visualization of different iterations of the learned trees and how they evolve in our method on the toy example.

- **Appendix** G provides additional notes on the motivation for incorporating wavelet filters into our iterative learning scheme. We explain why wavelet filters are considered over Laplacian-based filtering in our context. We clarify the objective of our method and highlight the key differences between our approach and unsupervised ground metric learning with trees (Tree-WSV). We discuss the interpretability of TWDs.

## A   Additional background

We include supplementary background in the appendix.

### A.1   Graph and tree

Trees are fundamental structures in graph theory. A tree is a type of graph characterized by hierarchical organization. Below, we outline key concepts and definitions of trees and their associated metrics.

**Graph and shortest path metric.** A graph $G = (V, E, \mathbf{A})$ consists of a set of nodes $V$, edges $E$, and a edge weight matrix $\mathbf{A} \in \mathbb{R}^{m \times m}$. For any two nodes $j, j' \in V$, the shortest path metric $d_G(j, j')$ represents the minimum sum of edge weights along the path connecting node $j$ and node $j'$.

**Tree metric.** A metric $d : V \times V \to \mathbb{R}$ is a tree metric if there exists a tree $T = (V, E, \mathbf{A})$ such that $d(j, j')$ corresponds to the shortest path metric $d_T(j, j')$ on $T$. Notably, tree metrics exhibit 0-hyperbolicity [107], making them well-suited for representing hierarchical data.

**Binary trees.** Binary trees are a specific class of trees where each internal node has exactly two child nodes. In a balanced binary tree, internal nodes (except the root) have a degree of 3, while leaf nodes have a degree of 1. A rooted and balanced binary tree is a further refinement where one internal node, designated as the root, has a degree of 2, and all other internal nodes maintain a degree of 3. Note that a flexible tree can be represented as a binary tree through a transformation, e.g., left-child right-sibling [108].

**Lemma A.1.** *Any tree metric (a metric derived from a tree graph where the distance between two nodes is the length of the unique path connecting them) can be isometrically embedded into an $\ell_1$ space [109].*

## A.2 Hyperbolic geometry

A hyperbolic geometry is a non-Euclidean geometry with a constant negative curvature. It is widely used in modeling structures with hierarchical or tree-like relationships [110]. There are four models commonly used to describe hyperbolic spaces. The Poincaré disk model maps the entire hyperbolic plane inside a unit disk, with geodesics represented as arcs orthogonal to the boundary or straight lines through the center. The Poincaré half-plane model uses the upper half of the Euclidean plane, where geodesics are semicircles orthogonal to the horizontal axis or vertical lines. The Klein model represents hyperbolic space in a unit disk but sacrifices angular accuracy, making it more suitable for some specific calculations. The hyperboloid model relies on a higher-dimensional Lorentzian inner product, offering computational efficiency.

In this work, we consider two equivalent models of hyperbolic space [110]: the Poincaré half-space and the hyperboloid model. The Poincaré half-space was used for embedding that reveals the hierarchical structure underlying high-dimensional data [50], while the hyperboloid model is advantageous for its simple closed-form Riemannian operations [14, 111] used in hyperbolic graph convolutional networks [21, 45].

The $d$-dimensional Poincaré half-space is defined as

$$\mathbb{H}^d = \{\mathbf{a} \in \mathbb{R}^d \big| \mathbf{a}(d) > 0\} \tag{11}$$

with the Riemannian metric tensor $ds^2 = (d\mathbf{a}^2(1) + \ldots + d\mathbf{a}^2(d))/\mathbf{a}^2(d)$ [112]. Let $\mathbb{L}^d$ denote the hyperboloid manifold in $d$ dimensions, defined by

$$\mathbb{L}^d = \{\mathbf{b} \in \mathbb{R}^{d+1} | \langle \mathbf{b}, \mathbf{b} \rangle_{\mathcal{L}} = -1, \mathbf{b}(1) > 0\}, \tag{12}$$

where $\langle \cdot, \cdot \rangle_{\mathcal{L}}$ is the Minkowski inner product $\langle \mathbf{b}, \mathbf{b} \rangle_{\mathcal{L}} = \mathbf{b}^{\top}[-1, \mathbf{0}^{\top}; \mathbf{0}, \mathbf{I}_d]\mathbf{b}$. Let $\mathcal{T}_{\mathbf{b}}\mathbb{L}^d$ be the tangent space at point $\mathbf{b} \in \mathbb{L}^d$, given by $\mathcal{T}_{\mathbf{b}}\mathbb{L}^d = \{\mathbf{v} \in \mathbb{R}^{d+1} | \langle \mathbf{v}, \mathbf{v} \rangle_{\mathcal{L}} = 0\}$. We denote $\|\mathbf{v}\|_{\mathcal{L}} = \sqrt{\langle \mathbf{v}, \mathbf{v} \rangle_{\mathcal{L}}}$ as the norm of $\mathbf{v} \in \mathcal{T}_{\mathbf{b}}\mathbb{L}^d$. For two different points $\mathbf{b}_1, \mathbf{b}_2 \in \mathbb{L}^d$ and $\mathbf{0} \neq \mathbf{v} \in \mathcal{T}_{\mathbf{b}_1}\mathbb{L}^d$, the exponential and logarithmic maps of $\mathbb{L}^d$ are respectively given by

$$\mathrm{Exp}_{\mathbf{b}_1}(\mathbf{v}) = \cosh(\|\mathbf{v}\|_{\mathcal{L}})\mathbf{b}_1 + \sinh(\|\mathbf{v}\|_{\mathcal{L}})\mathbf{v}/\|\mathbf{v}\|_{\mathcal{L}}, \tag{13}$$

$$\mathrm{Log}_{\mathbf{b}_1}(\mathbf{b}_2) = \cosh^{-1}(\eta)(\mathbf{b}_2 - \eta\mathbf{b}_1)/\sqrt{\eta^2 - 1}, \tag{14}$$

where $\eta = -\langle \mathbf{b}_1, \mathbf{b}_2 \rangle_{\mathcal{L}}$. The parallel transport (PT) of a vector $\mathbf{v} \in \mathcal{T}_{\mathbf{b}_1}\mathbb{L}^d$ along the geodesic path from $\mathbf{b}_1 \in \mathbb{L}^d$ to $\mathbf{b}_2 \in \mathbb{L}^d$ is given by

$$\mathrm{PT}_{\mathbf{b}_1 \to \mathbf{b}_2}(\mathbf{v}) = \mathbf{v} + \frac{\langle \mathbf{b}_2 - \lambda\mathbf{b}_1, \mathbf{v} \rangle_{\mathcal{L}}}{\lambda + 1}(\mathbf{b}_1 + \mathbf{b}_2), \tag{15}$$

where $\lambda = -\langle \mathbf{b}_1, \mathbf{b}_2 \rangle_{\mathcal{L}}$, while keeping the metric tensor unchanged. Due to the equivalence between the Poincaré half-space and the Lorentz model, there exists a diffeomorphism $\mathcal{P} : \mathbb{H}^d \to \mathbb{L}^d$ that

maps points from the Poincaré half-space $\mathbf{a} \in \mathbb{H}^d$ to the Lorentz model $\mathbf{b} \in \mathbb{L}^d$ by

$$\mathbf{b} = \mathcal{P}(\mathbf{a}) = \frac{(1 + \|\mathbf{c}\|^2, 2\mathbf{c}(1), \ldots, 2\mathbf{c}(d+1))}{1 - \|\mathbf{c}\|^2}, \tag{16}$$

where

$$\mathbf{c} = \frac{(2\mathbf{a}(1), \ldots, 2\mathbf{a}(d), \|\mathbf{a}\|^2 - 1)}{\|\mathbf{a}\|^2 + 2\mathbf{a}(d+1) + 1}.$$

## A.3 Diffusion geometry

Diffusion geometry [51] is a mathematical framework that analyzes high-dimensional data by capturing intrinsic geometric structures (i.e., manifolds with non-negative curvature) through diffusion processes. It is rooted in the study of diffusion propagation on graphs, manifolds, or general data spaces, where the spread of information or heat over time reflects the underlying connectivity and geometry. Diffusion geometry represents data as nodes in a graph and models their relationships using a diffusion operator. By simulating diffusion processes over this structure, the method identifies meaningful relationships based on proximity and connectivity in the diffusion space. We introduce the construction of the diffusion operator and its desired property below.

Consider a set of high-dimensional points $\mathcal{Z} = \{\mathbf{z}_j \in \mathbb{R}^n\}_{j=1}^m$ lying on a low-dimensional manifold. Let $\mathbf{K} = \exp(-\mathbf{M}^{\circ 2}/\epsilon) \in \mathbb{R}^{m \times m}$ be an affinity matrix, where $\mathbf{M} \in \mathbb{R}^{m \times m}$ is a suitable pairwise distance matrix between the points $\{\mathbf{z}_j\}_{j=1}^m$, $\circ$ is the Hadamard power, and $\epsilon > 0$ is the scale parameter. Note that the matrix $\mathbf{K}$ can be interpreted as an undirected weighted graph $G = (\mathcal{Z}, \mathbf{K})$, where $\mathcal{Z}$ is the node set and $\mathbf{K}$ represents the edge weights.

The diffusion operator [51] is then constructed by $\mathbf{P} = \mathbf{K}\mathbf{D}^{-1}$, where $\mathbf{D}$ is the diagonal degree matrix with entries $\mathbf{D}(j,j) = \sum_l \mathbf{K}(j,l)$. We remark that a density normalization affinity matrix can be considered to mitigate the effects of non-uniform data sampling [51]. Note that the diffusion operator $\mathbf{P}$ is column-stochastic, allowing it to be used as a transition probability matrix of a Markov chain on the graph $G$. Specifically, the vector $\mathbf{p}_j^t = \mathbf{P}^t \boldsymbol{\delta}_j$ is the propagated density after diffusion time $t \in \mathbb{R}_+$ of a density $\boldsymbol{\delta}_j$ initially concentrated at point $j$.

The diffusion operator has been demonstrated to have favorable convergence [51]. As $m \to \infty$ and $\epsilon \to 0$, the operator $\mathbf{P}^{t/\epsilon}$ converges pointwise to the Neumann heat kernel associated with the underlying manifold at the limits. This convergence indicates that the diffusion operator can be viewed as a discrete approximation of the continuous heat kernel, thereby effectively capturing the geometric structure of the underlying manifold in a finite-dimensional setting [51, 113, 114].

## A.4 Tree construction based on hyperbolic and diffusion geometries

Recently, the work [50] introduced a tree construction method that recovers the latent hierarchical structure underlying high-dimensional data $\mathcal{Z}$ based on hyperbolic embeddings and diffusion geometry. Specifically, this latent hierarchical structure can be modeled by a weighted tree $T$, which can be viewed as a discretization of an underlying manifold $\mathcal{J}$. This manifold $\mathcal{J}$ is assumed to be a complete, simply connected Riemannian manifold with *negative* curvature, embedded in a high-dimensional ambient space $\mathbb{R}^n$, and equipped with a geodesic distance $d_{\mathcal{J}}$.

Consider the diffusion operator constructed as in App. A.3, the multi-scale diffusion densities $\boldsymbol{\mu}_j^k = \mathbf{P}^{2^{-k}} \boldsymbol{\delta}_j$ are considered with dyadic diffusion time steps $t = 2^{-k}$ for $k \in \mathbb{Z}_0^+$ [50, 115], and are embedded into a Poincaré half-space by

$$(j,k) \mapsto \mathbf{y}_j^k = \left[ \left( \boldsymbol{\mu}_j^{k \circ 1/2} \right)^\top, 2^{k/2-2} \right]^\top \in \mathbb{H}^{m+1}, \tag{17}$$

where $\boldsymbol{\mu}_j^{k \circ 1/2}$ is the Hadamard root of $\boldsymbol{\mu}_j^k$. Note that diffusion geometry is effective in recovering the underlying manifold [51]; however, the study [50] demonstrated that considering propagated densities with diffusion times on dyadic grids can capture the hidden hierarchical structures by incorporating local information from exponentially expanding neighborhoods around each point. The multi-scale hyperbolic embedding is a function $\texttt{Embedding} : \mathcal{Z} \to \mathcal{M}$ defined by

$$\texttt{Embedding}(\mathbf{z}_j) = \left[ \left( \mathbf{y}^0 \right)^\top, \left( \mathbf{y}^1 \right)^\top, , \ldots, \left( \mathbf{y}^K \right)^\top \right]^\top, \tag{18}$$

**Algorithm 3** Tree Construction Using Hyperbolic and Diffusion Geometry [37]

---

**Input:** Pairwise distance matrix $\mathbf{M} \in \mathbb{R}^{m \times m}$, the maximal scale $K$, and the scale parameter $\epsilon$
**Output:** A binary tree $B$ with $m$ leaf nodes

**function** $\mathcal{T}(\mathbf{M})$
  $\mathbf{K} \leftarrow \exp(-\mathbf{M}^{\circ 2}/\epsilon), \mathbf{D} \leftarrow \mathtt{diag}(\mathbf{K}), \mathbf{P} \leftarrow \mathbf{K}\mathbf{D}^{-1}$
  **for** $k \in \{0, 1, \ldots, K\}$ and $j \in [m]$ **do**
    $\boldsymbol{\mu}_j^k \leftarrow \mathbf{P}^{2^{-k}} \boldsymbol{\delta}_j$
    $\mathbf{y}_j^k \leftarrow \left[\left(\boldsymbol{\mu}_j^{k \circ 1/2}\right)^\top, 2^{k/2-2}\right]^\top$
  $B \leftarrow \mathtt{leaves}(\{j\} : j \in [m])$
  **for** $j, j' \in [m]$ **do**
    **for** $k \in \{0, 1, \ldots, K\}$ **do**
      $\mathtt{proj}(\mathbf{y}_j^k \vee \mathbf{y}_{j'}^k) \leftarrow \left\| \left[\frac{1}{2}\left(\boldsymbol{\mu}_j^{k \circ 1/2} - \boldsymbol{\mu}_{j'}^{k \circ 1/2}\right)^\top, 2^{k/2-2}\right]^\top \right\|_2$
    $a_{j,j'} \leftarrow \sqrt[(K+1)]{\mathtt{proj}(\mathbf{y}_j^0 \vee \mathbf{y}_{j'}^0) \cdots \mathtt{proj}(\mathbf{y}_j^K \vee \mathbf{y}_{j'}^K)}$
  $\mathcal{S} \leftarrow \{(j, j') \mid \text{sorted by } a_{j,j'}\}$
  **for** $(j, j') \in \mathcal{S}$ **do**
    **if** $j$ and $j'$ are not in the same subtree
      $\mathcal{I}_j \leftarrow$ internal node for node $j$, $\mathcal{I}_{j'} \leftarrow$ internal node for node $j'$
      add an internal node for $\mathcal{I}_j$ and $\mathcal{I}_{j'}$, and assign the geodesic edge weight
  **return** $B$

---

where $\mathcal{M} = \mathbb{H}^{m+1} \times \mathbb{H}^{m+1} \times \ldots \times \mathbb{H}^{m+1}$ of $(K+1)$ elements. The geodesic distance induced in $\mathcal{M}$ is the $\ell_1$ distance on the product manifold $\mathcal{M}$, given by

$$d_{\mathcal{M}}(j, j') = \sum_{k=0}^K 2 \sinh^{-1}\left(2^{-k/2+1} \left\|\mathbf{y}_j^k - \mathbf{y}_{j'}^k\right\|_2\right). \tag{19}$$

**Theorem A.1** (Theorem 1 [50]). *For sufficiently large $K$, the embedding distance is bilipschitz equivalent to the underlying tree distance, i.e., $d_{\mathcal{M}} \simeq d_T$.*

The multi-scale hyperbolic embedding is used to construct a binary tree $\mathcal{T}(\mathbf{M})$ with $m$ leaves [37], where each leaf corresponds to a data point in $\mathcal{Z}$. Pairs of points are merged based on the Riemannian mean [116] of the radius of the geodesic connecting the hyperbolic embeddings $\mathbf{y}_j^k$ and $\mathbf{y}_{j'}^k$ for $k = \{0, \ldots, K\}$. Specifically, at the $k$-th hyperbolic embedding, the orthogonal projection on the $m + 1$-axis in $\mathbb{R}^{m+1}$, which is the radius of the geodesic of $\mathbf{y}_j^k$ and $\mathbf{y}_{j'}^k$, is given by

$$\mathtt{proj}(\mathbf{y}_j^k \vee \mathbf{y}_{j'}^k) = \left\| \left[\frac{1}{2}\left(\boldsymbol{\mu}_j^{k \circ 1/2} - \boldsymbol{\mu}_{j'}^{k \circ 1/2}\right)^\top, 2^{k/2-2}\right]^\top \right\|_2. \tag{20}$$

The Riemannian mean of the orthogonal projection on the $m + 1$-axis in $\mathbb{R}^{m+1}$ across all $K + 1$ hyperbolic embeddings has a closed form

$$\overline{\mathbf{h}}_{j,j'} = [0, \ldots, 0, a_{j,j'}], \text{ where } a_{j,j'} = \sqrt[(K+1)]{\mathtt{proj}(\mathbf{y}_j^0 \vee \mathbf{y}_{j'}^0) \cdots \mathtt{proj}(\mathbf{y}_j^K \vee \mathbf{y}_{j'}^K)}. \tag{21}$$

The values $a_{j,j'}$ serve as hierarchical linkage scores and guide the merging process in constructing the binary tree $B$. Tree edge weights are then assigned using the $\ell_1$ distance on the manifold $\mathcal{M}$, which corresponds to the geodesic distance. The tree construction based on hyperbolic embedding and diffusion densities is summarized in Alg. 3.

**Theorem A.2** (Theorem 1 [37].). *For sufficiently large $K$ and $m$, the tree metric $d_B$ in Alg. 3 is bilipschitz equivalent to the underlying tree metric $d_T$, where $T$ is the ground truth latent tree and $d_T$ is the hidden tree metric between features defined as the length of the shortest path on $T$.*

### A.5 Tree-Based wavelet

Haar wavelets can be constructed directly and adaptively from trees in a straightforward manner [43]. Without loss of generality, we focus here on the Haar basis induced by the feature tree, where the resulting wavelet filter is subsequently applied to the samples. Symmetrically, one can construct a Haar basis from the sample tree and apply the corresponding wavelet filter to the features.

Consider a scalar function $f : \mathbf{X} \to \mathbb{R}$ defined on the data matrix. Let $S = \{f | f : \mathbf{X} \to \mathbb{R}\}$ be the space of all functions on the dataset. For a complete binary tree $\mathcal{T}(\mathbf{M}_c)$ with $m$ leaves, let $\ell = 1, \ldots, L_c$ denote the levels in the tree, where $\ell = 1$ is the root level and $\ell = L_c$ is the leaf level with $m$ leaves. Define $\widetilde{\Upsilon}(\ell, \varepsilon)$ be the set of all leaves in the $\varepsilon$-th subtree of $\mathcal{T}(\mathbf{M}_c)$ at level $\ell$. Define $S_\ell$ as the space of features that are constant across all subtrees at level $\ell$, and let $\mathbf{1_X}$ be a constant function on $\mathbf{X}$ with the value 1. We have the following hierarchy

Figure 3: An illustration of a Haar basis induced by a binary tree.

$$S^1 = \text{Span}(\mathbf{1_X}), \ S^{L_c} = S, \text{ and } S^1 \subset \ldots \subset S^{L_c}. \quad (22)$$

Therefore, the space $S$ can be decomposed as

$$S = \left( \bigoplus_{l=1}^{L_c-1} Q^\ell \right) \bigoplus S^\ell, \quad (23)$$

where $Q^\ell$ is the orthogonal complement of $S^\ell$. As the tree $\mathcal{T}(\mathbf{M}_c)$ is a full binary tree, the Haar-like basis constructed from the tree is essentially the standard Haar basis [40], denoted by $\{\boldsymbol{\beta}_{\ell,\varepsilon} \in \mathbb{R}^m\}$. At level $\ell$, a subtree $\widehat{\Upsilon}_c(\ell, s)$ splits into two sub-subtrees $\widehat{\Upsilon}_c(\ell+1, s_1)$ and $\widehat{\Upsilon}_c(\ell+1, s_2)$. A zero-mean Haar wavelet $\boldsymbol{\beta}_{\ell,s} \in \mathbb{R}^m$ has non-zero values only at the indices corresponding to leaves in the sub-subtrees and is piecewise constant on each of them. The set of these Haar wavelets, along with a constant vector $\boldsymbol{\beta}_0$, is complete and forms an orthonormal Haar basis. We collect all of these basis vectors as columns of a matrix, denoted by $\mathbf{B} \in \mathbb{R}^{m \times m}$. Fig. 3 illustrates a Haar basis induced by a binary tree.

**Proposition A.1** (Function Smoothness and Coefficient Decay [43]). *Let $\boldsymbol{\beta}_{\ell,\varepsilon}$ be the Haar basis function supported on $\widetilde{\Upsilon}(\ell, \varepsilon)$. If the function $f$ is Lipschitz continuous, then for some $C > 0$,*

$$\left| \langle f, \boldsymbol{\beta}_{\ell,\varepsilon} \rangle_S \right| \leq 4 C d_{\mathcal{T}(\mathbf{M}_c)}(\widetilde{\Upsilon}(\ell, \varepsilon)), \quad (24)$$

*where $d_{\mathcal{T}(\mathbf{M}_c)}(\widetilde{\Upsilon}(\ell, \varepsilon))$ is the tree distance between the internal node rooted at $\varepsilon$-subtree to the leaves.*

Prop. A.1 indicates that the smoothness of the samples $\{\mathbf{X}_{i,:}\}$ leads to an exponential decay rate of their wavelet coefficients as a function of the tree level. Consequently, the wavelet coefficients can serve as a measure to assess the quality of the tree representation $\mathcal{T}(\mathbf{M}_c)$.

**Proposition A.2** ($L_1$ Sparsity [43]). *Consider a Haar basis $\boldsymbol{\beta}_\Theta$, where $\Theta \subset S$ and such that $|\boldsymbol{\beta}_\Theta(j)| \leq 1/|\Theta|^{1/2}$. Let $f = \sum_\Theta \alpha_\Theta \boldsymbol{\beta}_\Theta$ and assume $\sum_\Theta |\boldsymbol{\beta}_\Theta| \leq C$. For any $\kappa > 0$, the approximation $\widetilde{f} = \sum_{|\Theta| < \kappa} \alpha_\Theta h_\Theta$, then*

$$\left\| f - \widetilde{f} \right\|_1 = \sum_{j \in [m]} \left| f(j) - \widetilde{f}(j) \right| \leq C\sqrt{\kappa}. \quad (25)$$

Prop. A.2 implies that with $L_1$ approximation, it is sufficient to estimate the coefficients for function approximation. In the Haar domain, estimating these coefficients can be achieved using the fast wavelet transform [117].

Note that when two trees share the same branching, assigning different weights to their leaves changes the inner product that defines orthogonality. The inner product changes, each wavelet rebalances the weighted masses of its children, and coefficient magnitudes scale with the square roots of those

masses. Because the basis is built by successive weighted contrasts, altering the weights cascades through all levels: coarse vectors that once averaged equal-sized parts may now emphasize one side, while fine-scale vectors adjust in the opposite direction to keep the basis orthonormal. Consequently, although the supports of the basis vectors coincide, the weighted tree produces a non-equivalent orthonormal basis and, in turn, different expansions for the data.

## A.6 Wasserstein distance

The Wasserstein distance [46] measures the discrepancy between two probability distributions over a given metric space. It is related to optimal transport (OT) theory [47, 48], where the goal is to quantify the cost of transforming one distribution into another.

Consider two probability distributions $\mu$ and $\nu$ defined on a metric space $(\mathcal{X}, d)$, where $d$ is the ground metric. The Wasserstein distance is defined as

$$\mathrm{OT}(\mu, \nu, d) = \inf_{\gamma \in \Gamma(\mu, \nu)} \int_{\mathcal{X} \times \mathcal{X}} d(x, y) \, \mathrm{d}\gamma(x, y), \tag{26}$$

where $\Gamma(\mu, \nu)$ denotes the set of all joint probability measures on $\mathcal{X} \times \mathcal{X}$ with marginals $\mu$ and $\nu$, i.e.,

$$\gamma(A \times \mathcal{X}) = \mu(A), \quad \gamma(\mathcal{X} \times B) = \nu(B), \tag{27}$$

for all measurable sets $A, B \subseteq \mathcal{X}$. It represents the minimal "cost" to transport the mass of $\mu$ to $\nu$, which is also known as "earth mover's distance" [49, 43].

In the discrete setting, the Wasserstein distance is commonly applied to two discrete probability distributions $\mu = \sum_{i=1}^{n} \mu_i \delta_{x_i}$ and $\nu = \sum_{j=1}^{m} \nu_j \delta_{y_j}$, where $\delta_x$ denotes the Dirac delta function at $x$, and $\mathbf{x} = \{x_i\}_{i=1}^{n}, \mathbf{y} = \{y_j\}_{j=1}^{m}$ are the support points of $\mu$ and $\nu$, respectively. The distributions satisfy $\sum_{i=1}^{n} \mu_i = \sum_{j=1}^{m} \nu_j = 1$. In this case, the Wasserstein distance can be formulated as the solution to the following linear programming problem:

$$\mathrm{OT}(\mu, \nu, d) = \min_{\mathbf{\Gamma} \in \Gamma(\mu, \nu)} \sum_{i=1}^{n} \sum_{j=1}^{m} d(x_i, y_j) \Gamma_{ij}, \tag{28}$$

where $\mathbf{\Gamma} \in \mathbb{R}_{\geq 0}^{n \times m}$ is the transport matrix satisfying the marginal constraints:

$$\sum_{j=1}^{m} \Gamma_{ij} = \mu_i, \quad \sum_{i=1}^{n} \Gamma_{ij} = \nu_j, \quad \forall i, j. \tag{29}$$

When computing the Wasserstein distance between discrete probability distributions, the computational bottleneck often lies in constructing and processing the ground pairwise distance matrix. This matrix encodes the distances between every pair of discrete support points in the distributions, requiring $O(m^2)$ storage and $O(m^3 \log m)$ computation [49]. This complexity arises due to solving a linear program for the optimal transport problem, which scales poorly with the number of points $m$. Consequently, applying optimal transport directly becomes infeasible for large-scale datasets [118].

Several approximations have been proposed to reduce this computational complexity. The Sinkhorn distance [119, 120] introduces an entropy regularization term to the objective function, enabling the use of iterative matrix scaling algorithms in quadratic. Graph-based methods exploit sparsity in the transport graph to simplify the problem structure, while sampling-based approaches approximate distributions using a subset of support points, reducing computational demand [121–123].

## A.7 Tree-Wasserstein distance

The Wasserstein distance requires solving a linear programme whose computational cost is quadratic in the number of support points. A widely used strategy to mitigate this computation cost is to approximate the ground metric with a tree metric, thereby defining the tree-Wasserstein distance (TWD) [32]. Because transport on a tree admits a closed-form solution, TWD can be computed in $O(m)$ time for a tree with $m$ leaves, which makes it attractive for large-scale applications.

Let $T = (V, E, \mathbf{A})$ be a tree with $N_{\mathrm{leaf}}$ leaves. The tree distance $d_T : V \times V \to \mathbb{R}$ is the sum of weights of the edges on the shortest path between any two nodes on $T$. Let $\Upsilon_T(v)$ be the set of nodes

in the subtree of the tree $T$ rooted at the node $v \in V$. For any node $u \in V$, there exists a unique parent $v$ s.t. the weight is defined by the tree distance, i.e., $w_u = d_T(u, v)$ [62]. Given two discrete distributions $\boldsymbol{\rho}_1, \boldsymbol{\rho}_2 \in \mathbb{R}^{N_{\text{leaf}}}$ supported on the tree $T$, the TWD is formally defined by

$$\text{TW}(\boldsymbol{\rho}_1, \boldsymbol{\rho}_2, T) = \sum_{v \in V} w_v \left| \sum_{u \in \Upsilon_T(v)} (\boldsymbol{\rho}_1(u) - \boldsymbol{\rho}_2(u)) \right|, \tag{30}$$

where inner sum represents the net mass that must cross the edge $(v, \texttt{parent}(v))$, weighting by $w_v$, which accumulates the transport cost across the tree $T$.

Given a tree $T$ and probability measures supported on the tree $T$, the TWD computing using $T$ is the Wasserstein distance on the tree metric $d_T$ [35, 34], i.e.,

$$\text{OT}(\boldsymbol{\rho}_1, \boldsymbol{\rho}_2, d_T) = \text{TW}(\boldsymbol{\rho}_1, \boldsymbol{\rho}_2, T). \tag{31}$$

However, it is important to note that most of the TWD methods [32–36] are designed to approximate a Wasserstein distance with an *Euclidean* ground metric. Specifically, their goal is

$$\min_T \, \|\text{OT}(\boldsymbol{\rho}_1, \boldsymbol{\rho}_2, d_E) - \text{TW}(\boldsymbol{\rho}_1, \boldsymbol{\rho}_2, T)\|_2, \tag{32}$$

where $d_E$ denotes the Euclidean metric. Therefore, the tree construction in these methods is used to approximate the *Euclidean* ground metric. This design choice, while useful for accelerating computation, inherently biases the tree structure toward approximating Euclidean distances rather than representing the hierarchical structure of the data, which is the primary focus of our work.

## A.8    Graph convolutional networks

Graph Convolutional Networks (GCNs) [96] have gained significant attention in graph machine learning, where nodes in a graph are typically assumed in Euclidean spaces. By generalizing convolutional operations to graphs, GCNs effectively capture the dependencies between nodes. This capability enables high accuracy in tasks such as node classification, link prediction, and graph classification, making GCNs useful in applications like social networks [98], molecular structures [124, 125], knowledge graphs [126], recommendation systems [127], and drug discovery [128]. A brief overview of the GCN framework is presented below.

Consider a graph $G = (V, E, \mathbf{A})$ with node features $\{\mathbf{x}_i \in \mathbb{R}^m\}_{i=1}^n$, where $V$ is the vertex set containing $n$ nodes, $E \subset V \times V$ is the edge set, and $\mathbf{A} \in \mathbb{R}^{n \times n}$ is the adjacency matrix. Each node $i \in V$ is associated with a feature vector $\mathbf{x}_i \in \mathbb{R}^m$, representing $m$-dimensional node attributes. In each layer of GCN message passing, the graph convolution can be divided into two steps: feature transformation and neighborhood aggregation. Specifically, the feature transform is defined by

$$\mathbf{h}_i^{(\ell)} = \mathbf{J}^{(\ell)} \mathbf{h}_i^{(\ell-1)}, \tag{33}$$

where $\mathbf{h}_i^{(0)} = \mathbf{x}_i$ is the initial feature, and $\mathbf{J}^{(\ell)}$ is a learnable weight matrix. The neighborhood aggregation then updates the representation by

$$\mathbf{h}_i^{(\ell)} = \sigma \left( \mathbf{h}_i^{(\ell)} + \sum_{s \in [[i]]} w_{is} \mathbf{h}_s^{(\ell)} \right), \tag{34}$$

where $[[i]]$ denotes the neighbors of node $i$, $w_{is}$ is a weight associated with the edge between nodes $i$ and $s$, and $\sigma(\cdot)$ is a non-linear activation function.

By stacking multiple such layers, GCNs propagate information through the graph, enabling each node representation to integrate signals from multi-hop neighborhoods. The feature transformation step learns task-specific embeddings, while the aggregation step incorporates structural information from the graph topology.

## A.9    Hyperbolic graph convolutional networks

Hyperbolic Graph Convolutional Networks (HGCNs) [21, 45] generalize Graph Convolutional Networks (GCNs) [96] to hyperbolic spaces, using the inductive bias of hyperbolic geometry to better capture hierarchical structures in graph data. By embedding nodes in a negatively curved space,

HGCNs provide improved representational capacity for graphs with hierarchies, while preserving the scalability of standard GCNs. This extension requires adapting the steps in GCNs, including feature transformation and neighborhood aggregation, to conform with the geometry of hyperbolic space. HGCNs have shown empirical advantages across a range of tasks, including social network analysis, recommendation systems, and biological network modeling, demonstrating the effectiveness of geometric inductive biases in deep learning on non-Euclidean domains. We briefly review the HGCNs framework below.

HGCNs extend GCNs to hyperbolic spaces (e.g., Lorentz model) by embedding node features in hyperbolic geometry and redefining core operations to respect its geometric structure. Below, we outline the key components of the HGCN architecture.

**Mapping Euclidean node features to hyperbolic representations.** HGCNs [21] begin by mapping input Euclidean node features to a Lorentz model using the exponential map at the origin. Specifically, given a Euclidean feature vector $\mathbf{x}_i \in \mathbb{R}^m$, its hyperbolic embedding is initialized as

$$\mathbf{x}_i^{H,(0)} = \mathrm{Exp}_{\mathbf{0}}\left(\left[0, \mathbf{x}_i^\top\right]^\top\right) \in \mathbb{L}^m, \tag{35}$$

where the $\mathrm{Exp}$ is the exponential map.

**Hyperbolic feature transformation.** To perform feature transformations in hyperbolic space, HGCNs [21] use the logarithmic and exponential maps to move between the manifold and the tangent space at the origin. Let $\mathbf{W} \in \mathbb{R}^{m' \times m}$ be a learnable weight matrix. The hyperbolic equivalent of matrix multiplication is defined as

$$\mathbf{W} \otimes \mathbf{x}_i^H = \mathrm{Exp}_{\mathbf{0}}(\mathbf{W}(\mathrm{Log}_{\mathbf{0}}(\mathbf{x}_i^H))). \tag{36}$$

To incorporate a bias term $\mathbf{b} \in \mathbb{R}^{m'}$, HGCNs use parallel transport and define hyperbolic bias addition as

$$\mathbf{x}_i^H \oplus \mathbf{b} = \mathrm{Exp}_{\mathbf{x}_i^H}(\mathrm{PT}_{\mathbf{0} \to \mathbf{x}_i^H}(\mathbf{b})), \tag{37}$$

where $\mathrm{PT}$ is the parallel transport operator. Then, the full hyperbolic feature transformation is defined as

$$\mathbf{x}_i^{(l),H} = \left(\mathbf{W}^{(l)} \otimes \mathbf{x}_i^{(l-1),H}\right) \oplus \mathbf{b}^{(l)}. \tag{38}$$

**Hyperbolic neighborhood aggregation.** Neighborhood aggregation in HGCNs is performed in the tangent space and leverages hyperbolic attention to account for the hierarchical relationships within the graph. Given hyperbolic embeddings $\mathbf{x}_i^H$ and $\mathbf{x}_j^H$, both are first mapped to the tangent space at the origin. The attention weights are computed via an Euclidean Multi-Layer Perceptron (MLP) applied to the concatenation of the tangent vectors

$$w_{ij} = \mathrm{softmax}_{j \in [[i]]}(\mathrm{MLP}(\mathrm{Log}_{\mathbf{0}}(\mathbf{x}_i^H) || \mathrm{Log}_{\mathbf{0}}(\mathbf{x}_j^H))), \tag{39}$$

where $[[i]]$ denotes the neighbors of node $i$, and $||$ indicates concatenation. Aggregation is then performed via a hyperbolic weighted average in the tangent space of $\mathbf{x}_i^H$

$$\mathrm{AGG}(\mathbf{x}_i^H) = \mathrm{Exp}_{\mathbf{x}_i^H}\left(\sum_{j \in [[i]]} w_{ij} \mathrm{Log}_{\mathbf{x}_i^H}(\mathbf{x}_j^H)\right). \tag{40}$$

Then, a non-linear activation is applied in hyperbolic space to complete the layer's update.

**Remark A.1.** In the hyperbolic feature transformation step of HGCNs [21], the standard approach involves mapping points between the hyperbolic manifold and the Euclidean tangent space via logarithmic and exponential maps. This allows feature transformations to be performed using Euclidean operations, such as linear transformations followed by bias addition. However, this design only partially uses the advantage of the geometry of hyperbolic space, as the actual transformation takes place in the flat tangent space rather than directly on the curved manifold. To exploit the structure of hyperbolic space, [45] proposed a fully hyperbolic feature transformation. Specifically, they introduce a Lorentz linear transformation that operates directly in the hyperbolic space

$$\mathbf{x}_i^{(l),H} = \mathbf{J}^{(l)} \mathbf{x}_i^{(l-1),H} \ s.t. \ \mathbf{J}^{(l)} = \begin{bmatrix} 1, & \mathbf{0}^\top \\ \mathbf{0}, & \widetilde{\mathbf{J}}^{(l)} \end{bmatrix} \ and \ (\widetilde{\mathbf{J}}^{(l)})^\top \widetilde{\mathbf{J}}^{(l)} = \mathbf{I}. \tag{41}$$

Here, $\widetilde{\mathbf{J}}^{(l)}$ is an orthogonal matrix, while the first coordinate (the time-like component in Lorentz space) remains unchanged. This formulation preserves the hyperbolic structure and allows feature transformations to be conducted entirely within the manifold. When integrating our method into the HGCN framework, we adopt this hyperbolic-to-hyperbolic feature transformation to maintain geometric consistency.

## B  Extended related works

**Manifold learning with Wasserstein distance.**   Manifold learning is a nonlinear dimensionality reduction approach designed for high-dimensional data that intrinsically lie on a lower-dimensional manifold [58, 51, 114]. Most classical methods derive manifold representations from discrete data samples by constructing a graph, where nodes correspond to data points and edges reflect pairwise similarities. These similarities are often defined using a Gaussian affinity kernel based on the Euclidean norm between the data samples. Recent studies have demonstrated the benefits of using the Wasserstein distance as a more geometry-aware alternative to Euclidean distances in manifold learning frameworks [43, 52], where the Wasserstein distance incorporates the feature relationship. This direction has shown promising results in applications such as matrix organization [129], analysis of neuronal activity patterns [7], and molecular shape space modeling [68]. Building on these ideas, our work explores the integration of the diffusion operator [51] with the TWD to jointly learn hierarchical representations for both samples and features.

**Tree-Wasserstein distance.**   The Tree-Wasserstein Distance (TWD) offers a computationally efficient alternative to the Wasserstein distance by approximating the ground cost metric with a tree structure. It is designed to compare probability distributions by quantifying the amount of mass that must be transported between them, where the transport is governed by distances along a tree. By approximating the original Euclidean ground metric with a tree metric, TWD enables faster computation while maintaining a close approximation to the true Wasserstein distance. This trade-off between efficiency and fidelity has been validated in a range of studies [32–35], where TWD has been shown to provide a reliable and scalable surrogate for optimal transport in Euclidean settings. For instance, the Quadtree [32] recursively partitions the ambient space into hypercubes to build random trees, which are then used to calculate the TWD. Flowtree [88] refines this approach by focusing on optimal flow and its cost within the ground metric. Similarly, the Tree-Sliced Wasserstein Distance (TSWD) [35] further improves robustness by averaging TWD values computed over multiple randomly sampled trees, with variants such as TSWD-1, TSWD-5, and TSWD-10 corresponding to the number of sampled trees used. Recent advancements include WQTWD and WCTWD [34], which employ Quadtree or clustering-based tree structures and optimize tree weights to approximate the Wasserstein distance. UltraTree [36] introduces an ultrametric tree by minimizing OT regression cost, aiming to approximate the Wasserstein distance while respecting the original metric space.

Most recently, a tree construction method [37] in TWD literature that differs fundamentally from existing TWD approaches [32–35]. Whereas conventional TWD methods focus on how TWD can be close to the true Wasserstein distance with the Euclidean ground metric, this method [37] aims to efficiently compute a Wasserstein distance with a ground metric that recovers a latent hierarchical structure underlying hierarchical high-dimensional features. Unlike prior TWD methods that treat tree construction as a heuristic or approximation step to Euclidean distances, this approach [37] treats the tree as a geometric object that reflects intrinsic hierarchical relationships among features. In particular, the constructed tree is not merely a proxy for an ambient Euclidean metric, but rather a data-driven structure whose tree distances approximate geodesic paths on a latent, unobserved hierarchical metric space. This allows the resulting TWD to encode meaningful relationships in settings where the data are governed by hidden hierarchies. In our work, we adopt this tree construction method [37], as it provides a principled and geometry-aware foundation for learning hierarchical representations of both samples and features.

**Sliced-Wasserstein distance.**   Another widely used approach for efficiently approximating the Wasserstein distance is the Sliced-Wasserstein Distance (SWD) [130, 131]. Instead of solving the high-dimensional optimal transport problem directly, SWD projects the input distributions onto one-dimensional subspaces along random directions, where the Wasserstein distance admits a closed-form solution. Averaging the results over many such slices provides a meaningful measure of similarity while retaining much of the geometric structure inherent in the distributions. This approach is

advantageous in high-dimensional settings, where computational efficiency is critical, making it popular in applications such as generative modeling [132], domain adaptation [133], and statistical inference [134].

**Co-Manifold learning.** Co-manifold learning aims to simultaneously uncover the manifolds of both samples and features in a data matrix by treating their relationships as mutually informative. That is, the geometry of the features is informed by the samples, and vice versa. This joint modeling approach has been widely explored in tasks such as joint embedding [52–55] and dimensionality reduction [7, 56, 57], under the assumption that both samples and features lie on smooth, low-dimensional manifolds. By iteratively refining representations across both modes, co-manifold methods aim to capture the manifolds of both samples and features. These approaches typically assume that the manifolds involved are Riemannian and characterized by non-negative curvature [55, 58]. However, such assumptions may be limiting in scenarios where the data exhibit hierarchical organization, which is more naturally modeled in negatively curved spaces such as hyperbolic geometry.

In our work, we adopt the co-manifold learning perspective, using the geometry of the samples to inform the structure of the features and vice versa. However, unlike existing co-manifold approaches that primarily focus on smooth manifolds with non-negative curvature, our focus lies in capturing the hierarchical nature of both samples and features. By modeling these hierarchies explicitly through trees and employing the corresponding TWDs for cross-mode inference, we address a distinct setting where the data is high-dimensional with hierarchical structures.

**Unsupervised ground metric learning.** Ground metric learning [77] focuses on learning a distance function that defines the cost matrix in the optimal transport problem, thereby capturing meaningful relationships between data elements. The choice of ground metric plays a critical role in downstream tasks [135, 136] such as clustering, transport-based inference, and dimensionality reduction. Many existing approaches rely on prior knowledge of the cost matrix structure or access to label information to guide the learning process [137–139]. In many cases, however, labeled data is unavailable, making it necessary to develop unsupervised methods that rely solely on the data itself.

Unsupervised ground metric learning seeks to infer a distance metric that captures meaningful pairwise relationships directly from the data, without relying on external labels or prior knowledge. A recent work in this direction is the Wasserstein Singular Vectors (WSV) method [59], which jointly computes Wasserstein distances over samples and features by using the Wasserstein distance matrix in one domain (e.g., samples) as the ground metric for computing distances in the other (e.g., features). However, the method is computationally demanding due to the repeated evaluation of high-dimensional Wasserstein distances. To address this issue, the recent work proposed Tree-WSV [60], which uses tree-based approximations of the Wasserstein distance to reduce computational cost. Both WSV and Tree-WSV adopt an alternating scheme that iteratively updates distances across rows and columns, a strategy that aligns conceptually with our framework.

When working with hierarchical data, which is the central focus of our work, the Wasserstein metric is not inherently a hierarchical metric. As a result, methods such as WSV and Tree-WSV, which rely respectively on the Wasserstein metric and its tree-based approximation, are limited in their ability to represent hierarchical structures underlying samples and features. While Tree-WSV introduces tree metrics to improve computational efficiency, the trees are constructed to approximate Wasserstein distances, not to represent the hierarchical metric space.

In contrast, our method is explicitly designed to represent the hierarchical structures for both rows and columns. The key distinction lies in our use of the tree construction method [37] in the TWD: we construct the tree which employs hyperbolic embeddings and diffusion densities to reflect meaningful hierarchical relationships among data points. We then integrate this specific tree-based TWD into the construction of a diffusion operator, whose diffusion densities are used to generate hyperbolic embeddings that encode the joint structure of both samples and features. In doing so, we not only model each hierarchical structure but also capture their mutual influence through an alternating learning process.

Although WSV and Tree-WSV share a similar alternating framework, their focus remains on metric approximation for ground metric learning rather than representation learning. In contrast, our approach is centered on jointly learning hierarchical representations for samples and features and enhancing

them through wavelet filtering and integration within HGCNs. This provides additional insights into the hierarchical data that are not present in the WSV or Tree-WSV frameworks.

## C   Proofs

We present the proofs supporting our theoretical analysis in Sec. 4. To aid in the proofs, we include several separately numbered propositions and lemmas.

**Notation.**   Let $\mathcal{H}_m \subset \mathbb{R}_+^{m \times m}$ be a set of pairwise tree distance matrices for $m$ points. That is, given $\mathbf{H} \in \mathcal{H}_m$, the following properties satisfy: (i) there is a weighted tree $T = (V, E, \mathbf{A})$ such that $\mathbf{H}(j, j') = d_T(j, j') \ \forall \ j, j' \in [m]$, (ii) $\mathbf{H}(j, j) = 0 \ \forall \ j \in [m]$, (iii) $\mathbf{H}(j_1, j_2) \leq \mathbf{H}(j_1, j_3) + \mathbf{H}(j_3, j_2) \ \forall \ j_1, j_2, j_3 \in [m]$, and (v) $\mathbf{H}(j, j') = \mathbf{H}(j', j) \ \forall \ j, j' \in [m]$. Let $\mathcal{W}_{m \rightsquigarrow n} \subset \mathbb{R}_+^{n \times n}$ represent the set of pairwise OT distance matrices for $n$ points, where the ground metric is given by the tree distance matrix $\mathbf{H} \in \mathcal{H}_m$ with $m$ points. That is, given $\mathbf{W} \in \mathcal{W}_{m \rightsquigarrow n}$, the following properties satisfy: (i) $\mathbf{W}(i, i') = \mathrm{OT}(\boldsymbol{\mu}_i, \boldsymbol{\mu}_{i'}, \mathbf{H}) = \mathrm{TW}(\boldsymbol{\mu}_i, \boldsymbol{\mu}_{i'}, T)$ for any two probability distributions $\boldsymbol{\mu}_i, \boldsymbol{\mu}_{i'} \in \mathbb{R}^m$ supported on the tree $T$, (ii) $\mathbf{W}(i, i) = 0 \ \forall \ i \in [n]$, (iii) $\mathbf{W}(i_1, i_2) \leq \mathbf{W}(i_1, i_3) + \mathbf{W}(i_3, i_2)$ for any probability distributions $\boldsymbol{\mu}_{i_1}, \boldsymbol{\mu}_{i_2}, \boldsymbol{\mu}_{i_3} \in \mathbb{R}^m$ supported on the tree $T$, and (v) $\mathbf{W}(i, i') = \mathbf{W}(i', i)$ for any two probability distributions $\boldsymbol{\mu}_i, \boldsymbol{\mu}_{i'} \in \mathbb{R}^m$ supported on the tree $T$.

### C.1   Proof of Theorem 1

**Theorem 1.**   *The sequences $\mathbf{W}_r^{(l)}$ and $\mathbf{W}_c^{(l)}$ generated by Alg. 1 have at least one limit point, and all limit points are fixed points if $\gamma_r, \gamma_c > 0$.*

*Proof.* Consider the set $\mathcal{S}_m = \{\mathbf{H}_c \in \mathcal{H}_m | \ \|\mathbf{H}_c\|_\infty \leq \theta_c \text{ and } \mathbf{H}_c(j, j') \geq \widetilde{\theta}_c \ \forall j \neq j'\}$, the Wasserstein distance between samples using a hierarchical ground metric $\mathbf{H}_c$ is bounded by [59]

$$\frac{\widetilde{\theta}_c}{2} \|\mathbf{r}_i - \mathbf{r}_{i'}\|_1 \leq \mathrm{OT}(\mathbf{r}_i, \mathbf{r}_{i'}, \mathbf{H}_c) \leq \frac{\theta_c}{2} \|\mathbf{r}_i - \mathbf{r}_{i'}\|_1,$$

where $\mathbf{r}_i = \mathbf{X}_{i,:}^\top / \|\mathbf{X}_{i,:}\|_1$. Similarly, consider $\mathcal{S}_n = \{\mathbf{H}_r \in \mathcal{H}_n | \ \|\mathbf{H}_r\|_\infty \leq \theta_r \text{ and } \mathbf{H}_r(i, i') \geq \widetilde{\theta}_r \ \forall i \neq i'\}$, the Wasserstein distance between features using $\mathbf{H}_r$ as the hierarchical ground metric is bounded by

$$\frac{\widetilde{\theta}_r}{2} \|\widetilde{\mathbf{c}}_j - \mathbf{c}_{j'}\|_1 \leq \mathrm{OT}(\widetilde{\mathbf{c}}_j, \mathbf{c}_{j'}, \mathbf{H}_r) \leq \frac{\theta_r}{2} \|\widetilde{\mathbf{c}}_j - \mathbf{c}_{j'}\|_1,$$

where $\mathbf{c}_j = \mathbf{X}_{:,j} / \|\mathbf{X}_{:,j}\|_1$. Therefore, the updated TWDs in Eq. (6), where we consider a norm regularizer $\zeta(\cdot)$, can be respectively bounded by

$$\frac{\widetilde{\theta}_c}{2} \|\mathbf{r}_i - \mathbf{r}_{i'}\|_1 + \gamma_r \zeta(\mathbf{r}_i - \mathbf{r}_{i'}) \leq \mathbf{W}_r^{(l)}(i, i') \leq \frac{\theta_c}{2} \|\mathbf{r}_i - \mathbf{r}_{i'}\|_1 + \gamma_r \zeta(\mathbf{r}_i - \mathbf{r}_{i'})$$

and

$$\frac{\widetilde{\theta}_r}{2} \|\mathbf{c}_j - \mathbf{c}_{j'}\|_1 + \gamma_c \zeta(\mathbf{c}_j - \mathbf{c}_{j'}) \leq \mathbf{W}_c^{(l)}(j, j') \leq \frac{\theta_r}{2} \|\mathbf{c}_j - \mathbf{c}_{j'}\|_1 + \gamma_c \zeta(\mathbf{c}_j - \mathbf{c}_{j'}).$$

Moreover, from Prop. C.1 and Lemma C.2, the pairwise tree distances $\mathbf{H}_r$ and $\mathbf{H}_c$ constructed via Eq. (19) are bounded by the Wasserstein metrics:

$$\widetilde{\varrho}_{\mathrm{OT}(\cdot, \cdot, \mathbf{H}_r)} \leq \mathbf{H}_c(j, j') \leq \varrho_{\mathrm{OT}(\cdot, \cdot, \mathbf{H}_r)} \text{ and } \widetilde{\varrho}_{\mathrm{OT}(\cdot, \cdot, \mathbf{H}_c)} \leq \mathbf{H}_r(i, i') \leq \varrho_{\mathrm{OT}(\cdot, \cdot, \mathbf{H}_c)}.$$

Let

$$\mathcal{P}_n = \left\{ \mathbf{W}_r^{(l)} \in \mathcal{W}_{m \rightsquigarrow n} \ \middle| \ \left\|\mathbf{W}_r^{(l)}\right\|_\infty \leq \varrho_{\mathrm{OT}(\cdot, \cdot, \mathbf{H}_c)} \text{ and } \mathbf{W}_r^{(l)}(i, i') \geq \widetilde{\varrho}_{\mathrm{OT}(\cdot, \cdot, \mathbf{H}_c)} \ \forall \ i \neq i' \right\},$$

and

$$\mathcal{P}_m = \left\{ \mathbf{W}_c^{(l)} \in \mathcal{W}_{n \rightsquigarrow m} \ \middle| \ \left\|\mathbf{W}_c^{(l)}\right\|_\infty \leq \varrho_{\mathrm{OT}(\cdot, \cdot, \mathbf{H}_r)} \text{ and } \mathbf{W}_c^{(l)}(j, j') \geq \widetilde{\varrho}_{\mathrm{OT}(\cdot, \cdot, \mathbf{H}_r)} \ \forall \ j \neq j' \right\}.$$

The product set

$$\mathcal{K} = (\mathcal{S}_n \times \mathcal{S}_m) \times (\mathcal{P}_n \times \mathcal{P}_m)$$

is closed, bounded, and convex; hence compact in finite dimensions. Each update step of the iterative scheme remains in $\mathcal{K}$:

$$\mathbf{W}_r^{(l+1)} = \Phi(\widehat{\mathbf{X}}_r; \mathcal{T}(\mathbf{W}_c^{(l)})), \quad \mathbf{W}_c^{(l+1)} = \Phi(\widehat{\mathbf{X}}_c; \mathcal{T}(\mathbf{W}_r^{(l)})).$$

is contained in a compact set. We define the alternating step as the self-map

$$F(\mathbf{W}_r, \mathbf{W}_c) := \left( \Phi(\widehat{\mathbf{X}}_r; \mathcal{T}(\mathbf{W}_c)), \Phi(\widehat{\mathbf{X}}_c; \mathcal{T}(\mathbf{W}_r)) \right).$$

Note that the tree decoder function $\mathcal{T}$ is marginal-separate: there exists $\varepsilon > 0$ such that for every iteration $l$ and every distance pair $(i_1, i_2) \neq (i_1', i_2')$

$$\left| \mathbf{W}_{\cdot}^{(l)}(i_1, i_2) - \mathbf{W}_{\cdot}^{(l)}(i_1', i_2') \right| \geq \varepsilon.$$

On this $\varepsilon$-subset of $\mathcal{K}$, the decoded binary tree is locally constant and thus continuous. Therefore, $F$ is continuous on $\mathcal{K}$ and $F$ is non-expansive [140]. Non-expansiveness, combined with bounded iterates, implies asymptotic regularity. By Opial's lemma, every cluster point is a fixed point. Since the sequence is confined to the compact set $\mathcal{K}$, at least one limit point exists. Therefore, every limit point of the iteration is a fixed point, and the sequences generated by Alg. 1 admit at least one convergent subsequence whose limit is a fixed point of the alternating scheme.

$\square$

**Proposition C.1.** *Let $\| \cdot \|_{\mathcal{R}}$ be a norm on $\mathbb{R}^m$, and let $\mathcal{X} = \{\mathbf{x}_i\}_{i=1}^n \subset \mathbb{R}^m$ be a set of data points. Suppose the diffusion operator is constructed using a Gaussian kernel defined with respect to $\|\cdot\|_{\mathcal{R}}$. Then, for all $i \neq i'$, the embedding distance $d_{\mathcal{M}}(i, i')$ in Eq. (19) admits the bounds*

$$\widetilde{\varrho}_{\mathcal{R}} \leq d_{\mathcal{M}}(i, i') \leq \varrho_{\mathcal{R}}, \tag{42}$$

*where $\varrho_{\mathcal{R}}, \widetilde{\varrho}_{\mathcal{R}} \in \mathbb{R}_{>0}$ are constants determined by the norm $\|\cdot\|_{\mathcal{R}}$.*

*Proof.* Let $c_{\mathcal{R}}, C_{\mathcal{R}} > 0$ be constants such that

$$c_{\mathcal{R}} \leq \|\mathbf{x}_i - \mathbf{x}_{i'}\|_{\mathcal{R}}^2 \leq C_{\mathcal{R}}, \quad \forall i \neq i'.$$

This ensures that the Gaussian affinity matrix

$$\mathbf{K}(i, i') = \exp\left( -\frac{\|\mathbf{x}_i - \mathbf{x}_{i'}\|_{\mathcal{R}}^2}{\epsilon} \right)$$

is bounded as

$$\exp\left( -\frac{C_{\mathcal{R}}}{\epsilon} \right) \leq \mathbf{K}(i, i') \leq \exp\left( -\frac{c_{\mathcal{R}}}{\epsilon} \right).$$

Since the resulting graph is fully connected and the node degrees are uniformly bounded below (i.e., $\deg(i) \geq d_{\min}$), the continuous-time diffusion operator $\mathbf{P}^t = \exp(-t\mathbf{L})$ satisfies the following heat kernel bound [141]:

$$\mathbf{P}^t(i, i') \geq \frac{c_1}{n} \exp\left( -\frac{C_{\mathcal{R}}}{\epsilon t} \right),$$

for some constant $c_1 > 0$, where $\mathbf{L}$ is the Laplacian matrix.

Similarly, using the spectral decomposition of $\mathbf{L}$ and the exponential decay of eigenvalues, we can upper bound

$$\mathbf{P}^t(i, i') \leq \frac{c_2}{n} \exp\left( -\frac{c_{\mathcal{R}}}{\epsilon t} \right)$$

for a constant $c_2 > 0$.

Using the bounds above, we obtain:

$$\left\| \sqrt{\mathbf{P}^t(i, :)} - \sqrt{\mathbf{P}^t(i', :)} \right\|_2 \leq \sqrt{n} \cdot \left( \sqrt{\frac{c_2}{n} \exp\left( -\frac{c_{\mathcal{R}}}{\epsilon t} \right)} - \sqrt{\frac{c_1}{n} \exp\left( -\frac{C_{\mathcal{R}}}{\epsilon t} \right)} \right)$$

and, by Jensen's inequality

$$\left\| \sqrt{\mathbf{P}^t(i,:)} - \sqrt{\mathbf{P}^t(i',:)} \right\|_2 \geq \sqrt{2} \left( 1 - \sqrt{\frac{c_2}{n} \exp\left(-\frac{c_{\mathcal{R}}}{\epsilon t}\right) \frac{c_1}{n} \exp\left(-\frac{C_{\mathcal{R}}}{\epsilon t}\right)} \right)^{1/2}$$

and similarly, an upper bound

$$\left\| \mathbf{P}^t(i,:) - \mathbf{P}^t(i',:) \right\|_1 \leq n \left( \frac{c_2}{n} \exp\left(-\frac{c_{\mathcal{R}}}{\epsilon t}\right) - \frac{c_1}{n} \exp\left(-\frac{C_{\mathcal{R}}}{\epsilon t}\right) \right).$$

The embedding distance in the hyperbolic spaces is constructed from multiscale Hellinger-type distances of the form

$$d_{\mathcal{M}}(i,i') = \sum_{k=0}^{K} 2 \sinh^{-1}\left( 2^{-k/2+1} \left\| \sqrt{\mathbf{P}^{2^{-k}}(i,:)} - \sqrt{\mathbf{P}^{2^{-k}}(i',:)} \right\|_2 \right).$$

Substituting into the definition of $d_{\mathcal{M}}$ and use the monotonicity of we obtain:

$$d_{\mathcal{M}}(i,i') \leq \sum_{k=0}^{K} 2 \sinh^{-1}\left( 2^{-k/2+1} \cdot \sqrt{n} \cdot \left( \sqrt{\frac{c_2}{n} \exp\left(-\frac{c_{\mathcal{R}}}{\epsilon t}\right)} - \sqrt{\frac{c_1}{n} \exp\left(-\frac{C_{\mathcal{R}}}{\epsilon t}\right)} \right) \right) =: \varrho_{\mathcal{R}}.$$

For the lower bound, we use the inequality in Lemma C.1 and based on the result [55, Lemma 3]:

$$d_{\mathcal{M}}(i,i') \geq \sum_{k=0}^{K} 2^{-k/2} \cdot \sqrt{2} \left( 1 - \sqrt{\frac{c_2}{n} \exp\left(-\frac{c_{\mathcal{R}}}{\epsilon t}\right) \frac{c_1}{n} \exp\left(-\frac{C_{\mathcal{R}}}{\epsilon t}\right)} \right)^{1/2} =: \widetilde{\varrho}_{\mathcal{R}}.$$

These define constants $\varrho_{\mathcal{R}}$ and $\widetilde{\varrho}_{\mathcal{R}}$ that depend only on the ambient norm and the Gaussian kernel.

$\square$

**Lemma C.1.** *Let $p, q$ be two probability distributions on $\mathcal{X}$. For a constant $k \in \mathbb{Z}_{\geq 0}$, we have*

$$2 \sinh^{-1}\left( 2^{-k/2+1} \left\| \sqrt{p} - \sqrt{q} \right\|_2 \right) \geq 2^{-k/2} \|p - q\|_1, \tag{43}$$

*where $\left\| \sqrt{p} - \sqrt{q} \right\|_2$ is the unnormalized Hellinger distance between $p$ and $q$.*

**Lemma C.2.** *Fix a norm $\| \cdot \|_{\mathcal{R}}$ on $\mathbb{R}^m$ and a data cloud $\mathcal{X} = \{\mathbf{x}_i\}_{i=1}^n \subset \mathbb{R}^m$. Let $d_B$ denote the tree metric output by Alg. 3. Then there exist positive constants $\widetilde{\varrho}_{\mathcal{R}}'$, $\varrho_{\mathcal{R}}'$, depending only on $\| \cdot \|_{\mathcal{R}}$ and the kernel bandwidth, such that*

$$\widetilde{\varrho}_{\mathcal{R}}' \leq d_B(i,i') \leq \varrho_{\mathcal{R}}', \qquad \forall \, i \neq i'.$$

Thm. A.1 shows that the embedding distance in the hyperbolic spaces $d_{\mathcal{M}}$ is bi-Lipschitz equivalent to the intrinsic tree distance $d_T$ constructed from it: $d_T \simeq d_{\mathcal{M}}$. Thm. A.2 in turn relates the decoded tree metric $d_B$ produced by Alg. 3 to $d_T$, so that $d_B \simeq d_T$. Combining these relations gives $d_B \simeq d_{\mathcal{M}}$. Prop. C.1 provides explicit upper and lower bounds on $d_{\mathcal{M}}$ expressed through $\widetilde{\varrho}_{\mathcal{R}}$ and $\varrho_{\mathcal{R}}$. Transferring those bounds through the bi-Lipschitz chain yields the stated inequalities for $d_B$.

## C.2 Proof of Theorem 2

**Theorem 2.** *The sequences $\widehat{\mathbf{W}}_r^{(l)}$ and $\widehat{\mathbf{W}}_c^{(l)}$ generated by Alg. 2 have at least one limit point, and all limit points are fixed points if $\gamma_r, \gamma_c > 0$.*

*Proof.* The discrete Haar transform is an orthonormal linear operator $H$. At each step the algorithm keeps only a subset of Haar coefficients chosen by an $L_1$-based rule and applies soft-thresholding with thresholds $\vartheta_c$ and $\vartheta_r$. Soft-thresholding $S_\vartheta(\cdot)$ is 1-Lipschitz and non-expansive. Therefore each filter

$$\Psi(\cdot; \mathcal{T}(\widehat{\mathbf{W}}_c^{(l)})) = H^{-1} \circ S_{\vartheta_c} \circ P^{(l)} \circ H,$$

is a composition of operators whose spectral norms do not exceed 1, where $P^{(l)}$ is the projection (or masking) operator acting on the vector of Haar coefficients at the $l$-th iteration. It is analogously for the feature mode. Therefore, for every $l$,

$$\left\| \Psi(\mathbf{X}^{(l)}; \mathcal{T}(\widehat{\mathbf{W}}_c^{(l)})) \right\|_F \leq \left\| \mathbf{X}^{(l)} \right\|_F \ \forall \ l.$$

Because $\mathbf{X}^{(0)}$ and $\mathbf{Z}^{(0)}$ are finite, all subsequent filtered data and, through the convex updates that solve the regularized least-squares sub-problems, all matrices $\widehat{\mathbf{W}}_r^{(l)}$, $\widehat{\mathbf{W}}_c^{(l)}$ remain in a ball of finite radius, which respectively form sets that is convex, closed, and bounded, therefore compact.

Because each of $\Psi(\mathbf{X}^{(l)}; \mathcal{T}(\widehat{\mathbf{W}}_c^{(l)}))$ and $\Psi(\mathbf{Z}^{(l)}; \mathcal{T}(\widehat{\mathbf{W}}_r^{(l)}))$ is firmly non-expansive, their composition is non-expansive, and it yields asymptotic regularity [140]. Together with boundedness, the asymptotic regularity activates Opial's lemma, which guarantees that every cluster point of the sequence is a fixed point. The sequence $\widehat{\mathbf{W}}_r^{(l)}$ and $\widehat{\mathbf{W}}_c^{(l)}$ lies in the compact set. By Bolzano–Weierstrass it has at least one convergent sequence.

$\square$

# D   Incorporating learned hierarchical representation within HGCNs

We introduce an iterative learning scheme for joint hierarchical representation learning based on TWD, where the initial pairwise distance matrices $\mathbf{M}_r$ and $\mathbf{M}_c$ are typically computed using standard data-driven metrics, such as Euclidean distance or the distance based on cosine similarity. A key advantage of our approach is its ability to incorporate prior structural knowledge when a hierarchical structure is available for one of the modes. In such cases, the distance metric of the known hierarchy can be used to initialize the corresponding distance matrix, allowing the iterative scheme to refine the structure of the other mode. We demonstrate the applicability of this formulation to hierarchical graph data and show how it can be integrated into HGCNs [21, 45].

In hierarchical graph data, the input is modeled as a hierarchical graph $H = ([n], E, \mathbf{A})$, where $[n] = \{1, \ldots, n\}$ represents the nodes, and each node is associated with an $m$-dimensional attribute vector. We denote the collection of node features as a data matrix $\mathbf{X} \in \mathbb{R}^{n \times m}$, consistent with the high-dimensional data matrix setting in our framework. In hierarchical graph data, the node hierarchy (corresponding to the sample hierarchy in the data matrix) is known, while the feature structure remains implicit. Recent works have proposed developing graph convolutional networks in hyperbolic space [21, 94, 45, 142], where node features are projected from Euclidean to hyperbolic space.

We show that the hierarchical representations learned by our method can be used as a preprocessing step for HGCNs. Specifically, we first derive multi-scale hyperbolic embeddings from the data matrix $\mathbf{X}$ and use them as hyperbolic node features. During the neighborhood aggregation step, we replace the predefined hierarchical graph with the sample tree obtained from our iterative learning scheme after convergence for neighborhood inference. It enables hierarchy-aware message passing, where the learned tree reflects structured relations among samples and features in a data-driven manner. We provide technical details of this integration below.

## D.1   Hyperbolic feature transformation

It is important to note that the used tree construction [37] decodes a binary tree from multi-scale hyperbolic embeddings (see App. A). As a result, after convergence, the produced hyperbolic embeddings can be directly used as initial hyperbolic node representations in HGCNs. For each node $i$, the multi-scale hyperbolic embedding is denoted by

$$[(\mathbf{y}_i^0)^\top, \ldots, (\mathbf{y}_i^K)^\top]^\top, \tag{44}$$

where each $\mathbf{y}_i^k$ lies in a Poincaré half-space $\mathbb{H}^{m+1}$, forming a point in the product manifold of hyperbolic spaces $\mathbb{H}^{m+1} \times \ldots \times \mathbb{H}^{m+1}$. This representation serves as a hyperbolic transformation of the original Euclidean node features and provides a geometry-aware initialization for downstream hyperbolic graph learning tasks.

The Poincaré half-space $\mathbb{H}^{m+1}$ is advantageous for representing the exponentially increasing scale of diffusion densities in hyperbolic space [50]. However, due to the computational advantages of

performing Riemannian operations in the Lorentz model $\mathbb{L}^m$, and the known equivalence between the Poincaré and Lorentz models with diffeomorphism [110, 143], we adopt the Lorentz model for feature transformation [21, 45]. The hyperbolic feature transformation is performed by first transforming points from the Poincaré half-space to the Lorentz model, followed by applying the Lorentz linear transformation [45] and mapping back to the Poincaré half-space.

To enable efficient hyperbolic feature transformation, we first map each point $\mathbf{y}_i^k$ from the Poincaré half-space to the Lorentz model using

$$\widehat{\mathbf{y}}_i^k = \mathcal{P}(\mathbf{y}_i^k) = \frac{(1 + \|\widetilde{\mathbf{y}}_i^k\|^2, 2\widetilde{\mathbf{y}}_i^k(1), \ldots, 2\widetilde{\mathbf{y}}_i^k(n+1))}{1 - \|\widetilde{\mathbf{y}}_i^k\|^2}, \tag{45}$$

where

$$\widetilde{\mathbf{y}}_i^k = \frac{(2\mathbf{y}_i^k(1), \ldots, 2\mathbf{y}_i^k(n), \|\mathbf{y}_i^k\|^2 - 1)}{\|\mathbf{y}_i^k\|^2 + 2\mathbf{y}_i^k(n+1) + 1}.$$

The hyperbolic feature transformation is then applied via a Lorentz linear map [45]

$$\mathbf{h}_i^k = \mathbf{J}\widehat{\mathbf{y}}_i^k \ \text{ s.t. } \mathbf{J} = \begin{bmatrix} 1, & \mathbf{0}^\top \\ \mathbf{0}, & \widetilde{\mathbf{J}} \end{bmatrix} \text{ and } \widetilde{\mathbf{J}}^\top\widetilde{\mathbf{J}} = \mathbf{I}. \tag{46}$$

This transformation preserves the hyperbolic geometry (i.e., manifold-preserving) and corresponds to a linear feature map in hyperbolic space. After the transformation, the features $\mathbf{h}_i^k$ are projected back to the Poincaré half-space, completing the integration of our learned embeddings into HGCNs.

### D.2 Hyperbolic neighborhood aggregation

Since the hyperbolic feature transformation is manifold-preserving, we use the neighborhood aggregation in hyperbolic space as a weighted Riemannian mean of neighboring points, followed by a non-linear activation function. Specifically, during the aggregation step in HGCNs, we replace the predefined hierarchical graph $H$ with the sample tree inferred by our method after convergence for neighborhood inference. This learned tree serves as a hierarchy-aware structure that guides message passing, capturing data-driven relations among samples and across features.

After mapping the transformed features back to the Poincaré half-space, the hyperbolic neighborhood aggregation for node $i$ at layer $\ell$ and scale $k$ is given by

$$\widehat{\mathbf{h}}_i^{(\ell),k} = \sigma \left( \sum_{s \in [[i]]} \left( w_{is} \mathbf{h}_s^{(\ell),k} \right) \right), \tag{47}$$

where $[[i]]$ denotes the neighborhood of node $i$, $w_{is}$ are aggregation weights, and $\sigma$ is a non-linear activation function. In contrast to the Lorentz model formulation [21, 45], applying non-linear activations in the Poincaré half-space does not violate manifold constraints, as the operations remain in the same space. Notably, the weighted sum within the activation function corresponds to a weighted Riemannian mean in the Poincaré half-space model.

The integrated hyperbolic feature transformation and aggregation are conducted directly on the manifold [45]. This contrasts with the earlier method [21], which relies on projecting points to the tangent space for linear operations and then mapping back. By operating entirely within the hyperbolic manifold, our integrated method maintains geometric fidelity throughout the learning process in HGCNs.

## E  Details on experimental study

We provide a detailed description of the experimental setups and additional implementation details for the experimental study in Sec. 6. The experiments are conducted on NVIDIA DGX A100.

### E.1  Datasets

We evaluate our methods on word-document data, single-cell RNA-sequencing (scRNA-seq data), and hierarchical graph datasets. We report the details of the dataset statistics in Tab. 4.

Table 4: Summary of dataset statistics for word-document, single-cell RNA-sequencing (scRNA-seq), and hierarchical graph benchmarks.

| Dataset | # Samples / # Nodes | # Classes | # Edges | # Features |
|---|---|---|---|---|
| BBCSPORT | 517 | 5 | - | 13,243 BOW |
| TWITTER | 2,176 | 3 | - | 6,344 BOW |
| CLASSIC | 4,965 | 4 | - | 24,277 BOW |
| AMAZON | 5,600 | 4 | - | 42,063 BOW |
| ZEISEL | 3,005 | 47 | - | 4,000 Genes |
| CBMC | 8,617 | 56 | - | 500 Genes |
| DISEASE | 1,044 | 2 | 1,043 | 1,000 |
| AIRPORT | 3,188 | 4 | 18,631 | 4 |
| PUBMED | 19,717 | 3 | 88,651 | 500 |
| CORA | 2,708 | 7 | 5,429 | 1,433 |

**Word-document benchmarks.** We evaluate our method on four standard word-document benchmarks commonly used in the word mover's distance [25] and TWD literature [34–37]: (i) the BBCSPORT dataset, consisting of 13,243 bags of words (BOW) and 517 articles categorized into five sports types, (ii) the TWITTER dataset, comprising 6,344 BOW and 2,176 tweets classified into three types of sentiment, (iii) the CLASSIC dataset, including 24,277 BOW and 4,965 academic papers from four publishers, and (iv) the AMAZON dataset, containing 42,063 BOW and 5,600 reviews of four products. The pre-trained Word2Vec embeddings [144] are considered as the word embedding vector, trained on the Google News dataset, which includes approximately 3 million words and phrases. Word2Vec represents these words and phrases as vectors in $\mathbb{R}^{300}$. The document types serve as labels for classification tasks in Sec. 6.2.

**Single-cell RNA-sequencing data.** Two scRNA-seq datasets [84] are considered:(i) the ZEISEL dataset: From the mouse cortex and hippocampus [145], comprising 4,000 gene markers and 3,005 single cells, and (ii) the CBMC dataset: from a cord blood mononuclear cell study [146], consisting of 500 gene markers and 8,617 single cells. We used the divisive biclustering method [145] to obtain 47 classes for Zeisel and 56 classes for the CBMC. The Gene2Vec [147] is used as the gene embedding vectors [2]. Cell types are used as classification labels in the classification experiments presented in Sec. 6.2.

**Hierarchical graph datasets.** For hierarchical graph data, we consider the following datasets: (i) the DISEASE dataset: constructed by simulating the SIR disease spreading model [78], where node labels indicate infection status and node features indicate susceptibility to the disease [21], (ii) the AIRPORT dataset: a flight network dataset where nodes represent airports and edges represent airline routes. The label of a node indicates the population of the country where the airport is located [21], (iii) the CORA dataset: a citation network containing 2,708 nodes, 5,429 edges, and 1,433 features per node, with papers classified into seven machine learning categories [9], and (iv) the PUBMED dataset: Another citation network with 19,717 nodes, 44,338 edges, and 500 features per node, encompassing three classes of medicine publications [9]. The node labels are used for node classification experiments in Sec. 6.3.

### E.2 Hyperparameters settings

We describe the components required to initialize and configure our method, including the choice of initial distance metrics, the hyperparameter configuration, and the norm regularization.

**Initial distance metric.** The initial pairwise distance matrices $\mathbf{M}_r \in \mathbb{R}^{n \times n}$ and $\mathbf{M}_c \in \mathbb{R}^{m \times m}$ are typically data-driven and constructed from the observations using standard similarity measures such as Euclidean or cosine distances [148]. These initial distances are then used to define a Gaussian kernel for the diffusion operator, as detailed in App. A. For word-document and scRNA-seq datasets, the initial distance matrices, $\mathbf{M}_c \in \mathbb{R}^{m \times m}$ (feature distances) and $\mathbf{M}_r \in \mathbb{R}^{n \times n}$ (sample distances), are computed using cosine similarity in the ambient space, following prior work [149, 150]. Empirically, cosine-based distances yield more stable and reliable tree representations

compared to using Euclidean distances as initial distance, which were found to be less robust and led to decreased performance in our experiments. While different initial distance metrics lead to different tree representations, we observe that the resulting trees from Alg. 1 and Alg. 2 converge to unique representations when the regularization parameters $\gamma_r$ and $\gamma_c$ are sufficiently large (see App. F). However, we emphasize that such large regularization does not necessarily produce the most meaningful hierarchical representations. When $\gamma_r$ and $\gamma_c$ are too large, the regularization terms dominate the distance computation in the iterative learning scheme, leading to trees that may lack meaningful hierarchical structure.

**Hyperparameter configuration.** We explore a broad range of hyperparameters to accommodate variations across tasks. For hyperparameter optimization, we conduct a grid search for each dataset using Optuna [151]. The Gaussian kernel scale is selected from the set $\{0.1, 1, 2, 5, 10\} \times \chi$, where $\chi$ denotes the median of the pairwise distances [152, 153]. The number of hyperbolic components in the product manifold is set with the range $K \in \{0, 1, \ldots, 19\}$. The regularization parameters $\gamma_r$ and $\gamma_c$ are chosen from the set $\{10^{-3}, 5 \times 10^{-3}, 10^{-2}, 5 \times 10^{-2}, 10^{-1}, 5 \times 10^{-1}, 1, 5, 10^1, 5 \times 10^1, 10^2\}$. The thresholds $\vartheta_c$ and $\vartheta_r$ in the filtering step are set as follows. At the first iteration ($l = 0$), the total $L_1$ norms of the Haar coefficients across all samples and all features are computed, denoted by $\eta_c$ and $\eta_r$, respectively. Thresholds $\vartheta_c$ and $\vartheta_r$ are then respectively selected from the sets $\{0.1\eta_c, 0.2\eta_c, \ldots, 0.9\eta_c\}$ and $\{0.1\eta_r, 0.2\eta_r, \ldots, 0.9\eta_r\}$. As the optimal hyperparameter configuration varies across tasks and there is no universal choice that works for all cases, Optuna [151] is used to systematically explore the parameter space and efficiently identify task-specific configurations.

**Norm regularizer.** We adopt a norm regularizer based on the snowflake penalty [57], given by

$$\zeta(\mathbf{r}_i - \mathbf{r}_{i'}) = \frac{1}{2} \int_0^{\|\mathbf{r}_i - \mathbf{r}_{i'}\|_2} \frac{1}{\sqrt{\xi} + 10^{-6}} d\xi. \tag{48}$$

This smoothness-promoting regularizer increases monotonically over $[0, \infty)$ and imposes stronger penalties on small differences, thereby encouraging local regularity in the representations. While alternative regularization strategies, such as entropic regularization [119], could be considered, we demonstrate in App. F that they are less effective.

## E.3 Scaling to large datasets

In the tree construction [37], one key step involves building diffusion operators [51] between both samples and features. Although such construction can be computationally intensive, recent developments in diffusion geometry literature have introduced various techniques to significantly reduce the runtime and space complexity of this process. In particular, one can use the diffusion landmark approach [76], which enhances scalability by reducing the complexity from $O(n^3)$ to $O(n^{1+2c})$ for samples and from $O(m^3)$ to $O(m^{1+2c})$ for features, where $c < 1$ represents the proportional size of the landmark set relative to the original dataset. We briefly describe the diffusion landmark approach below for the setup of the sample diffusion operator and note that it can be seamlessly applied to the feature diffusing operator.

Given a set of samples $\mathcal{X} = \{\mathbf{x}_i\}_{i=1}^n \subseteq \mathbb{R}^m$ in an ambient space $\mathbb{R}^m$, let $\mathcal{X}' = \{\mathbf{x}_i'\}_{i=1}^{i'} \subset \mathcal{X} \subseteq \mathbb{R}^m$ be the landmark set of $\mathcal{A}$, where $c = \log_n(n') < 1$. The affinity matrix between the landmark set $\mathcal{X}'$ and the original set $\mathcal{X}$ is denoted by $\widehat{\mathbf{K}} = \exp(-\widehat{\mathbf{M}}^{\circ 2}/\epsilon)$, where $\widehat{\mathbf{M}}(i, i')$ represents a suitable distance between the sample $\mathbf{x}_i \in \mathcal{X}$ and the sample $\mathbf{x}_i' \in \mathcal{X}'$, and $\epsilon > 0$ is the scale parameter. Let $\widehat{\mathbf{D}}$ be a diagonal matrix, where $\widehat{\mathbf{D}}(i, i) = \boldsymbol{\delta}_i^\top \widehat{\mathbf{K}} \widehat{\mathbf{K}}^\top \mathbf{1}_n$ and $\mathbf{1}_n = [1, \ldots, 1] \in \mathbb{R}^n$. The landmark-affinity matrix $\widehat{\mathbf{Y}} = \widehat{\mathbf{K}} \widehat{\mathbf{K}}^\top \in \mathbb{R}^{n \times n}$ has an eigen-structure similar to the diffusion operator, which can be computed by applying SVD to the matrix $\widehat{\mathbf{D}}^{-1/2} \widehat{\mathbf{K}} = \widehat{\mathbf{U}} \widehat{\mathbf{\Lambda}} \widehat{\mathbf{V}}$. To construct the diffusion operator on $\mathcal{X}$, one can use the landmark set $\mathcal{X}'$ and its eigenvectors.

In our method, for large datasets, we fix the landmark set size to $c = 0.1$, which reduces the computational complexity of tree construction at each iteration to $O(n^{1.2})$ for samples and $O(m^{1.2})$ for features. Additionally, the computation of the TWD is linear, resulting in an overall per-iteration complexity of $O(n^{1.2} + m^{1.2})$. In contrast, a naive approach without diffusion landmarks leads to significantly higher complexity: $O(mn^3)$ for tree construction and $O(n^3 \log n)$ and $O(m^3 \log m)$ for computing Wasserstein distances over rows and columns, respectively. The total per-iteration cost

becomes $O(mn^3 + nm^3 + n^3 \log n + m^3 \log m)$. As shown in Sec. 6, our method scales efficiently and can handle datasets with thousands of samples and features.

### E.4 Experimental setup

We describe the experimental setups for sparse approximation to evaluate the hierarchical representations, for document and single-cell classification using the learned TWD, and for link prediction and node classification on hierarchical graph data.

**Sparse approximation setup.** We assess the learned hierarchical representations using a sparse approximation criterion. Specifically, we compute the $L_1$ norm of the Haar coefficients across all samples and features. Let $\widetilde{\mathbf{B}}_c$ and $\widetilde{\mathbf{B}}_r$ denote the Haar bases induced by the feature tree and sample tree after convergence, respectively. We measure sparsity by computing:

$$\frac{1}{n} \sum_i \sum_j \left| (\mathbf{X}_{i,:} \widetilde{\mathbf{B}}_c)^\top (j) \right|, \quad \frac{1}{m} \sum_j \sum_i \left| (\mathbf{Z}_{j,:} \widetilde{\mathbf{B}}_r)^\top (i) \right|. \tag{49}$$

These quantities reflect the average sparsity of the transformed data in the respective Haar bases. A lower $L_1$ norm indicates a more compact representation, where the signal is concentrated on fewer significant coefficients. This suggests that the learned tree structures more meaningfully represent the hierarchical organization of the data [43, 53].

**Document and single-cell classification setup.** For document and single-cell classification tasks (i.e., sample classification), we use the sample TWD matrix to perform classification. The dataset is split by partitioning the TWD matrix into 70% training samples and 30% testing samples. We apply a $k$-nearest neighbors classifier with $k \in \{1, 3, \ldots, 19\}$. The random split is repeated five times [25, 26], and we report the best average classification accuracy across these runs.

**Link prediction and node classification setup.** We follow experimental setups [21, 45] to maintain consistency. For the link prediction task, the set of edges is split into 85% for training, 5% for validation, and 10% for testing. A Fermi-Dirac decoder [93, 13] is used to compute probability scores for edges, and the model is trained using cross-entropy loss with negative sampling. The link prediction performance is assessed by the area under the ROC curve (AUC). For the node classification task, dataset-specific splits are applied [21, 45]: 70%/15%/15% for AIRPORT, 30%/10%/60% for DISEASE, and the standard setup of 20 training examples per class for CORA and PUBMED. The node classification is performed using a centroid-based method [94], where each class is associated with a prototype and predictions are made using a softmax classifier trained with cross-entropy loss. Additionally, an LP-based regularization objective is incorporated during training [21, 45]. The NC task is evaluated using the F1 score for binary-class datasets and accuracy for multi-class datasets: F1 score for DISEASE, and accuracy for AIRPORT, CORA, and PUBMED.

## F Additional experimental results

We present additional experimental results supporting our method, including a toy problem, ablation studies, empirical convergence, runtime analysis, and co-clustering performance.

### F.1 Toy example: synthetic video recommendation system

We consider a toy example motivated by a video recommendation system, where samples correspond to users and features correspond to videos. Both users and videos are assumed to follow hierarchical structures, and representations are generated through a probabilistic diffusion process over these trees. The video hierarchy is based on content categories. The root node represents all videos and branches into three main types: fiction, documentary, and animation, as illustrated in Fig. 4 (right). Each category further divides into subgenres. For example, fiction splits into action, drama, and sci-fi; documentary into biography and historical documentary; and animation into family and comedy. We generate video embeddings $\{\mathbf{w}_j\}_{j=1}^m \subset \mathbb{R}^d$ using a hierarchical probabilistic model. The root node embedding is drawn from a Gaussian prior, and embeddings for child nodes are recursively sampled from a Gaussian centered at their parent's embedding:

$$\mathbf{w}_{\text{root}} \sim \mathcal{N}(\mathbf{0}, \sigma_0^2 \mathbf{I}), \mathbf{w}_c | \mathbf{w}_p \sim \mathcal{N}(\mathbf{w}_p, \sigma_c^2 \mathbf{I}).$$

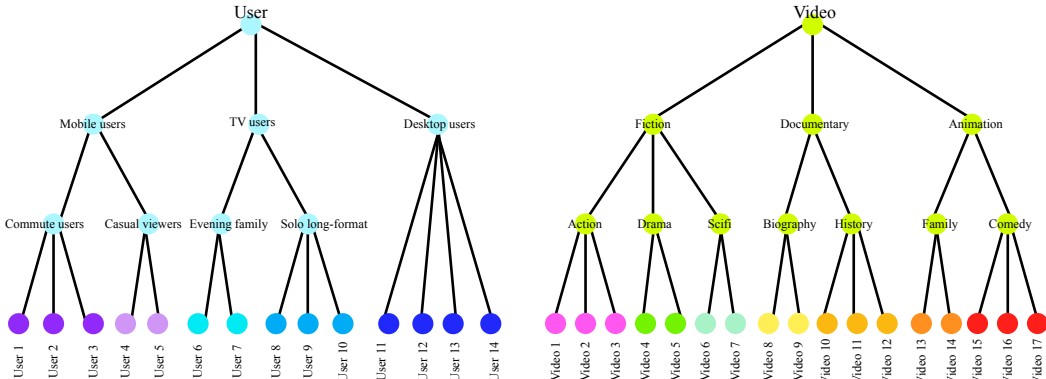

Figure 4: Hierarchical structure used in the toy video recommendation example. The right tree depicts the user hierarchy based on device type and viewing context. The left tree represents the feature hierarchy of videos, branching by genre and subgenre (e.g., fiction → action, drama, sci-fi). Users and videos are colored according to their first-level subcategory.

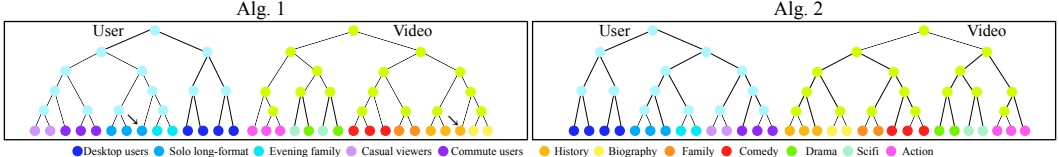

Figure 5: Learned tree representations for the toy video recommendation example using Alg. 1 and Alg. 2. Users and videos are colored according to their first-level subcategory.

This recursive diffusion continues until all leaf nodes (i.e., individual videos) receive their embeddings. As a result, videos belonging to the same (sub)category remain close in the space, reflecting their semantic proximity. A similar process is used to generate user embeddings $\{\mathbf{s}_i\}_{i=1}^{n} \subset \mathbb{R}^d$, based on a hierarchical structure defined by user device and consumption context, shown in Fig. 4 (left). The user embeddings are sampled as:

$$\mathbf{s}_{\text{root}} \sim \mathcal{N}(\mathbf{0}, \sigma_1^2 \mathbf{I}), \mathbf{s}_c | \mathbf{s}_p \sim \mathcal{N}(\mathbf{s}_p, \sigma_r^2 \mathbf{I}).$$

Given user embeddings $\{\mathbf{s}_i\}_{i=1}^{n}$ and movie embeddings $\{\mathbf{w}_j\}_{j=1}^{m}$, the interaction matrix $\mathbf{Y}$ is defined based on their proximity:

$$\mathbf{Y}_{ij} = \|\mathbf{s}_i - \mathbf{w}_j\|_2 + \epsilon_{ij}, \text{ where } \epsilon_{ij} \sim \mathcal{N}(0, \sigma_{\text{noise}}^2).$$

Note that other functions of the distance can also be considered, e.g., elastic potential operator [154]. In our synthetic experiment, we set $\sigma_c = 1$, $\sigma_r = 0.6$, $\sigma_0 = 0.5$, $\sigma_1 = 0.25$, $\sigma_{\text{noise}}$, and $d = 30$. Finally, to simulate an unstructured observation, we apply random permutations [129, 54] to the rows and columns of $\mathbf{Y}$ to obtain the observed data matrix $\mathbf{X}$.

Fig. 5 presents the learned hierarchical representations for users and videos in the example of toy video recommendation. The trees are obtained by Alg. 1 and Alg. 2 after convergence. Leaves are colored according to their first-level subcategory labels. The hierarchical representations learned by Alg. 2 exhibit a clearer structure and a more distinct separation of subcategories, closely aligning with the ground-truth hierarchies shown in Fig. 4. It highlights the benefit of incorporating wavelet filters into the joint hierarchical representation learning process. This visual evidence supports the effectiveness of filtering in learning meaningful hierarchical representations. We remark that in the absence of noise, we observe that both algorithms produce comparable results. In the presence of noise, the tree learned by Alg. 1 deviates slightly from the ground-truth hierarchy, resulting in a less accurate representation (see arrow in Fig. 5). In contrast, Alg. 2 effectively attenuates the noise, leading to a more accurate and coherent hierarchical structure, as shown in Fig. 5.

Fig. 6 shows the trees produced when alternative TWD constructions are integrated into our iterative learning scheme. In contrast to the clear hierarchical structures produced by our approach, these

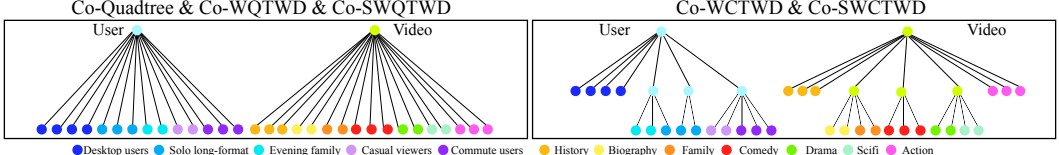

Figure 6: Learned tree representations for the toy video recommendation example using alternative TWD methods in our iterative learning scheme. Users and videos are colored according to their first-level subcategory.

trees fail to represent the hierarchical structures of the samples and the features. This result further illustrates that when using alternative TWD methods in the iterative learning scheme, their trees are not designed for hierarchical representation learning and therefore produce disorganized structures that are not meaningful for visual comparison. In addition, we present quantitative results in Tab. 5 below, reporting the performance on sparse approximation and classification accuracy. The experimental setup follows that of Sec. 6. For the classification tasks, we use the first-level subcategory labels as the ground truth labels. Tab. 5 shows that our method consistently outperforms baselines, validating its effectiveness in representing meaningful hierarchical structures.

Table 5: Quantitative results for the toy example. As in Sec. 6, sparse approximation is measured by the $L_1$ norm of the Haar coefficient expansion, reported in the format samples / features (lowest values in bold, second-lowest values underlined). Classification accuracy for users and videos is reported in the same format, with the highest values in bold and the second-highest underlined.

|  | Sparse Approximation | Classification Accuracy |
|---|---|---|
| Co-Quadtree | 9.2 / 9.9 | 90.1±0.8 / 88.4±0.4 |
| Co-Flowtree | 9.1 / 9.9 | 89.2±0.7 / 89.6±0.5 |
| Co-WCTWD | 9.3 / 9.8 | 82.6±1.0 / 86.4±0.8 |
| Co-WQTWD | 9.3 / 9.7 | 83.1±0.8 / 85.9±0.7 |
| Co-UltraTree | 8.9 / 9.0 | 88.5±1.4 / 82.3±0.9 |
| Co-TSWD-1 | 10.3 / 11.4 | 85.6±1.4 / 83.2±1.1 |
| Co-TSWD-5 | 10.1 / 10.5 | 87.2±0.8 / 84.9±0.9 |
| Co-TSWD-10 | 9.7 / 10.0 | 89.7±0.8 / 87.4±0.8 |
| Co-SWCTWD | 8.2 / 8.9 | 89.4±1.0 / 88.3±0.4 |
| Co-SWQTWD | 8.4 / 8.6 | 88.6±0.2 / 88.4±0.9 |
| Co-MST-TWD | 10.5 / 10.9 | 82.1±1.7 / 80.6±1.3 |
| Co-TR-TWD | 9.8 / 10.2 | 92.6±0.4 / 93.8±0.7 |
| Co-HHC-TWD | 10.3 / 11.4 | 89.2±2.0 / 88.1±1.7 |
| Co-gHHC-TWD | 10.1 / 10.9 | 88.4±1.6 / 87.0±1.3 |
| Co-UltraFit-TWD | 9.7 / 10.3 | 90.1±1.3 / 89.5±1.2 |
| QUE | 7.6 / 8.1 | 93.4±1.7 / 92.5±0.5 |
| WSV | - | 92.5±0.8 / 91.6±0.6 |
| Tree-WSV | 8.4 / 8.7 | 93.9±0.6 / 92.7±0.8 |
| Alg. 1 | 3.2 / 4.0 | 96.2±0.3 / 95.3±0.4 |
| Alg. 2 | **2.5** / **3.1** | **99.4±0.2** / **99.6±0.1** |

## F.2 Empirical uniqueness of hierarchical representations under strong regularization

For sufficiently large $\gamma_r$ and $\gamma_c$, the resulting TWDs are unique. This means that even if the initial distances are random, our methods will converge to unique trees and TWDs. Fig. 7 illustrates that using different random initializations yields identical final expansion coefficients for the Zeisel dataset. However, we would like to point out that such large regularization does not necessarily lead to the best hierarchical representation. When $\gamma_r$ and $\gamma_c$ are too large, the regularization terms dominate the distance computation in the iterative learning scheme, leading to trees that may lack meaningful hierarchical structure. This is evident when comparing these results to those in Fig. 2 in Sec. 6, where moderate regularization (tuned by Optuna [151]) yields more informative trees. The purpose of this experiment is therefore to demonstrate empirical uniqueness under strong regularization, rather than to advocate for such settings in practice.

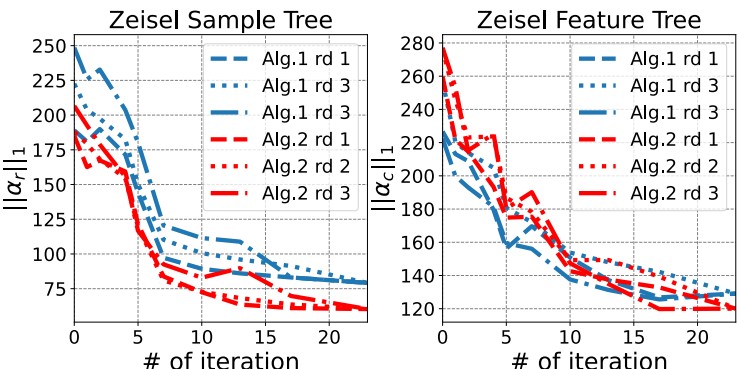

Figure 7: Illustration of empirical uniqueness under strong regularization for ZEISEL datasets. The $L_1$ norm of the Haar coefficients from the sample tree and the feature tree during the sparse approximation task across iterations on the ZEISEL dataset. Despite different random distance initializations, the final expansion coefficients are identical when using sufficiently large $\gamma_r$ and $\gamma_c$, indicating convergence to unique hierarchical representations.

## F.3 Ablation study: effectiveness of iterative learning scheme in classification tasks

To demonstrate the effectiveness of our iterative procedure in improving the joint hierarchical representation learning for both samples and features, we evaluate its impact on the learned TWDs in downstream classification tasks. Specifically, we compare the classification performance on document and single-cell datasets using both the initial TWDs, i.e., computed without any iterative interaction between the sample and feature trees, and the TWDs obtained after applying our full iterative learning scheme. Tab. 6 reports the classification results using the initial TWDs, serving as a non-iterative baseline, alongside the results from our iterative method as reported in Tab. 2. The comparison shows a consistent improvement in classification accuracy when using the TWDs refined through our iterative process. This suggests that alternating updates between sample and feature trees help better inference for the data, regardless of the specific TWD method used. Importantly, our proposed methods Alg. 1 and Alg. 2 not only improve the quality of the learned TWDs but also achieve better performance than all baseline methods. These results highlight the advantage of our approach in jointly refining hierarchical structures across data modes, validating the effectiveness of the proposed iterative learning framework for hierarchical data.

## F.4 Ablation study: incorporating Wavelet filtering with alternative TWD methods

The key difference between Alg. 1 and Alg. 2 is the application of a filtering step in each iteration of Alg. 2, which aims to suppress noise and other nuisance components. This raises the question of whether wavelet filtering could also benefit other TWD-based baseline methods when considering in our iterative learning scheme. To explore this, we apply Haar wavelet filtering using trees constructed from various TWD baselines and report the resulting $L_1$ norms of the Haar coefficients across all samples and features in Tab. 7. For each baseline, we append "-Wavelet" to indicate the variant with wavelet filtering applied. We observe that incorporating wavelet filtering into these baselines does not consistently lead to improved hierarchical representations. We argue that this is because the trees used in these baselines are primarily optimized to approximate Wasserstein distances as ground metrics, rather than to represent the hierarchical structure of the data. As such, they do not serve as meaningful hierarchical representations, since the Wasserstein metric is inherently not a hierarchical metric. Consequently, applying wavelet filters, whose effectiveness depends on the data and the tree structure, does not improve, and may even hinder, hierarchical representation learning.

We further evaluate the effect of wavelet filtering on downstream classification tasks for document and single-cell datasets. Tab. 8 reports the classification accuracies obtained by applying wavelet filtering within the iterative learning scheme using alternative TWD methods. Similarly, for each baseline, we append "-Wavelet" to indicate the variant with wavelet filtering applied. We see that the results show inconsistent improvements across datasets and domains, with any gains being marginal at best. In contrast, when wavelet filtering is applied within our proposed method, the improvement

Table 6: Document and single-cell classification accuracy using the initial TWD (without iterative refinement) and the updated TWD (with iterative refinement). Results marked with * are taken from the work of Lin et al. [37]. Arrows (↑) indicate improvements achieved through the iterative learning process. The best performance is marked in bold, and the second-best is underlined.

| | BBCSPORT | TWITTER | CLASSIC | AMAZON | ZEISEL | CBMC |
|---|---|---|---|---|---|---|
| Quadtree | 95.5±0.5* | 69.6±0.8* | 95.9±0.4* | 89.3±0.3* | 80.1±1.2* | 80.6±0.6* |
| Co-Quadtree | 96.2±0.4 (↑) | 69.6±0.3 | 95.9±0.2 | 89.4±0.2 (↑) | 81.7±1.0 (↑) | 80.7±0.3 (↑) |
| Flowtree | 95.3±1.1* | 70.2±0.9* | 94.4±0.6* | 90.1±0.3* | 81.7±0.9* | 81.8±0.9* |
| Co-Flowtree | 95.7±0.9 (↑) | 71.5±0.7 (↑) | 95.6±0.5 (↑) | 91.4±0.4 (↑) | 84.3±0.7 (↑) | 83.0±1.2 (↑) |
| WCTWD | 92.6±2.1* | 69.1±2.6* | 93.7±2.9* | 88.2±1.4* | 81.3±4.9* | 78.4±3.3* |
| Co-WCTWD | 93.2±1.2 (↑) | 70.2±2.1 (↑) | 94.7±2.6 (↑) | 87.4±1.0 | 82.5±2.9 (↑) | 79.4±2.1 (↑) |
| WQTWD | 94.3±1.7* | 69.4±2.4* | 94.6±3.2* | 87.4±1.8* | 80.9±3.5* | 79.1±3.0* |
| Co-WQTWD | 95.7±1.8 (↑) | 70.7±2.2 (↑) | 95.5±1.3 (↑) | 88.2±2.1 (↑) | 82.3±3.1 (↑) | 80.5±2.8 (↑) |
| UltraTree | 93.1±1.5* | 68.1±3.2* | 92.3±1.9* | 86.2±3.1* | 83.9±1.6* | 82.3±2.6* |
| Co-UltraTree | 95.3±1.4 (↑) | 70.1±2.8 (↑) | 93.6±2.0 (↑) | 86.5±2.8 (↑) | 85.8±1.1 (↑) | 84.6±1.3 (↑) |
| TSWD-1 | 87.6±1.9* | 69.8±1.3* | 94.5±0.5* | 85.5±0.6* | 79.6±1.8* | 72.6±1.8* |
| Co-TSWD-1 | 88.2±1.4 (↑) | 70.4±1.2 (↑) | 94.7±0.9 (↑) | 86.1±0.5 (↑) | 80.2±1.4 (↑) | 73.2±1.0 |
| TSWD-5 | 88.1±1.3* | 70.5±1.1* | 95.9±0.4* | 90.8±0.1* | 81.3±1.4* | 74.9±1.1* |
| Co-TSWD-5 | 88.7±1.7 (↑) | 71.0±1.5 (↑) | 96.7±0.8 (↑) | 91.5±0.4 (↑) | 82.0±0.9 (↑) | 75.4±0.7 (↑) |
| TSWD-10 | 88.6±0.9* | 70.7±1.3* | 95.9±0.6* | 91.1±0.5* | 83.2±0.8* | 76.5±0.7* |
| Co-TSWD-10 | 89.2±1.1 (↑) | 71.4±1.8 (↑) | 95.5±0.2 | 91.8±0.7 (↑) | 83.8±0.5 (↑) | 77.2±0.9 (↑) |
| SWCTWD | 92.8±1.2* | 70.2±1.2* | 94.1±1.8* | 90.2±1.2* | 81.9±3.1* | 78.3±1.7* |
| Co-SWCTWD | 93.5±2.4 (↑) | 70.5±1.0 (↑) | 94.4±1.3 (↑) | 90.7±1.5 (↑) | 82.7±1.7 (↑) | 79.0±0.9 (↑) |
| SWQTWD | 94.5±1.0* | 70.6±1.9* | 95.4±2.0* | 89.8±1.1* | 80.7±2.5* | 79.8±2.5* |
| Co-SWQTWD | 96.2±1.2 (↑) | 72.4±2.1 (↑) | 96.0±1.1 (↑) | 90.6±2.3 (↑) | 82.4±1.4 (↑) | 81.3±1.1 (↑) |
| MST-TWD | 88.4±1.9* | 68.2±1.9* | 90.0±3.1* | 86.4±1.2* | 80.1±3.1* | 76.2±2.5* |
| Co-MST-TWD | 88.7±2.4 (↑) | 68.4±3.3 (↑) | 91.3±2.9 (↑) | 87.1±1.4 (↑) | 80.1±2.8 | 76.5±1.3 (↑) |
| TR-TWD | 89.2±0.9* | 70.2±0.7* | 92.9±0.8* | 88.7±1.1* | 80.3±0.7* | 78.4±1.2* |
| Co-TR-TWD | 89.5±1.2 (↑) | 70.9±1.7 (↑) | 93.4±2.2 (↑) | 89.5±1.4 (↑) | 80.7±0.8 (↑) | 78.5±0.9 (↑) |
| HHC-TWD | 85.3±1.8* | 70.4±0.4* | 93.4±0.8* | 88.5±0.7* | 82.3±0.7* | 77.3±1.1* |
| Co-HHC-TWD | 86.1±2.1 (↑) | 70.1±1.3 (↑) | 93.6±1.5 (↑) | 88.5±0.5 | 83.2±1.4 (↑) | 77.6±0.8 (↑) |
| gHHC-TWD | 83.2±2.4* | 69.9±1.8* | 90.3±2.2* | 86.9±2.0* | 79.4±1.9* | 73.6±1.6* |
| Co-gHHC-TWD | 84.0±2.0 (↑) | 70.4±1.6 (↑) | 90.7±1.7 (↑) | 87.2±1.9 (↑) | 79.9±1.4 (↑) | 84.2±1.2 (↑) |
| UltraFit-TWD | 84.9±1.4* | 69.5±1.2* | 91.6±0.9* | 87.4±1.6* | 81.9±3.3* | 77.8±1.2* |
| Co-UltraFit-TWD | 86.8±0.9 (↑) | 70.9±1.1 (↑) | 91.9±1.0 (↑) | 89.9±2.0 (↑) | 83.7±2.9 (↑) | 79.1±1.8 (↑) |
| WMD | 95.4±0.7* | 71.3±0.6* | 97.2±0.1* | 92.6±0.3* | - | - |
| GMD | - | - | - | - | 84.2±0.7* | 81.4±0.7* |
| HD-TWD | 96.1±0.4* | 73.4±0.2* | 96.9±0.2* | 93.1±0.4* | 89.1±0.4* | 84.3±0.3* |
| Alg. 1 | 96.7±0.3 (↑) | 74.1±0.5 (↑) | 97.3±0.2 (↑) | 94.0±0.4 | 90.1±0.4 (↑) | 86.7±0.5 (↑) |
| Alg. 2 | **97.3**±0.5 (↑) | **76.7**±0.7 (↑) | **97.6**±0.1 (↑) | **94.2**±0.2 (↑) | **94.0**±0.6 (↑) | **93.3**±0.7 (↑) |

is significantly more pronounced. These findings are consistent with the sparse approximation results in Tab. 7, reinforcing the observation that wavelet filtering effectively improves hierarchical representation learning only when applied to trees that meaningfully represent the data hierarchy.

## F.5 Ablation study: integrating wavelet filters to HGCNs without iterative learning scheme

In Sec. 5, we demonstrate that our method can serve as a preprocessing step for hierarchical graph data and be integrated into HGCNs. The implementation details are provided in App. D. Here, we examine whether wavelet filtering alone, without our iterative learning scheme, can also improve the performance of HGCNs.

It is important to note that Haar wavelet filters are induced by a tree structure [43]. However, in the case of hierarchical graph data, the structure is not always a tree. To quantify how closely a graph resembles a tree, one can use the notion of $\delta$-hyperbolicity [107]. A tree has $\delta = 0$, whereas hierarchical graphs that deviate from tree-like geometry exhibit larger $\delta$-hyperbolicity values,

Table 7: The $L_1$ norm of the Haar coefficients across all samples and features when applying wavelet filtering induced by trees constructed from various TWD-based baselines. Lower values indicate sparser representations. Arrows ($\uparrow$) denote cases where wavelet filtering leads to an improvement (i.e., reduction in $L_1$ norm) compared to the unfiltered baseline. Values are reported in the format samples / features (the lowest in bold and the second lowest underlined).

| | BBCSPORT | TWITTER | CLASSIC | AMAZON | ZEISEL | CBMC |
|---|---|---|---|---|---|---|
| Co-Quadtree | 26.6 / 27.8 | 59.5 / 24.9 | 64.3 / 108.7 | 87.3 / 102.7 | 157.0 / 242.4 | 1616.6 / 77.3 |
| Co-Quadtree-Wavelet | 29.4 / 29.7 | 75.8 / 35.4 | 65.9 / 120.8 | 86.1 ($\uparrow$) / 106.3 | 173.9 / 214.7 ($\uparrow$) | 1792.4 / 89.2 |
| Co-Flowtree | 27.4 / 31.5 | 69.1 / 20.9 | 73.7 / 97.5 | 88.1 / 110.1 | 173.7 / 240.0 | 1688.8 / 81.7 |
| Co-Flowtree-Wavelet | 26.3 ($\uparrow$) / 31.9 | 69.9 / 22.7 | 74.8 /100.1 | 87.3 ($\uparrow$) / 109.2 ($\uparrow$) | 180.4 / 245.1 | 1537.9 ($\uparrow$) / 89.5 |
| Co-WCTWD | 26.5 / 26.7 | 57.6 / 17.1 | 63.3 / 81.7 | 77.4 / 96.4 | 136.4 / 202.9 | 1308.1 / 68.4 |
| Co-WCTWD-Wavelet | 28.1 / 28.4 | 58.5 / 20.7 | 67.1 / 84.3 | 80.9 / 103.4 | 139.6 / 211.7 | 1412.3 / 79.6 |
| Co-WQTWD | 25.2 / 32.1 | 56.9 / 30.2 | 61.3 / 100.1 | 74.7 / 111.0 | 135.0 / 224.8 | 1324.8 / 67.4 |
| Co-WQTWD-Wavelet | 27.1 / 40.3 | 63.2 / 37.1 | 68.7 / 121.3 | 80.9 / 114.7 | 149.2 / 237.6 | 1417.5 / 79.6 |
| Co-UltraTree | 37.6 / 32.1 | 69.8 / 18.8 | 76.0 / 125.5 | 86.3 / 133.3 | 155.4 / 226.6 | 1450.0 / 82.2 |
| Co-UltraTree-Wavelet | 42.5 / 28.3 ($\uparrow$) | 73.1 / 25.9 | 80.4 / 137.2 | 93.4 / 155.0 | 172.3 / 257.8 | 1524.6 / 95.4 |
| Co-TSWD-1 | 26.0 / 29.6 | 70.1 / 15.0 | 69.8 / 91.6 | 100.4 / 111.2 | 179.5 / 211.5 | 1716.1 / 79.5 |
| Co-TSWD-1-Wavelet | 29.7 / 34.5 | 76.2 / 14.8 ($\uparrow$) | 73.9 / 98.4 | 103.6 / 129.1 | 184.7 / 239.4 | 1788.5 / 94.3 |
| Co-TSWD-5 | 30.5 / 24.7 | 58.9 / 16.0 | 65.7 / 98.0 | 83.3 / 102.3 | 170.2 / 234.7 | 1560.2 / 70.7 |
| Co-TSWD-5-Wavelet | 33.1 / 28.0 | 64.2 / 19.1 | 66.7 / 104.5 | 90.5 / 121.3 | 196.7 / 251.2 | 1795.4 / 93.1 |
| Co-TSWD-10 | 21.1 / 34.5 | 52.6 / 23.6 | 59.3 / 140.8 | 74.5 / 135.8 | 141.4 / 229.7 | 1373.3 / 81.2 |
| Co-TSWD-10-Wavelet | 24.5 / 37.9 | 55.7 / 27.0 | 63.4 / 151.2 | 73.9 ($\uparrow$) / 142.7 | 153.6 / 232.4 | 1425.6 / 79.2 ($\uparrow$) |
| Co-SWCTWD | 35.0 / 29.5 | 69.0 / 24.7 | 79.8 / 93.6 | 95.9 / 104.6 | 170.4 / 215.9 | 1595.0 / 89.7 |
| Co-SWCTWD-Wavelet | 36.1 / 31.4 | 72.4 / 33.9 | 84.6 / 96.7 | 102.3 / 124.1 | 169.5 ($\uparrow$) / 224.7 | 1723.4 / 94.6 |
| Co-SWQTWD | 33.5 / 25.6 | 57.6 / 13.9 | 61.3 / 86.0 | 77.0 / 98.9 | 141.8 / 209.6 | 1369.8 / 68.0 |
| Co-SWQTWD-Wavelet | 34.9 / 27.7 | 63.4 / 14.8 | 62.7 / 92.4 | 86.5 / 106.7 | 149.2 / 203.1 ($\uparrow$) | 1428.4 / 70.3 |
| Co-MST-TWD | 30.4 / 34.1 | 69.1 / 22.1 | 77.5 / 109.3 | 97.0 / 126.4 | 179.1 / 254.0 | 1755.3 / 83.7 |
| Co-MST-TWD-Wavelet | 39.4 / 38.6 | 73.2 / 26.4 | 87.1 / 123.0 | 105.6 / 139.7 | 192.8 / 276.3 | 1863.7 / 94.2 |
| Co-TR-TWD | 36.8 / 24.5 | 59.1 / 15.2 | 66.1 / 94.7 | 82.2 / 102.3 | 124.6 / 238.6 | 926.8 / 68.8 |
| Co-TR-TWD-Wavelet | 34.2 ($\uparrow$) / 25.6 | 59.3 / 19.2 | 69.5 / 92.7 ($\uparrow$) | 84.6 / 105.7 | 139.6 / 274.8 | 1031.9 / 72.3 |
| Co-HHC-TWD | 22.5 / 22.7 | 51.5 / 13.6 | 58.7 / 93.3 | 74.1 / 110.1 | 192.1 / 215.5 | 1148.8 / 79.0 |
| Co-HHC-TWD-Wavelet | 28.7 / 36.1 | 53.4 / 19.6 | 64.2 / 100.4 | 86.5 / 126.8 | 203.1 / 276.4 | 1253.6 / 86.1 |
| Co-gHHC-TWD | 27.9 / 10.6 | 65.0 / 16.7 | 72.8 / 112.9 | 88.5 / 115.5 | 162.7 / 240.7 | 1612.8 / 70.8 |
| Co-gHHC-TWD-Wavelet | 32.6 / 16.1 | 78.2 / 23.4 | 84.1 / 125.3 | 96.0 / 139.7 | 182.6 / 277.4 | 1823.5 / 84.4 |
| Co-UltraFit-TWD | 22.5 / 22.1 | 51.9 / 13.4 | 58.4 / 81.3 | 73.1 / 92.8 | 133.7 / 202.0 | 1294.6 / 78.1 |
| Co-UltraFit-TWD-Wavelet | 26.4 / 29.3 | 58.2 / 19.6 | 64.3 / 89.2 | 79.6 / 104.8 | 159.4 / 241.0 | 1432.5 / 89.0 |
| QUE | 22.8 / 19.8 | 57.8 / 10.4 | 71.6 / 67.6 | 88.8 / 72.0 | 93.1 / 173.3 | 906.4 / 58.5 |
| QUE | 23.1 / 17.6 ($\uparrow$) | 56.9 ($\uparrow$) / 13.7 | 74.2 / 69.4 | 90.6 / 78.5 | 94.5 / 162.7 ($\uparrow$) | 802.4 ($\uparrow$) / 69.0 |
| Tree-WSV | 23.6 / 24.9 | 54.3 / 18.2 | 65.4 / 99.2 | 84.2 / 106.3 | 139.7 / 201.9 | 1637.4 / 73.2 |
| Tree-WSV-Wavelet | 27.2 / 29.1 | 63.4 / 22.5 | 67.2 / 105.6 | 93.1 / 120.4 | 145.7 / 212.3 | 1739.2 / 79.8 |
| Alg. 1 | 12.9 / 7.0 | 25.4 / 3.7 | 40.4 / 24.6 | 51.0 / 30.1 | 64.4 / 110.8 | 511.3 / 50.1 |
| Alg. 2 | 10.1 ($\uparrow$) / 4.8 ($\uparrow$) | 22.4 ($\uparrow$) / 3.4 ($\uparrow$) | 37.2 ($\uparrow$) / 19.4 ($\uparrow$) | 46.6 ($\uparrow$) / 26.6 ($\uparrow$) | 50.9 ($\uparrow$) / 93.7 ($\uparrow$) | 489.4 ($\uparrow$) / 45.6 ($\uparrow$) |

indicating lower tree-likeness or increased curvature. For the hierarchical graph datasets considered in our experiments, following prior work on HGCNs [21, 45], the measured $\delta$-hyperbolicity values are as follows: DISEASE ($\delta = 0$), AIRPORT ($\delta = 1$), PUBMED ($\delta = 3.5$), and CORA ($\delta = 11$). Among them, only DISEASE forms a tree, allowing the Haar wavelet to be applied directly. For the other datasets, which are not trees, we can either convert the graph into a tree before applying Haar wavelets or construct Haar-like wavelets using the multiscale tree approximation framework [43].

For the naive transformation, we consider two strategies to approximate the graph with a tree structure: (i) we construct an MST by connecting all nodes using edges with the smallest cumulative weights; (ii) we apply agglomerative clustering to iteratively merge nodes and form a hierarchical tree, as in standard hierarchical clustering. These transformations alter the original graph topology, but the resulting tree is intended to closely approximate the structure of the original graph. Once the tree is constructed, it can be used to define a Haar wavelet basis and build the corresponding wavelet filters, as described in Sec. 4.2, which are then applied to the node attributes (i.e., features). To construct Haar-like wavelets using the multiscale tree approximation framework [43], we first build a hierarchical partition tree over the data using a suitable distance metric (e.g., shortest path distance). Each wavelet basis function is then defined over a node in the tree by assigning opposite-signed, normalized values to its children. We denote wavelet filtering applied to the node attributes (i.e., features) using these approximations, without our iterative learning scheme, by "-MST" (for MST-based tree), "-HC" (for hierarchical clustering), and "-Wavelet" (for multiscale tree approximation). We incorporate these variants into HGCNs [21, 45] and evaluate their performance on link prediction and node classification tasks.

Table 8: Classification accuracy on document and single-cell datasets using various TWD-based baselines with and without wavelet filtering, applied within the iterative learning scheme. The "-Wavelet" suffix denotes the inclusion of Haar wavelet filtering. Arrows (↑) denote cases where wavelet filtering leads to an improvement compared to the unfiltered baseline. While some baselines show marginal improvement, the gains are inconsistent across datasets, highlighting the importance of using meaningful hierarchical structures for wavelet filtering to enhance representation learning. The best performance is marked in bold, and the second-best is underlined.

| | BBCSPORT | TWITTER | CLASSIC | AMAZON | ZEISEL | CBMC |
|---|---|---|---|---|---|---|
| Co-Quadtree | 96.2±0.4 | 69.6±0.3 | 95.9±0.2 | 89.4±0.2 | 81.7±1.0 | 80.7±0.3 |
| Co-Quadtree-Wavelet | 95.6±0.6 | 69.6±0.2 | 96.0±0.1 (↑) | 89.4±1.0 | 81.4±1.1 | 80.6±0.4 (↑) |
| Co-Flowtree | 95.7±0.9 | 71.5±0.7 | 95.6±0.5 | 91.4±0.4 | 84.3±0.7 | 83.0±1.2 |
| Co-Flowtree-Wavelet | 95.5±0.8 | 71.0± 0.8 | 95.0±0.7 | 90.8±0.5 | 82.7±0.8 | 82.2±1.0 |
| Co-WCTWD | 93.2±1.2 | 70.2±2.1 | 94.7±2.6 | 87.4±1.0 | 82.5±2.9 | 79.4±2.1 |
| Co-WCTWD-Wavelet | 93.2±1.1 | 70.6±1.9 (↑) | 94.5±2.2 | 88.3±1.3 (↑) | 82.2±3.0 | 79.0±2.9 |
| Co-WQTWD | 95.7±1.8 | 70.7±2.2 | 95.5±1.3 | 88.2±2.1 | 82.3±3.1 | 80.5±2.8 |
| Co-WQTWD-Wavelet | 95.0±1.6 | 70.2±1.8 | 95.2± 1.8 | 88.2±1.9 | 81.9±3.0 | 80.7±2.7 (↑) |
| Co-UltraTree | 95.3±1.4 | 70.1±2.8 | 93.6±2.0 | 86.5±2.8 | 85.8±1.1 | 84.6±1.3 |
| Co-UltraTree-Wavelet | 95.0±1.5 | 70.4±2.9 (↑) | 93.0±1.8 | 86.3±2.7 | 85.7±1.3 | 83.9±2.0 |
| Co-TSWD-1 | 88.2±1.4 | 70.4±1.2 | 94.7±0.9 | 86.1±0.5 | 80.2±1.4 | 73.2±1.0 |
| Co-TSWD-1-Wavelet | 88.0±1.3 | 70.1±1.2 | 94.6±1.0 | 85.7±0.7 | 80.1±1.2 | 72.9±1.1 |
| Co-TSWD-5 | 88.7±1.7 | 71.0±1.5 | 96.7±0.8 | 91.5±0.4 | 82.0±0.9 | 75.4±0.7 |
| Co-TSWD-5-Wavelet | 88.5±1.4 | 70.9±1.3 | 96.5±0.7 | 91.1±0.6 | 82.3±1.0 (↑) | 75.6±0.9 (↑) |
| Co-TSWD-10 | 89.2±1.1 | 71.4±1.8 | 95.5±0.2 | 91.8±0.7 | 83.8±0.5 | 77.2±0.9 |
| Co-TSWD-10-Wavelet | 89.0±0.9 | 71.2±1.4 | 95.4±0.5 | 91.3±0.8 | 83.7±0.7 | 76.9±1.0 |
| Co-SWCTWD | 93.5±2.4 | 70.5±1.0 | 94.4±1.3 | 90.7±1.5 | 82.7±1.7 | 79.0±0.9 |
| Co-SWCTWD-Wavelet | 93.2±1.6 | 71.3±0.9 (↑) | 94.2±1.5 | 90.3±1.7 | 82.3±2.0 | 79.0±0.5 |
| Co-SWQTWD | 96.2±1.2 | 72.4±2.1 | 96.0±1.1 | 90.6±2.3 | 82.4±1.4 | 81.3±1.1 |
| Co-SWQTWD-Wavelet | 95.9±0.9 | 71.8±1.6 | 95.6±1.3 | 90.2±2.0 | 82.0±1.7 | 80.9±1.4 |
| Co-MST-TWD | 88.7±2.4 | 68.4±3.3 | 91.3±2.9 | 87.1±1.4 | 80.1±2.8 | 76.5±1.3 |
| Co-MST-TWD-Wavelet | 88.6±1.3 | 68.7±2.9 (↑) | 90.8±2.7 | 86.9±1.3 | 80.2±2.6 (↑) | 76.1±2.1 |
| Co-TR-TWD | 89.5±1.2 | 70.9±1.7 | 93.4±2.2 | 89.5±1.4 | 80.7±0.8 | 78.5±0.9 |
| Co-TR-TWD-Wavelet | 90.6±0.7 (↑) | 70.9±1.3 | 93.2±1.9 | 89.0±1.3 | 80.5±0.9 | 78.5±0.7 |
| Co-HHC-TWD | 86.1±2.1 | 70.1±1.3 | 93.6±1.5 | 88.5±0.5 | 83.2±1.4 | 77.6±0.8 |
| Co-HHC-TWD-Wavelet | 85.9±1.7 | 70.4±1.2 (↑) | 93.4±1.1 | 89.2±0.9 (↑) | 82.7±1.0 | 77.5±0.9 |
| Co-gHHC-TWD | 84.0±2.0 | 70.4±1.6 | 90.7±1.7 | 87.2±1.9 | 79.9±1.4 | 84.2±1.2 |
| Co-gHHC-TWD-Wavelet | 83.5±1.7 | 70.2±1.9 | 90.6±2.1 | 87.0±1.4 | 82.7±2.5 | 78.6±1.6 |
| Co-UltraFit-TWD | 86.8±0.9 | 70.9±1.1 | 91.9±1.0 | 89.9±2.0 | 83.7±2.9 | 79.1±1.8 |
| Co-UltraFit-TWD-Wavelet | 86.7±1.2 | 70.3±1.4 | 92.1±0.9 (↑) | 88.7±1.8 | 83.4±2.7 | 79.0±1.7 |
| QUE | 84.7±0.5 | 72.4±0.6 | 91.9±0.5 | 91.6±0.9 | 83.6±1.4 | 82.5±1.9 |
| QUE-Wavelet | 86.4±0.7 (↑) | 72.0±0.8 | 92.3±0.8 (↑) | 90.8±1.0 | 85.1±1.0 (↑) | 82.4±1.6 |
| WSV | 85.9±1.0 | 71.4±1.3 | 92.6±0.7 | 89.0±1.5 | 81.6±2.4 | 77.5±1.7 |
| Tree-WSV | 86.3±1.5 | 71.2±1.9 | 92.4±1.0 | 88.7±1.9 | 82.0±2.9 | 76.4±2.4 |
| Tree-WSV-Wavelet | 86.2±1.1 | 71.5±1.4 (↑) | 92.0±0.8 | 89.3±1.4 (↑) | 82.0±2.3 | 78.1±1.9 (↑) |
| Alg. 1 | 96.7±0.3 | 74.1±0.5 | 97.3±0.2 | 94.0±0.4 | 90.1±0.4 | 86.7±0.5 |
| Alg. 2 | **97.3**±0.5 (↑) | **76.7**±0.7 (↑) | **97.6**±0.1 (↑) | **94.2**±0.2 (↑) | **94.0**±0.6 (↑) | **93.3**±0.7 (↑) |

Tab. 9 presents the performance of integrating the wavelet filtering variants ("-MST", "-HC", and "-Wavelet") into HGCNs for link prediction and node classification, compared to our full method that has iterative learning with wavelet filtering. The results show that the MST- and HC-based variants do not improve performance, while the improvement from the Wavelet-based variant is marginal. Applying wavelet filtering alone, without the iterative learning scheme, fails to achieve performance comparable to our full approach. In contrast, our full method incorporates the relation between samples and features through TWD, and uses this structure to inform the wavelet filtering process. It plays a key role in improving hierarchical representation and downstream task performance. To the best of our knowledge, this is the first work to integrate wavelet filtering with TWD and apply it within HGCNs.

Table 9: Performance comparison of HGCNs integrated with wavelet filtering variants ("-MST", "-HC", and "-Wavelet") versus our full method that combines iterative learning with wavelet filtering. Results are reported for link prediction and node classification tasks across hierarchical graph datasets. ROC AUC for LP, F1 score for the DISEASE dataset, and accuracy for the AIRPORT, PUBMED, and CORA datasets for NC tasks. Arrows (↑) denote cases with improvement. The highest performance is marked in bold, and the second-highest is underlined.

| | DISEASE | | AIRPORT | | PUBMED | | CORA | |
|---|---|---|---|---|---|---|---|---|
| | LP | NC | LP | NC | LP | NC | LP | NC |
| HGCNs [21] | 90.8±0.3 | 74.5±0.9 | 96.4±0.1 | 90.6±0.2 | 96.3±0.0 | 80.3±0.3 | 92.9±0.1 | 79.9±0.2 |
| HGCNs-MST | - | - | 92.3±0.2 | 88.4±0.4 | 91.7±0.2 | 77.9±0.3 | 90.4±0.3 | 77.6±0.6 |
| HGCNs-HC | - | - | 90.9±0.2 | 87.9±0.3 | 92.6±0.3 | 76.6±0.4 | 91.3±0.5 | 76.4±0.7 |
| HGCNs-Wavelet | 91.2±0.6 (↑) | 76.7±1.0 (↑) | 96.0±0.2 | 90.6±0.3 | 96.4±0.1 (↑) | 80.1±0.4 | 93.0±0.2 (↑) | 80.4±0.3 (↑) |
| H2H-GCN [45] | 97.0±0.3 | 88.6±1.7 | 96.4±0.1 | 89.3±0.5 | 96.9±0.0 | 79.9±0.5 | 95.0±0.0 | 82.8±0.4 |
| H2H-GCN-MST | - | - | 90.5±0.3 | 88.6±0.4 | 91.4±0.7 | 78.6±0.6 | 92.4±0.3 | 81.3±0.5 |
| H2H-GCN-HC | - | - | 92.1±0.2 | 88.9±0.4 | 92.6±0.4 | 79.8±0.2 | 92.5±0.5 | 81.7±0.4 |
| H2H-GCN-Wavelet | 97.2±0.5 (↑) | 87.9±1.8 | 96.4±0.4 | 90.1±0.4 (↑) | 95.2±0.1 | 80.2±0.5 (↑) | 94.7±0.2 | 82.4±0.3 |
| HGCN-Alg. 1 | 93.2±0.6 (↑) | 87.9±0.7 (↑) | 93.7±0.2 (↑) | 89.9±0.4 | 94.1±0.7 | 81.7±0.2 (↑) | 93.1±0.1 (↑) | 82.9±0.3 (↑) |
| HGCN-Alg. 2 | **98.4**±0.4 (↑) | **89.4**±0.3 (↑) | **97.2**±0.1 (↑) | **92.1**±0.3 (↑) | **97.2**±0.2 (↑) | **83.6**±0.4 (↑) | **96.9**±0.3 (↑) | **83.9**±0.2 (↑) |

Table 10: Comparison of classification accuracy across word-document and scRNA-seq datasets for three variants: Alg. 1, Alg. 1-Wavelet (post-hoc wavelet filtering after convergence), and Alg. 2 (wavelet filtering applied iteratively during learning). Arrows (↑) indicate improvements. We see that while post-hoc filtering offers minor improvements, iterative integration yields consistently better performance.

| | BBCSPORT | TWITTER | CLASSIC | AMAZON | ZEISEL | CBMC |
|---|---|---|---|---|---|---|
| Alg. 1 | 96.7±0.3 | 74.1±0.5 | 97.3±0.2 | 94.0±0.4 | 90.1±0.4 | 86.7±0.5 |
| Alg. 1-Wavelet | 96.8±0.3 (↑) | 74.1±0.4 | 97.3±0.3 | 94.0±0.3 | 91.1±0.4 (↑) | 88.2±0.6 (↑) |
| Alg. 2 | 97.3±0.5 | 76.7±0.7 | 97.6±0.1 | 94.2±0.2 | 94.0±0.6 | 93.3±0.7 |

## F.6 Ablation study: effect of post-hoc wavelet filtering vs. iterative integration

The main difference between Alg. 1 and Alg. 2 lies in the inclusion of a filtering step at each iteration of Alg. 2, designed to suppress noise and other undesired components of the data. This raises the question of whether applying wavelet filtering once Alg. 1 has converged could also be beneficial.

To assess whether applying wavelet filtering as a post-processing step after the convergence of Alg. 1 is beneficial, we evaluate a variant denoted as Alg. 1-Wavelet. Tab. 10 demonstrates that this post-hoc filtering yields marginal improvements over the unfiltered version (Alg. 1). However, the performance remains consistently lower than that of the fully integrated approach in Alg. 2, where wavelet filtering is applied iteratively during the learning process. This indicates that while post-processing offers slight gains, applying the filtering step within the iterative scheme is more effective for hierarchical representation learning.

## F.7 Sparse approximation analysis via haar coefficient across iterations

Fig. 8 shows the evolution of the $L_1$ norm of the Haar coefficients over iterations for all methods. Our methods exhibit a consistent and monotonic decrease in the $L_1$ norm as the iterations progress, reaching convergence. It reflects the sparse representations as a result of the joint hierarchical representation learning of both samples and features. Notably, sparsity is not explicitly enforced by our algorithm but emerges naturally from the learning process. In contrast, the competing methods are less stable over iterations. Some show slight reductions in the $L_1$ norm, but the decrease is not consistent and, in many cases, convergence is not reached. Even among those that converge, the final $L_1$ norm values are higher than those attained by our methods, indicating less effective sparse approximation.

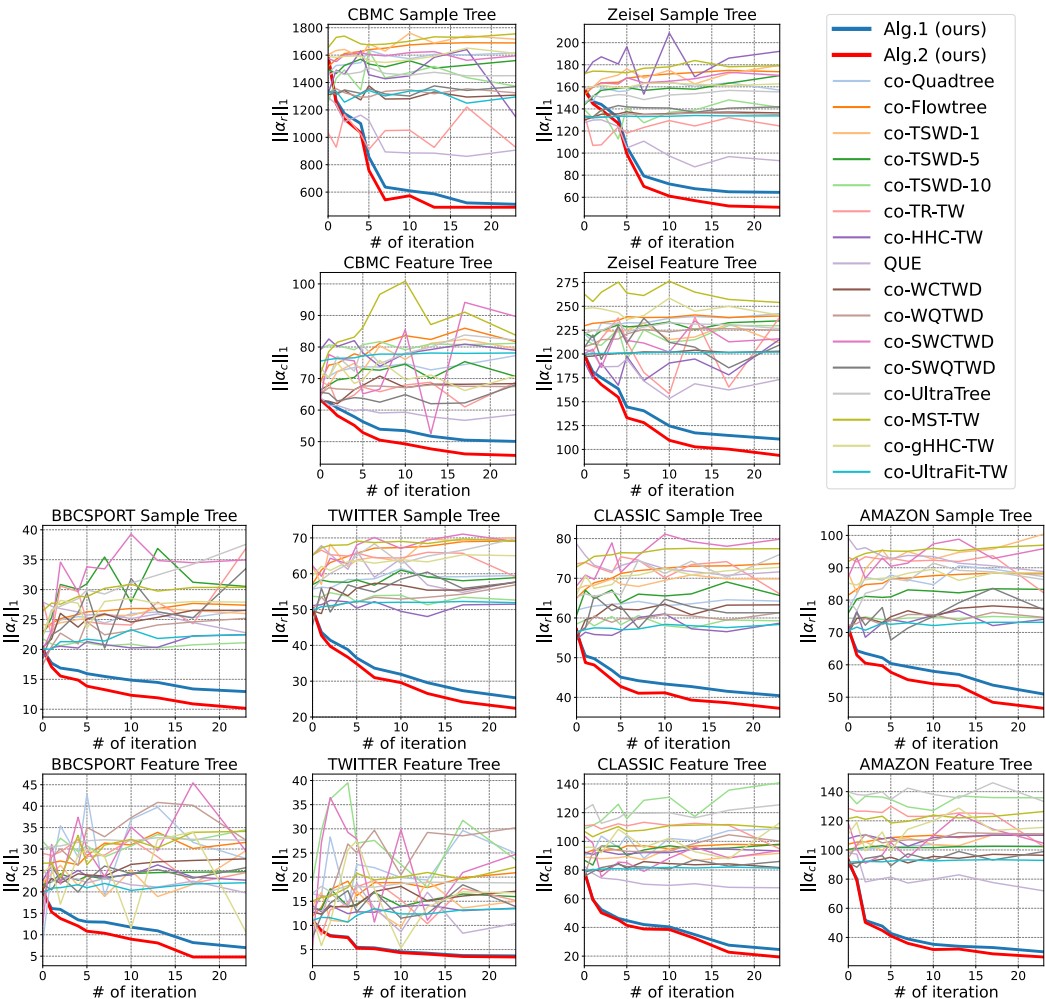

Figure 8: Evolution of the $L_1$ norm of the Haar coefficients over iterations for all methods. Our methods exhibit a consistent decrease in the $L_1$ norm, indicating improved sparse approximation through joint hierarchical learning. Competing methods show less stable behavior, with many failing to reach convergence within 25 iterations.

## F.8 Classification accuracy over iterations

We report the document and single-cell classification accuracy at each iteration for our method and the competing baselines. Fig. 9 presents the results on both scRNA-seq and word-document datasets. Our method shows consistent improvement in classification accuracy over iterations, outperforming the initial performance after the first half iteration of Alg. 1, which corresponds to using TWD without iterative refinement (see Tab. 6). This demonstrates the effectiveness of our iterative learning scheme in both data domains.

Moreover, our method consistently outperforms all competing baselines across iterations. While some baseline methods also benefit from iterative refinement, showing higher accuracy than their original TWD variants reported in Tab. 6, these gains are not uniformly observed. In addition, methods such as Co-Quadtree, Co-TSWD, Co-TR-TWD, and Co-HHC-TWD do not exhibit consistent improvements across all tasks, likely due to random sampling and initialization in their tree construction procedures.

Importantly, we observe that our method converges within a few iterations. While QUE and WSV also reach convergence, their final accuracy remains lower than ours. This highlights a key distinction: our method explicitly learns hierarchical representation learning by jointly refining the hierarchical

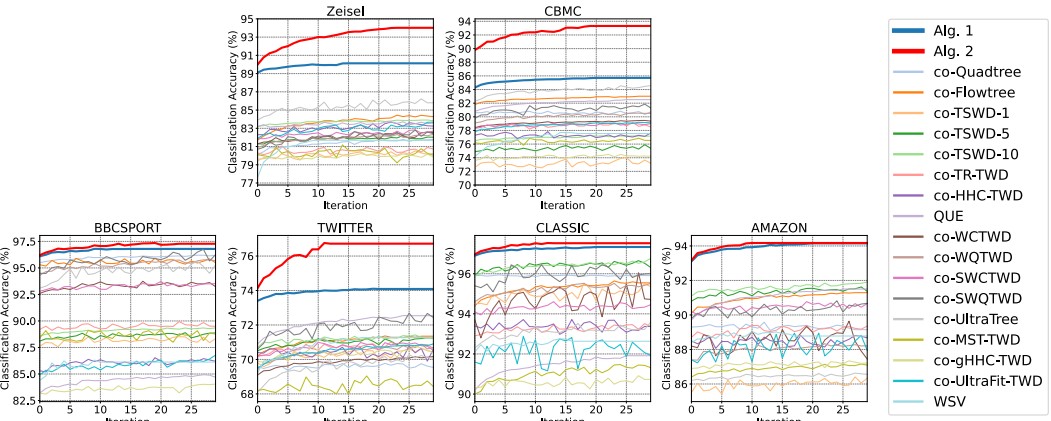

Figure 9: Classification accuracy over iterations for scRNA-seq and word-document datasets. Our method consistently improves accuracy across iterations and converges rapidly, outperforming both its non-iterative variant and all competing baselines. While some baselines benefit from iterative refinement, they show less consistent improvement and lower final accuracy.

structures of samples and features. It is further enhanced by the wavelet filtering, leading to improved classification performance.

## F.9    Runtime analysis

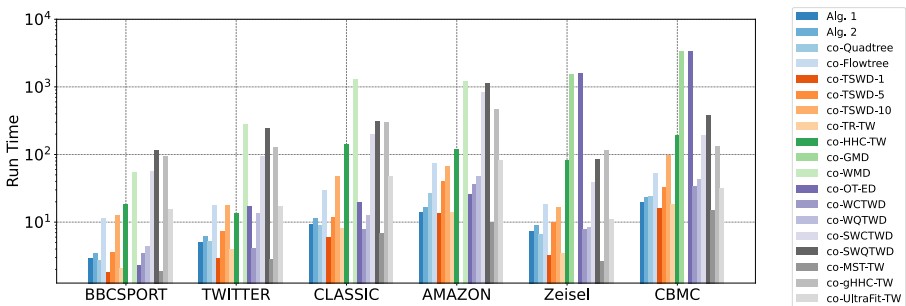

Figure 10: Runtime comparison of our methods and competing baselines.

Fig. 10 presents the comparison of runtime between our methods and the competing baselines. Co-WMD and Co-GMD correspond to the WSV variants initialized with WMD and GMD, respectively. While our methods show slightly higher runtime than Co-TSWD-1 and Co-TR-TWD, they achieve significantly better sparse approximation and classification accuracy across all baselines (see Tab. 1 and Tab. 2). This trade-off demonstrates that our approach remains computationally efficient while providing superior performance.

## F.10    Effect of Sinkhorn regularization in the iterative learning scheme

Our method adopts a snowflake-based regularization function [57], which promotes smoothness across both rows and columns of the data matrix. This regularizer increases monotonically over $[0, \infty)$ and penalizes small differences more strongly, encouraging local consistency. While effective, other regularization strategies may also be considered. One notable alternative is entropic regularization [119], which solves the OT problem using matrix scaling.

To explore this, we implement the entropic regularization using the Sinkhorn solver [155] within our iterative learning framework. We denote these variants as Alg. 1-Sinkhorn and Alg. 2-Sinkhorn. Tab. 11 reports the classification results on scRNA-seq and word-document datasets using the Sinkhorn regularization. In all cases, incorporating the Sinkhorn penalty leads to noticeably lower

Table 11: Classification accuracy on scRNA-seq and word-document datasets using Sinkhorn-regularized variants of our methods. "-Sinkhorn" denotes the use of entropic regularization in place of the snowflake penalty. Across all datasets, the Sinkhorn-based variants perform worse than the original methods, highlighting the effectiveness of the snowflake regularization.

|                | BBCSPORT | TWITTER | CLASSIC | AMAZON | ZEISEL | CBMC |
|----------------|----------|---------|---------|--------|--------|------|
| Alg. 1         | 96.7±0.3 | 74.1±0.5 | 97.3±0.2 | 94.0±0.4 | 90.1±0.4 | 86.7±0.5 |
| Alg. 1-Sinkhorn | 90.3±0.5 | 70.9±0.4 | 94.0±0.5 | 91.2±0.5 | 86.3±0.5 | 82.4±0.6 |
| Alg. 2         | 97.3±0.5 | 76.7±0.7 | 97.6±0.1 | 94.2±0.2 | 94.0±0.6 | 93.3±0.7 |
| Alg. 2-Sinkhorn | 94.2±0.4 | 73.8±0.5 | 95.4±0.3 | 91.5±0.1 | 89.7±0.5 | 85.9±0.6 |

Table 12: Clustering performance on gene expression datasets (Breast Cancer and Leukemia) evaluated using Clustering Accuracy (CA), Normalized Mutual Information (NMI), and Adjusted Rand Index (ARI). Our methods (Alg. 1 and Alg. 2) outperform all baselines. Alg. 2, which incorporates wavelet filtering, achieves the highest overall performance. Results for BCOT, CCOT, CCOT-GW, and COOT are reported from the work of Fettal et al. [156]. OOM indicates methods that ran out of memory.

| Dataset | Breast Cancer | | | Leukemia | | |
|---------|-----|-----|-----|-----|-----|-----|
| Evaluation Metric | CA | NMI | ARI | CA | NMI | ARI |
| QUE | 71.9±1.2 | 45.3±0.7 | 39.8±0.4 | 72.3±0.5 | 56.6±0.6 | 42.9±0.4 |
| WSV | 72.4±3.6 | 33.8±2.2 | 39.0±0.8 | 64.3±4.1 | 61.7±2.3 | 44.8±5.1 |
| Tree-WSV | 73.1±1.8 | 36.7±1.7 | 41.5±0.9 | 66.2±3.0 | 60.4±1.8 | 42.6±3.1 |
| BCOT* | 76.9±0.0 | 37.2±0.0 | 26.7±0.0 | 71.2±5.4 | 59.6±6.9 | 39.9±6.3 |
| BCOT$_\lambda$* | 84.6±0.0 | 48.3±0.0 | 46.0±0.0 | 80.9±3.8 | 70.9±4.1 | 55.3±3.3 |
| CCOT* | OOM | OOM | OOM | 40.6±0.0 | 0.0±0.0 | 0.0±0.0 |
| CCOT-GW* | OOM | OOM | OOM | OOM | OOM | OOM |
| COOT* | 63.1±5.2 | 5.4±8.7 | -1.2±2.9 | 36.2±2.7 | 14.0±3.6 | 5.4±3.2 |
| COOT$_\lambda$* | 61.5±0.0 | 5.4±0.0 | 2.2±0.0 | 32.5±3.3 | 8.7±2.7 | -0.5±2.1 |
| Alg. 1 | 85.6±0.7 | 49.2±0.5 | 50.4±0.9 | 75.8±0.9 | 64.2±1.0 | 54.2±1.1 |
| Alg. 2 | 87.2±0.4 | 51.3±0.3 | 57.6±0.7 | 81.2±1.1 | 75.4±0.8 | 59.7±1.2 |

performance compared to the original versions of our algorithms. The classification accuracy degrades, indicating that the entropic regularization may overly smooth or distort the learned hierarchical structures. This suggests that the snowflake regularization is more suitable in our setting, where the goal is to represent fine-grained hierarchical relationships across both samples and features.

### F.11 Co-Clustering performance on gene expression data

We further evaluate the effectiveness of our methods in clustering tasks using two gene expression datasets [157]. The first is a breast cancer (BC) dataset containing 42,945 gene expression values across 26 samples divided into two classes. The second is a leukemia (LEU) dataset with 22,283 genes measured across 64 patients divided into five classes. We apply Alg. 1 and Alg. 2 to the data and perform $k$-means clustering [158] on the resulting sample TWD matrices to obtain clustering assignments. We evaluate the OT-based co-clustering baselines including: QUE [52], WSV [59], Tree-WSV [60], biclustering with optimal transport (BCOT and BCOT$_\lambda$) [156], Co-clustering through OT (CCOT and CCOT-GW) [159], and Co-optimal transport (COOT and COOT$_\lambda$) [160]. We follow the evaluation setup [156], reporting Clustering Accuracy (CA), Normalized Mutual Information (NMI) [161], and Adjusted Rand Index (ARI) [162]. The results of BCOT, CCOT, CCOT-GW, and COOT are reported from the work of Fettal et al. [156].

Tab. 12 shows that our methods outperform all competing baselines. Alg. 1 already achieves strong performance, while Alg. 2 further improves upon it with consistently higher CA, NMI, and ARI scores. In particular, the improvements are pronounced on the Leukemia dataset, where our method significantly exceeds the baselines. Some methods, such as CCOT and CCOT-GW, run out of

**Algorithm 4** Incorporating Fixed Hierarchical Distance for One Mode in Iterative Learning

---

**Input:** Data matrix $\mathbf{X} \in \mathbb{R}_+^{n \times m}$, $\mathbf{M}_c \in \mathbb{R}^{m \times m}$ and $\mathbf{M}_r \in \mathbb{R}^{n \times n}$, $\gamma_r$ and $\gamma_c$, thresholds $\vartheta_c$ and $\vartheta_r$
**Output:** Trees $\mathcal{T}(\widetilde{\mathbf{W}}_r^{(l)})$ and $\mathcal{T}(\widetilde{\mathbf{W}}_c^{(l)})$, and TWDs $\widetilde{\mathbf{W}}_r^{(l)}$ and $\widetilde{\mathbf{W}}_c^{(l)}$

$l \leftarrow 0, \mathbf{X}^{(0)} \leftarrow \mathbf{X}, \mathbf{Z}^{(0)} \leftarrow \mathbf{X}^\top, \widetilde{\mathbf{W}}_c^{(0)} \leftarrow \mathbf{M}_c$            ▷ Initialization
**repeat**
     $\mathbf{X}^{(l+1)} \leftarrow \Psi(\mathbf{X}^{(l)}; \mathcal{T}(\widetilde{\mathbf{W}}_c^{(l)}))$          ▷ Tree haar wavelet filtering with updated tree
     $\mathbf{Z}^{(l+1)} \leftarrow \Psi(\mathbf{Z}^{(l)}; \mathcal{T}(\mathbf{M}_r))$          ▷ Tree haar wavelet filtering with fixed tree
     $\widetilde{\mathbf{X}}_r^{(l+1)} \leftarrow \left\{ \mathbf{r}_i^{(l+1)} = \left( \mathbf{X}_{i,:}^{(l+1)} \right)^\top / \|\mathbf{X}_{i,:}^{(l+1)}\|_1 \right\}$
     $\widetilde{\mathbf{X}}_c^{(l+1)} \leftarrow \left\{ \mathbf{c}_j^{(l+1)} = \left( \mathbf{Z}_{j,:}^{(l+1)} \right)^\top / \|\mathbf{Z}_{j,:}^{(l+1)}\|_1 \right\}$
     $\widetilde{\mathbf{W}}_r^{(l+1)} \leftarrow \Phi(\widetilde{\mathbf{X}}_r^{(l+1)}; \mathcal{T}(\widetilde{\mathbf{W}}_c^{(l)}))$          ▷ Iterative update with updated tree
     $\widetilde{\mathbf{W}}_c^{(l+1)} \leftarrow \Phi(\widetilde{\mathbf{X}}_c^{(l+1)}; \mathcal{T}(\mathbf{M}_r))$          ▷ Iterative update with fixed tree
     $l \leftarrow l + 1$
**until** convergence

---

Table 13: Link prediction and node classification performance on hierarchical graph datasets using different variants of the proposed HGCN framework. HGCN-Alg. 1 uses only the reference update without filtering. HGCN-Alg. 2 updates both filtering and referencing functions adaptively. HGCN-Alg. 4 applies filtering with a fixed sample distance matrix $\mathbf{M}_r = d_H$. Results show that while fixing $\mathbf{M}_r$ improves over the reference-only variant, the superior performance is achieved when both filtering and referencing steps are adaptively updated throughout the iterations.

|  | DISEASE | | AIRPORT | | PUBMED | | CORA | |
| --- | --- | --- | --- | --- | --- | --- | --- | --- |
|  | LP | NC | LP | NC | LP | NC | LP | NC |
| HGCN-Alg. 1 | 93.2±0.6 | 87.9±0.7 | 93.7±0.2 | 89.9±0.4 | 94.1±0.7 | 81.7±0.2 | 93.1±0.1 | 82.9±0.3 |
| HGCN-Alg. 4 | 95.1±0.5 | 88.2±0.9 | 95.9±0.1 | 90.7±0.3 | 95.7±0.4 | 82.0±0.3 | 95.3±0.1 | 83.2±0.4 |
| HGCN-Alg. 2 | 98.4±0.4 | 89.4±0.3 | 97.2±0.1 | 92.1±0.3 | 97.2±0.2 | 83.6±0.4 | 96.9±0.3 | 83.9±0.2 |

memory (OOM), and others like COOT and COOT$_\lambda$ produce poor clustering results. These findings demonstrate the robustness and effectiveness of our method, especially when enhanced by wavelet filtering at each iteration.

### F.12 Incorporating fixed hierarchical distances in the iterative learning scheme for HGCNs

In Sec. 5, we demonstrated how prior knowledge of a hierarchical structure on one of the modes can be incorporated by initializing the sample distance matrix $\mathbf{M}_r$ with the shortest-path distances $d_H$ induced by a given hierarchical graph $H$. This initialization was evaluated in the context of hierarchical graph data using the HGCN framework. Beyond initialization, another way to incorporate $d_H$ into our iterative learning scheme is to fix the sample pairwise distance matrix as $\mathbf{M}_r = d_H$ during the filtering step $\Psi(\cdot)$ and the joint reference computation $\Phi(\cdot)$.

However, it is important to distinguish between the two variants of our learning scheme. The iterative algorithm in Alg. 1 only involves the reference computation $\Phi(\cdot)$. Therefore, if $\mathbf{M}_r$ is fixed within $\Phi(\cdot)$, no update occurs after the first iteration, and the procedure halts. In contrast, Alg. 2 includes an additional filtering step $\Psi(\cdot)$ that continues to evolve even when the used distance in $\Phi(\cdot)$ remains fixed. As a result, fixing $\mathbf{M}_r$ is only meaningful within Alg. 2, where the iterative updates can still proceed. We summarize this approach in Alg. 4 as an alternative variant that allows the incorporation of external hierarchical information. This approach is also evaluated on hierarchical graph data using the HGCNs framework.

Tab. 13 reports the link prediction and node classification results when the sample distance matrix $\mathbf{M}_r$ is fixed in the iterative learning scheme (denoted as HGCN-Alg. 4). Compared to HGCN-Alg. 1, fixing $\mathbf{M}_r$ and applying filtering improves performance. This shows the benefit of incorporating the

Table 14: Link prediction and node classification results for GCN, GRAPHSAGE, and HGCNs, with and without initialization using our learned hierarchical representations ("-Alg. 2").

| | DISEASE | | AIRPORT | | PUBMED | | CORA | |
|---|---|---|---|---|---|---|---|---|
| | LP | NC | LP | NC | LP | NC | LP | NC |
| GCN | 64.7±0.5 | 69.7±0.4 | 89.3±0.4 | 81.4±0.6 | 91.1±0.5 | 78.1±0.2 | 90.4±0.2 | 81.3±0.3 |
| GCN-Alg. 2 | 68.1±0.4 (↑) | 71.2±0.5 (↑) | 90.6±0.2 (↑) | 83.0±0.9 (↑) | 90.5±0.3 | 78.7±0.1 (↑) | 91.2±0.4 (↑) (↑) | 81.9±0.2 |
| GRAPHSAGE | 65.9±0.3 | 69.1±0.6 | 90.4±0.5 | 82.1±0.5 | 86.2±1.0 | 77.4±2.2 | 85.5±0.6 | 77.9±2.4 |
| GRAPHSAGE-Alg. 2 | 67.8±0.1 (↑) | 70.5±0.3 (↑) | 90.2±0.4 | 83.7±0.5 (↑) | 90.3±1.1 (↑) | 78.8±1.7 (↑) | 89.6±0.8 (↑) | 80.5±1.6 (↑) |
| HGCNs | 90.8±0.3 | 74.5±0.9 | 96.4±0.1 | 90.6±0.2 | 96.3±0.0 | 80.3±0.3 | 92.9±0.1 | 79.9±0.2 |
| H2H-GCN | 97.0±0.3 | 88.6±1.7 | 96.4±0.1 | 89.3±0.5 | 96.9±0.0 | 79.9±0.5 | 95.0±0.0 | 82.8±0.4 |
| HGCN-Alg. 2 | **98.4**±0.4 (↑) | **89.4**±0.3 (↑) | **97.2**±0.1 (↑) | **92.1**±0.3 (↑) | **97.2**±0.2 (↑) | **83.6**±0.4 (↑) | **96.9**±0.3 (↑) | **83.9**±0.2 (↑) |

Table 15: Comparison of representation quality for Independent MST, Co-MST-TWD, and our full method (Alg. 2), measured by the $L_1$ norm of Haar coefficients (lower is better). Results are reported as samples / features for each dataset. Lower values indicate that the learned hierarchical representations more effectively capture the hierarchical information across the two data modes.

| | BBCSPORT | TWITTER | CLASSIC | AMAZON | ZEISEL | CBMC |
|---|---|---|---|---|---|---|
| Independent MST | 37.9 / 42.8 | 82.4 / 31.1 | 83.6 / 131.4 | 109.6 / 144.1 | 194.3 / 281.5 | 1928.7 / 97.6 |
| Co-MST-TWD | 30.4 / 34.1 | 69.1 / 22.1 | 77.5 / 109.3 | 97.0 / 126.4 | 179.1 / 254.0 | 1755.3 / 83.7 |
| Alg. 2 | **10.1 / 4.8** | **22.4 / 3.4** | **37.2 / 19.4** | **46.6 / 26.6** | **50.9 / 93.7** | **489.4 / 45.6** |

hierarchical structure during filtering. However, HGCN-Alg. 2, which adaptively updates both filtering and referencing functions via the evolving $\widehat{\mathbf{W}}_r^{(l)}$ along the iteration, consistently outperforms the fixed-distance variant. This highlights the importance of mutual adaptation between the sample and feature structures during the iterative learning process, where the joint updates allow the hierarchical relations to be refined jointly across iterations.

## F.13 Evaluating the preprocessing step with additional backbones

To evaluate the effectiveness of our method as a preprocessing step across different neural network backbones, we extend our experiments to include GCN and GraphSAGE, in addition to the hyperbolic models considered in Tab. 3. Tab. 14 reports link prediction and node classification results, where models initialized with our learned hierarchical representations (denoted "-Alg. 2") consistently outperform their baselines. Improvements are indicated with an upward arrow (↑). These findings demonstrate the broad applicability of our approach across diverse neural architectures. Among all models, HGCN-Alg. 2 achieves the highest overall performance, which we attribute to the advantages of hyperbolic representations in modeling hierarchical structures, distinguishing HGCNs from their Euclidean counterparts.

## F.14 Discussion and comparison with other tree distance metrics

In our framework, TWD plays a central role due to its ability to compare probability distributions supported on hierarchical structures. This is essential for our alternating, joint-learning process, where the hierarchical structure from one mode (e.g., samples) is used to inform and refine the hierarchy in the other mode (e.g., features), and vice versa. That is, the tree structure provides the foundation for expressing hierarchy, while the Wasserstein metric provides the mechanism for distributional comparison and transfer from one mode to the other. This cross-mode interaction is central to our method and relies on the Wasserstein nature of TWD.

TWD differs from other tree distance metrics such as minimum spanning tree (MST) [163], neighbor joining, or low-stretch trees, which typically operate on tree structures or discrete node sets but do not directly support distribution-based comparison needed for cross-mode refinement. To clarify this difference, we consider two alternative setups using MST: (1) using independently constructed MSTs for samples and features without cross-mode interaction, and (2) using an MST as the tree structure in the TWD framework for our iterative learning process (Co-MST-TWD). We compare these alternatives to our full method (Alg. 2) by measuring representation quality using the $L_1$

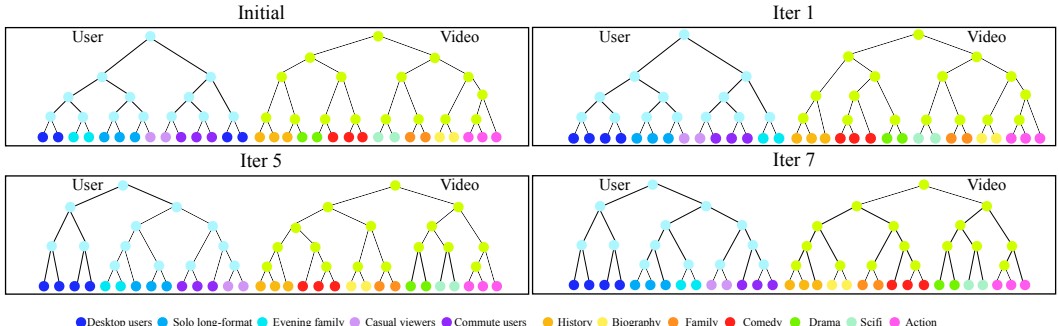

Figure 11: Stepwise hierarchical refinement on the toy user–video recommendation in App. F.1.

norm of Haar coefficients. Results are shown in Tab. 15 (format: samples / features). We see that both alternatives yield substantially higher $L_1$ norms. These results suggest that the learned joint hierarchical representation from our iterative process better reflects the hierarchical information of the two data modes. We attribute this advantage to the specific design of our proposed method, where the TWD supports iterative cross-mode refinement, which is not feasible with independent tree metrics.

Replacing TWD with other graph-based distances is conceptually possible, but there are important considerations:

- Most graph-based distances are designed to compare node positions or global connectivity in general graphs, not to measure how distributions are supported on a tree or reveal explicit nested groupings.

- Gromov-Wasserstein distance [164] could, in principle, be integrated into an iterative scheme as it compares distributions across different metric spaces or graphs. However, such approaches are much more computationally intensive.

- Another possible direction is to construct a graph from high-dimensional data and transform this general graph into unrolling trees [165]. Further investigation would be needed to determine if and how such transformations can be integrated within our iterative learning process.

### F.15 Stepwise hierarchical visualizations in recommendation toy problem

Fig. 11 presents the stepwise visualizations based on the toy video recommendation example in App. F.1 for hierarchical refinement. The iterative refinement process starts with an initial tree structure, which is unstructured since the preliminary grouping is based on initial data relationships ($\mathbf{M}_c$ or $\mathbf{M}_r$). At this point, major clusters are not well formed, and samples or features from different categories are mixed. After the first iteration, the algorithm uses the current tree from one mode (e.g., users) to compute a TWD for the other mode (e.g., videos). This leads to a newly induced tree that starts to separate the major categories, and major branches start to reflect groupings, but subcategory assignments are not fully sorted. As the algorithm proceeds through subsequent iterations, it alternately refines the trees for users and videos. The updated tree for one mode informs a more accurate computation of TWD in the other mode, enabling finer divisions within the existing branches. Subcategories like action, drama, and sci-fi (for fiction videos) become more distinct and better grouped. After several iterations, the subcategories become more cohesive and consistently organized, and the tree structures stabilize. It illustrates how the trees evolve with each iteration and points out key groupings formed in the learned trees.

## G   Additional remarks

We provide additional remarks on our methods, including the motivation for using wavelet filtering, and the comparison with WSV and Tree-WSV.

## G.1 Motivation for tree-based wavelet filtering

To refine the hierarchical representations learned through our iterative learning scheme, we incorporate a filtering step $\Psi(\cdot)$ at each iteration. This filtering is based on Haar wavelets [38–41] induced by the learned trees. Each sample $\mathbf{X}_{i,:}^\top \in \mathbb{R}^m$ can be viewed as a signal defined over the feature tree $\mathcal{T}(\mathbf{M}_c)$, whose $m$ leaves correspond to features [42]. Given a feature tree $\mathcal{T}(\mathbf{M}_c)$, we construct an orthonormal Haar basis $\mathbf{B}_c \in \mathbb{R}^{m \times m}$, as described in Sec. 3 and App. A. Each sample is expanded in this basis, yielding coefficients $\boldsymbol{\alpha}_i = (\mathbf{X}_{i,:}\mathbf{B}_c)^\top \in \mathbb{R}^m$, where $\boldsymbol{\alpha}_i(j) = \langle \mathbf{X}_{i,:}^\top, \boldsymbol{\beta}_j \rangle$ and $\boldsymbol{\beta}_j$ is the $j$-th wavelet in $\mathbf{B}_c$. To define a wavelet filter, we select a subset of the Haar basis vectors as follows. For each coefficient index $j$, we compute the aggregate $L_1$ norm $\sum_{i=1}^n |\boldsymbol{\alpha}_i(j)|$ across samples and sort the indices in descending order. We then select the subset $\Omega$ such that the cumulative contribution $\eta_\Omega = \sum_{q \in \Omega} \sum_{i=1}^n |\boldsymbol{\alpha}_i(q)|$ exceeds a threshold $\vartheta_c > 0$. A similar filtering procedure can be applied to features using the Haar wavelet construct by the sample tree.

It is important to note that this $L_1$-based filtering retains the components that contribute most to the hierarchical decomposition, as represented by the tree [69]. We postulate that the residual, i.e., the remaining coefficients outside the set $\Omega$, is the variation that is not aligned with the tree structure. These components are treated as noise or nuisance terms that are irrelevant for the hierarchical representation learning. By discarding this residual, the filter emphasizes signal content that is structurally consistent with the hierarchical structure, leading to representations that are more stable and meaningful across iterations [166, 167]. Our filtering step adapts to the learned trees at each iteration. That is, the Haar wavelets are constructed from the trees $\mathcal{T}(\widehat{\mathbf{W}}_r^{(l)})$ and $\mathcal{T}(\widehat{\mathbf{W}}_c^{(l)})$ via TWD, making the filtering aligned with the evolving hierarchical representations. Empirically, we show that applying the wavelet filtering in the iterative learning scheme improves performance in various tasks on hierarchical data, including link prediction and node classification on hierarchical graph data, as well as sparse approximation and metric learning on scRNA-seq and document datasets.

## G.2 Advantages of Haar wavelet filtering over Laplacian-based filtering for hierarchical data

In Sec. 4.2, we introduced a filtering step based on the Haar wavelet, where the wavelet basis is induced by the tree structure [43] learned during our iterative scheme. This filtering step is designed to reflect the intrinsic hierarchical structure of the data, enhancing meaningful hierarchical representations while suppressing noise and other nuisance components. An alternative filtering approach could involve using the Laplacian matrix derived from the tree graph. This would allow spectral filtering based on the eigendecomposition of the tree [42, 168], offering another way to apply filtering that incorporates the hierarchical information from the tree [169].

Consider a feature tree $\mathcal{T}(\mathbf{M}_c)$ with the affinity matrix $\mathbf{A}_c \in \mathbb{R}^{m \times m}$. We can construct the Laplacian matrix for the feature tree by $\mathbf{L}_c = \mathbf{D}_c - \mathbf{A}_c$, where $\mathbf{D}_c$ is a diagonal degree matrix with $\mathbf{D}_c(j,j) = \sum_i \mathbf{A}(i,j)$. The eigenvectors of the Laplacian matrix are then treated as an orthonormal basis, often referred to as the graph Fourier basis [170–172]. Subsequently, each sample can be expanded in this eigenbasis, and we denote $\widetilde{\boldsymbol{\alpha}}_i = (\mathbf{X}_{i,:}\mathbf{U}_c)^\top \in \mathbb{R}^m$ as the vector of the expansion coefficients of the $i$-th sample, where $\mathbf{U}_c$ is the matrix consisting of eigenvectors of $\mathbf{L}_c$. This expansion allows the interpretation of the Laplacian expansion coefficient $\widetilde{\boldsymbol{\alpha}}_i(j)$ as the contribution of the $j$-th frequency component to the signal. By sorting these coefficients according to the corresponding Laplacian eigenvalues, we can construct a low-pass filter (containing the first few expansions) that suppresses high-frequency components and noise while promoting signal smoothness [173]. A similar decomposition can also be applied to the features using the Laplacian matrix of the sample tree.

We evaluate this alternative filtering technique based on the Laplacian matrix of the tree graph. Specifically, we modify the iterative scheme in Alg. 2 by replacing the Haar wavelet filtering step with the low-pass Laplacian-based filtering, and refer to this variant as "Alg. 2-Laplacian." We test this variant on both document and single-cell classification tasks, as well as on hierarchical graph data for link prediction and node classification. Tab. 16 reports the classification accuracy for the document and single-cell tasks. We see that compared to the wavelet-based approach, the Laplacian variant consistently underperforms, showing a noticeable drop. This suggests that while Laplacian filters offer a spectral interpretation, they are less effective at representing hierarchical information in our setting. Similarly, Tab. 17 presents link prediction and node classification performance for integrating the Laplacian filtering into our iterative learning scheme and applied to HGCNs. Here too, we observe that HGCN-Alg. 2-Laplacian performs worse than HGCN-Alg. 2. These results show the

Table 16: Classification accuracy on document and single-cell datasets. We compare the performance of the iterative method without filtering step (Alg. 1), our full method with Haar wavelet filtering (Alg. 2), and the Laplacian filtering variant (Alg. 2-Laplacian). Haar wavelet filtering consistently achieves the better performance.

|                 | BBCSPORT | TWITTER  | CLASSIC  | AMAZON   | ZEISEL   | CBMC     |
|-----------------|----------|----------|----------|----------|----------|----------|
| Alg. 1          | 96.7±0.3 | 74.1±0.5 | 97.3±0.2 | 94.0±0.4 | 90.1±0.4 | 86.7±0.5 |
| Alg. 2-Laplacian| 95.2±0.7 | 72.6±0.9 | 94.4±1.0 | 91.2±0.4 | 83.7±0.9 | 81.6±1.2 |
| Alg. 2          | 97.3±0.5 | 76.7±0.7 | 97.6±0.1 | 94.2±0.2 | 94.0±0.6 | 93.3±0.7 |

Table 17: Performance of different filtering variants in our iterative learning scheme applied to hierarchical graph data within the HGCN framework for link prediction and node classification. We compare HGCNs using the hierarchical representations from Alg. 1, the variant with Laplacian filtering (HGCN-Alg. 2-Laplacian), and the full method with Haar wavelet filtering (HGCN-Alg. 2). The Haar wavelet variant consistently achieves superior performance.

|                        | DISEASE | | AIRPORT | | PUBMED | | CORA | |
|------------------------|---------|---------|---------|---------|---------|---------|---------|---------|
|                        | LP      | NC      | LP      | NC      | LP      | NC      | LP      | NC      |
| HGCN-Alg. 1            | 93.2±0.6| 87.9±0.7| 93.7±0.2| 89.9±0.4| 94.1±0.7| 81.7±0.2| 93.1±0.1| 82.9±0.3|
| HGCN-Alg. 2-Laplacian  | 89.4±0.3| 79.2±1.1| 93.1±0.4| 88.4±0.7| 93.5±0.4| 80.9±0.2| 92.7±0.2| 81.9±0.3|
| HGCN-Alg. 2            | 98.4±0.4| 89.4±0.3| 97.2±0.1| 92.1±0.3| 97.2±0.2| 83.6±0.4| 96.9±0.3| 83.9±0.2|

benefit of Haar wavelet filtering, which more directly reflects the hierarchical structure and is more effective for learning meaningful representations.

We attribute this performance gap to the differences between the two filtering methods. Laplacian-based filtering is not designed to represent hierarchical structure explicitly [42, 72]. Its eigenvectors are global and do not distinguish between different levels of abstraction in the data. In addition, from a computational perspective, computing the eigendecomposition of the graph Laplacian can be expensive: full decomposition scales cubically with the number of nodes, while partial decomposition (e.g., top $k$ eigenvectors) still incurs significant cost, often quadratic or worse in $n$ (depending on the method and graph sparsity). Furthermore, it has been shown that the expansion coefficients in the Haar wavelet basis decay at an exponential rate (Prop. A.1), in contrast to the typically polynomial rate of decay observed with Laplacian eigenfunction coefficients [43]. This rapid decay reflects the ability of Haar wavelets to provide compact representations of piecewise-smooth signals, using the learned tree structure to concentrate energy in fewer coefficients. Finally, Haar wavelets on trees yield interpretable components [38]: each wavelet coefficient corresponds to a specific scale and position within the tree, often representing differences between sibling nodes or between a node and its parent. This locality and hierarchical structure are absent in Laplacian eigenbases. Therefore, for data where a hierarchical relationship between samples or features is important, Haar wavelets provide a more structurally aligned approach compared to graph signal filtering based on the Laplacian.

We note that more advanced filtering strategies, such as those learned through neural networks [96], could also be explored. We leave this extension for future work.

### G.3   Comparison with Tree-WSV and the role of hierarchical representation learning

While preparing this manuscript, a new method called Tree-WSV [60] was introduced. It uses Tree-Wasserstein Distance (TWD) to place samples and features as leaves on trees and was introduced to reduce the complexity that follows ideas from existing TWD research [35, 34]. Specifically, this approach speeds up WSV [59] for unsupervised ground metric learning.

The main idea of WSV is to simultaneously compute the Wasserstein distance among samples (resp. features) by using the Wasserstein distance among features (resp. samples) as the ground metric. This design conceptually aligns with co-manifold learning methods, which use relationships among samples to guide the relationships among features [52–55, 7, 56, 57, 63]. Tree-WSV extends WSV by using TWD as a low-rank approximation of the Wasserstein distance. It does this by computing TWD among samples (resp. features) while using TWD among features (resp. samples) as the

ground metric. Tree-WSV reduces the complexity of WSV from $O(n^2m^2(n\log n + m\log m))$ to $O(n^3 + m^3 + mn)$, making it more efficient in practice. In this context, the constructed trees primarily serve as computational tools for approximating Wasserstein distances.

In contrast, our work takes a geometric approach where the trees induced by TWD are used to represent hierarchical structures among samples and features. Based on this geometric perspective, we propose Alg. 1 and Alg. 2, which jointly learn hierarchical representations for both samples and features. Rather than using TWD merely as a computationally efficient approximation, we treat it as an input distance to construct diffusion operators that encode manifold information [68]. These operators are then used to construct trees based on hyperbolic embeddings and diffusion densities [37], which recursively guide the computation of TWD for the other data mode.

This geometric view relates more closely to co-manifold learning in spirit, which use the geometry of one data mode to inform the geometry of the other, and was not explored in the WSV or Tree-WSV frameworks. Our primary goal is not unsupervised ground metric learning (though this arises naturally) but rather joint samples and features hierarchical representations. We also use multiple resolutions to better represent the hierarchical structure of the data, similar to ideas found in tree-based Earth mover's distance [52, 57].

Empirically, our methods outperform WSV and Tree-WSV across tasks involving hierarchical data, including sparse approximation, document classification, and single-cell analysis, as detailed in Sec. 6. These improvements highlight the benefit of explicitly modeling and learning hierarchical representations rather than using trees solely for efficient computation. Furthermore, we demonstrate that the learned hierarchical representations can serve as effective inputs for downstream applications, such as hyperbolic graph neural networks (HGCNs), where incorporating our wavelet-based hierarchical representations leads to improved link prediction and node classification performance. This integration further illustrates the utility and effectiveness of our approach beyond unsupervised ground metric learning.

## G.4   On the interpretability of TWDs

Note that one strength of TWD lies in its transparent and decomposable structure. Given two distributions (e.g., feature or sample vectors), the TWD quantifies the minimal "mass transport" needed to align them over the tree, with transport cost reflecting branch length. This formulation enables users to understand where and at what scale differences occur. For instance, if most of the transport cost is concentrated at high-level branches, the two distributions differ in broad terms; if the cost is concentrated near the leaves, they are largely similar but differ in fine details. As a result, TWD not only quantifies similarity but also supports qualitative interpretation: users can trace which parts of the tree contribute most to the distance between any two data points. In classification tasks, this interpretability reveals which regions of the hierarchy drive class separation and can help identify key transitions in the data.

