# OpenReview forum: "Joint Hierarchical Representation Learning of Samples and Features via Informed Tree-Wasserstein Distance"
_NeurIPS.cc/2025/Conference — NeurIPS 2025 spotlight_

### Official Review · Reviewer_GK8p · 2025-06-02

**Clarity:** 3
**Significance:** 3
**Originality:** 3
**Rating:** 5
**Confidence:** 4

**Summary:**

The paper proposes a novel approach to simultaneously modeling hierarchies in both samples and features. The joint learning process facilitates the convergence to the shared optimal point, which is theoretically proved and discussed in the paper, leading to an accurate and efficient output. The alternating algorithm, which iteratively constructs trees for one mode and leverages TWD to refine the other applies the joint learning criteria to the field of representation learning successfully.. This bidirectional refinement captures the interplay between data modes effectively, a critical insight for high-dimensional datasets like word-document matrices and single-cell RNA-sequencing data.

**Questions:**

1. The writing around the citations should be improved. The text should remain correctly readable in case of the removal of the citations (e.g., on line 94, , the method recently proposed in [34] computes multiscale diffusion densities...).

2. The algorithms should be presented better. The current representation is concise, but the level of description is very limited. The algorithms can also benefit from an informative title, instead of just a number.

3. "We provide detailed descriptions of our method’s implementation within the paper and report the hyperparameters in App. E. Our source code will be made publicly available on GitHub upon publication.", it would be better if the code and the instructions to reproduce the results were available for the review. It could be easily provided by using an anonymous GitHub link using services like https://anonymous.4open.science/ .

4. The trees and TWDs produce hierarchical structures, but how interpretable are they to end users? This investigation could further improve the paper, especially with regards to the application in classificaiton.

5. It would be interesting to see the visualization of the different levels of the trees to visually investigate how they are extracted and refined in each step or level.

**Ethical Concerns:**

["NO or VERY MINOR ethics concerns only"]

**Final Justification:**

The author rebuttals and other reviews have not changed my perception that this an very good to excellent contribution to the field. I have, however, raised my confidence.

**Limitations:**

The main limitations of the paper is the lack of the implementation and code to reproduce the results. The investigation of the structure in different hierarchies and the explaianbility of the hierarchical structure is also expected.

**Paper Formatting Concerns:**

None.

**Quality:**

4

**Strengths And Weaknesses:**

The paper’s primary strength is the novel approach to simultaneously modeling hierarchies in both samples and features.

he proposed method seems very sound. The theoretical analysis of convergence is a standout feature. The authors provide a robust proof that the iterative process stabilizes, providing confidence to the method’s reliability. There are many theoretical and empirical analyses presented in the Appendix which shed light further onto the different properties of the proposed method.

Empirically, the experimental results show that the proposed method can be successfully applied to different fields such as document and single-cell classification, link predication and node classification.

On the other hand, no implementation or code in order to reproduce and validate the results are included, which is expected from a paper of this type.

---

> ### Author Rebuttal · Authors · 2025-07-30
>
> We thank the reviewer for their thorough and valuable review, and for recognising the novelty of our work for simultaneously modelling hierarchies in both samples and features. In terms of the weaknesses and questions:
>
> >**Improving citation-dependent writing**
>
> We thank the reviewer for highlighting the need to improve the readability of the manuscript around citations.  We will revise the manuscript to make sure that the text remains clear and grammatically correct even if the citation markers are removed. This revision will be applied consistently throughout the manuscript to improve readability and maintain clarity independently of the citation placement. Thank you for pointing it out.
>
> >**Improving algorithm presentation and descriptions**
>
> Thank you for your helpful feedback. We agree that the current algorithm presentation is concise but could benefit from clearer structure and more descriptive detail. In the revised version, we will:
>
> - Add informative titles to each algorithm to clarify its purpose (e.g., "Iterative Joint Hierarchical Representation Learning via Tree-Wasserstein Distance" instead of a generic "Algorithm 1" and "Haar Filtering over Alternating Tree Refinement"  instead of "Algorithm 2")
> - Expand the descriptions in each line within the algorithm boxes, making key steps more explicit and easier to follow (e.g., construct an initial sample and feature trees, compute sample and pairwise TWD matrices,  Tree Haar Wavelet filtering, etc)
>
> We hope that these changes will improve clarity without sacrificing conciseness, and make the algorithmic components more accessible and self-contained. If there are any additional specific suggestions for improving the algorithm presentation, we would be happy to incorporate them.
>
> >**Code availability for reproducibility**
>
> We thank the reviewer for highlighting the importance of reproducibility. We fully agree and are currently finalising and cleaning our code to ensure it is clear and easy to use for the community. The full implementation will be released, and we will include a link in the final version of the paper. Additionally, we have provided detailed implementation information in Appendix E, including model settings, training procedures, and hyperparameters, allowing one to fully reproduce our results. If there are any specific details that are missing or unclear, we would be happy to clarify them.  We apologise for any inconvenience caused by releasing our code upon publication, and appreciate the reviewer’s understanding.
>
> >**On the interpretability of learned trees and TWDs for end users**
>
> We thank the reviewer for raising this important aspect. This is indeed a valuable question. Our motivation for applying the method to document and scRNA-seq datasets because the hierarchical structures present in both domains. In document data, the features (words) exhibit hierarchical organisation through semantic groupings (e.g., "animal" → "mammal" → "dog"), while the samples (documents) are structured by topics and subtopics (e.g., science → biology → genomics). In scRNA-seq data, the features are genes, which are functionally related through biological pathways and gene ontologies, forming hierarchical groupings (e.g., immune response genes → cytokine genes → specific interleukins). The samples, i.e., individual cells, follow developmental trajectories or cell-type taxonomies (e.g., hematopoietic stem cell → progenitor cell → mature blood cell types), which are modelled as hierarchical structures in computational biology.
>
>
> Our method produces binary trees. While internal nodes do not always correspond to domain-driven groupings, we observe that the primary branches near the root can sometimes align with prominent structures in the data. In the case of scRNA-seq, this suggests a possible connection to major developmental stages or lineage bifurcations, though a more systematic evaluation would be needed to confirm such alignment. Investigating the biological interpretability of these branches remains an interesting direction for future work. In applications such as clustering or classification, we find that the coarse structure of the binary tree can highlight major distinctions in the data. These observations also point to a promising future direction: relaxing the binary tree constraint and allowing more flexible branching, for example, using Haar coefficients to automatically adjust the number of internal nodes. This could improve the interpretability of finer internal groupings.
>
>
> In terms of the TWD, its strength lies in its transparent and decomposable structure. Given two distributions (e.g., feature or sample vectors), the TWD quantifies the minimal "mass transport" needed to align them over the tree, with transport cost reflecting branch length. This formulation enables users to understand where and at what scale differences occur. For instance, if most of the transport cost is concentrated at high-level branches, the two distributions differ in broad terms; if the cost is concentrated near the leaves, they are largely similar but differ in fine details. As a result, TWD not only quantifies similarity but also supports qualitative interpretation: users can trace which parts of the tree contribute most to the distance between any two data points. In classification tasks, this interpretability reveals which regions of the hierarchy drive class separation and can help identify key transitions in the data.
>
>
> To further highlight this interpretability, we will include in the revised manuscript visual breakdowns of TWD computation and learned tree structures for real datasets, demonstrating how the hierarchy enables multiscale reasoning and class separation. We hope these visualisations will help clarify how particular branches, at varying scales, relate to potential biological or functional groupings and will give users insight into the decision boundaries and similarity patterns uncovered by the model.
>
> >**On visualising tree levels and their refinement**
>
> We thank the reviewer for this valuable comment regarding the visualisation of different levels of the learned trees and how they evolve during each iteration in our method. We agree that stepwise visualisations would offer valuable insight into the process of hierarchical refinement. Thank you for this suggestion! Since, at this stage, we are unable to include new figures in the PDF, we provide here a detailed textual description of the tree refinement throughout the iterative process, based on the toy video recommendation example in Appendix F.1.
>
> The iterative refinement process starts with an initial tree structure, which is unstructured since the preliminary grouping is based on initial data relationships ($\mathbf{M}_c$ or $\mathbf{M}_r$). At this point, major clusters are not well formed, and samples or features from different categories are mixed. After the first iteration, the algorithm uses the current tree from one mode (e.g., users) to compute a TWD for the other mode (e.g., videos). This leads to a newly induced tree that starts to separate the major categories, and major branches start to reflect groupings such as fiction and documentary in videos, but subcategory assignments are not fully sorted. As the algorithm proceeds through subsequent iterations, it alternately refines the trees for users and videos. The updated tree for one mode informs a more accurate computation of TWD in the other mode, enabling finer divisions within the existing branches. Subcategories like action, drama, and sci-fi (for fiction videos) become more distinct and better grouped together. After several iterations, the subcategories become more cohesive and consistently organised, and the tree structures stabilise.
>
> In our revised manuscript, we will include these visualisations that illustrate how the trees evolve with each iteration and point out key groupings formed in the learned trees. We hope these new figures will make it possible to directly observe how broad groupings emerge in early iterations and how finer subcategory assignments are progressively refined, ultimately converging to clearly organised hierarchies.
>
> ---
>
> We thank the reviewer again for their insightful comments and questions. We look forward to discussing any ongoing concerns and suggestions further.

---

### Official Review · Reviewer_DBYz · 2025-06-29

**Clarity:** 3
**Significance:** 3
**Originality:** 3
**Rating:** 5
**Confidence:** 2

**Summary:**

This paper presents an unsupervised method that jointly learns hierarchical representations of samples and features using the Tree Wasserstein Distance (TWD). The approach is shown to perform well across a variety of datasets, demonstrating its general applicability. Additionally, the authors provide a theoretical analysis proving the convergence of the proposed method.

**Questions:**

While the method is generally well-motivated, the paper would benefit from a more detailed explanation of its applicability to specific data domains. In particular:

What is the intuition or motivation for applying this method to document and scRNA-seq datasets and how many features and samples for these datasets?

What downstream analyses or tasks could benefit from the learned hierarchical representations? For example, can these representations improve clustering, trajectory inference, or biomarker discovery in the scRNA-seq context?

Providing more domain-specific insights and potential applications would enhance the clarity and impact of the work.

**Ethical Concerns:**

["NO or VERY MINOR ethics concerns only"]

**Final Justification:**

I will keep my score as accept

**Limitations:**

While the proposed method is technically sound, the paper would benefit from a clearer discussion of the practical implications of the learned hierarchical structures. Specifically, it would be helpful to elaborate on what these hierarchies represent in various real-world datasets, and why they are expected to be meaningful or useful. Additionally, outlining potential downstream applications—such as improved interpretability, clustering, or other analytical tasks—could further strengthen the motivation and demonstrate the broader utility of the approach.

**Quality:**

3

**Strengths And Weaknesses:**

Strength:

The paper addresses an important and interesting problem—learning latent hierarchical structures in both data samples and features. The methodology is clearly presented, and the authors provide a sound theoretical proof of convergence. The effectiveness of the proposed algorithm is demonstrated through a diverse set of experiments, including classification and prediction tasks across multiple data types such as word-document corpora, single-cell RNA-sequencing data, and graph datasets. The method is also thoroughly compared against a range of existing approaches.

Weakness:
The paper would benefit from a clearer explanation of the motivation for applying this method to specific datasets, such as documents data and scRNA-seq data. In particular, it is unclear what hierarchical structure in samples and features means in the context of scRNA-seq, and why this structure is expected to be meaningful or beneficial. Providing more domain-specific insights or interpretations would help justify the applicability and impact of the method in these settings.

---

> ### Author Rebuttal · Authors · 2025-07-30
>
> We thank the reviewer for their insightful review and for recognising the soundness of our theoretical results and the significance of the problem we address. In terms of the weaknesses and questions:
>
> >**Motivation and hierarchical structure in domain-specific applications**
>
> We thank the reviewer for this helpful suggestion. We agree that the paper would benefit from a more explicit discussion of why our method is well-suited to document and scRNA-seq datasets. In document data, the features (words) exhibit hierarchical organisation through semantic groupings (e.g., "animal" → "mammal" → "dog"), while the samples (documents) are structured by topics and subtopics (e.g., science → biology → genomics). Representing these hierarchical relationships can improve representation learning in unsupervised settings. We evaluate our method on four word-document benchmarks used in the word mover’s distance and TWD literature (dataset sizes are reported in Appendix E): (i) the BBCSPORT dataset, consisting of 13,243 bags of words (BOW) and 517 articles, (ii) the TWITTER dataset, comprising 6,344 BOW and 2,176 tweets, (iii) the CLASSIC dataset, including 24,277 BOW and 4,965 academic papers, and (iv) the AMAZON dataset, containing 42,063 BOW and 5,600 reviews.
>
> In scRNA-seq data, the features are genes, which are functionally related through biological pathways and gene ontologies, forming hierarchical groupings (e.g., immune response genes → cytokine genes → specific interleukins). The samples, i.e., individual cells, follow developmental trajectories or cell-type taxonomies (e.g., hematopoietic stem cell → progenitor cell → mature blood cell types), which are modelled as hierarchical structures in computational biology. For example, in the ZEISEL dataset, which includes 4,000 marker genes across 3,005 single cells from the mouse cortex and hippocampus, hierarchical gene programs underlie neuronal specialisation and brain region identity. In the CBMC dataset, which contains 500 gene markers and 8,617 cells from cord blood mononuclear cells, gene modules related to immune function form structured groupings, while the cells follow a developmental hierarchy from cell types. Representing these hierarchical structures can provide insights into lineage progression, cell type classification, and gene module activation across conditions.
>
> Our method jointly learns hierarchies across both modes, samples and features, making it suitable for these domains. We will incorporate this domain-specific motivation into the revised introduction. We thank the reviewer again for encouraging this clarification.
>
> >**Downstream analyses benefiting from the learned hierarchical representations**
>
> We thank the reviewer for this helpful suggestion. We agree that explicitly outlining downstream applications would further strengthen the motivation and demonstrate the utility of our method. The hierarchical representations learned by our framework support a variety of downstream tasks. In Section 6, we demonstrated improvements in document and cell classification, as well as link prediction and node classification in hierarchical graph data. Additionally, in Appendix F, we reported results on clustering performance on gene expression data, showing that our method provides meaningful groupings in biological domains.
>
> From an application perspective, we believe our method is relevant for biological and semantic data. In scRNA-seq analysis, the sample hierarchy represents biological structure such as cell types or subpopulations, often supporting improved clustering compared to flat representations. In document data, the feature trees over words and the sample trees over documents capture semantic and topical organisation, which is valuable for tasks like topic modelling and document classification. In addition, the feature hierarchy learned from gene expression data groups genes by co-expression and functional relevance. This supports tasks like gene module discovery and potentially biomarker identification at multiple levels of biological resolution. In biomedical applications, we learned that such hierarchical representation could also facilitate visualisation and hypothesis generation.
>
> Our method is also directly applicable to hierarchical graph datasets, where it can be used as a preprocessing step to initialise HGCNs. We demonstrate this on several benchmark datasets: the DISEASE dataset, which simulates the SIR disease spreading process with infection status and susceptibility features; the AIRPORT dataset, representing flight networks between airports; the CORA citation network, where nodes represent research papers classified into machine learning categories; and the PUBMED citation network, which includes three classes of medicine publications. In these settings, initialising HGCNs with our learned hierarchical representation leads to consistent performance gains for graph learning tasks, suggesting the utility of our method for hyperbolic graph learning.
>
> Beyond these domains, our approach could be beneficial in several other areas. For example, in social network analysis, users (samples) and attributes or relationships (features) often display group or role hierarchies; our method supports community detection. In chemoinformatics and drug discovery, both molecules and their substructural or chemical property features can reflect hierarchical organisations, assisting in clustering and identification. In education analytics, student data and knowledge components, such as skills or assessment questions, could exhibit hierarchical structure as well, enabling adaptive curriculum design and hierarchical clustering of learners or resources.
>
> In the context of trajectory inference in scRNA-seq, our method produces binary trees. While internal nodes do not always correspond to domain-driven groupings, we observe that the primary branches near the root can sometimes align with prominent structures in the data. In the case of scRNA-seq, this suggests a possible connection to major developmental stages or lineage bifurcations, though a more systematic evaluation would be needed to confirm such alignment. Investigating the biological interpretability of these coarse-scale branches remains an interesting direction for future work. Although our method is not explicitly designed for trajectory inference, the hierarchical structure it produces could potentially serve as a meaningful scaffold for reconstructing developmental processes, without requiring additional modelling assumptions. These observations also point to a promising future direction: relaxing the binary tree constraint and allowing more flexible branching, for example, using Haar coefficients to adjust the number of internal nodes automatically. This could improve the interpretability of finer internal groupings.
>
>
> We will incorporate these domain-specific applications and future directions into the revised manuscript. We thank the reviewer again for encouraging us to include the broader relevance of our method.
>
> ---
>
> We thank the reviewer again for their helpful comments and hope to discuss any ongoing concerns or suggestions further.

---

### Official Review · Reviewer_7yBG · 2025-07-01

**Clarity:** 3
**Significance:** 2
**Originality:** 3
**Rating:** 5
**Confidence:** 3

**Summary:**

This paper focuses on learning hierachical representations via Tree Wasserstein Distance (TWD). The paper proposes a novel tree-based representation algorithm that iteratively aligns the feature and sample hierachy by optimizing the TWD distance with coordinate descent method. A variant based on Haar wavelets is introduced to reduce the noisy components in the constructed feature tree. Experiments on word-document and sRNA-seq datasets verify the quality and effectiveness of the proposed representation algorithms.

**Questions:**

1. Is it possible to replace TWD with other graph-based distance while preserving the hierachical features in the outputs?

**Ethical Concerns:**

["NO or VERY MINOR ethics concerns only"]

**Final Justification:**

The rebuttal message addresses the majority of my previous concerns and clarrify the main contribution of the paper, therefore I would raise the score.

**Limitations:**

yes

**Quality:**

3

**Strengths And Weaknesses:**

Strengths:
+ A novel framework that generates informative document and word-level hierachical representations.
+ The convergence of the methods is grounded by theoretical analyses and demonstrations.
+ Comprehensive experiments validate the effectiveness of the algorithms.

Weaknesses:
- As the core of the optimization algorithms, the tree construction approach and the TWD metric are both adopted from prior research works.
- Due to the discerete nature of TWD, it appears challenging to integrate the proposed optimization objective into other neural network-based models. On the other hand, the effectiveness of the algorithm as a preprocessing step should be evaluated among a wider range of backbone models, such as GCN and GraphSAGE.
- While only the tree-based Wasserstein distance is utilized as the similarity metric, its advantage over other tree distance metrics is not adequately discussed or explored via experiments.

---

> ### Author Rebuttal · Authors · 2025-07-30
>
> We thank the reviewer for their insightful and in-depth evaluation of our paper, and for recognising the theoretical grounding of our method. In terms of the weaknesses and questions:
>
> >**Use of established methods and our novel contributions**
>
> We thank the reviewer for commenting on this point. We agree that our method indeed builds on existing components: the TWD as introduced in prior work by Indyk and Thaper (2003), and tree construction methods based on diffusion geometry and hyperbolic embedding as proposed by Lin et al. (2025). However, our main contribution lies in the formulation of the joint hierarchical representation learning problem for both samples and features, and in the development of a novel iterative algorithm that alternates between these two modes, so that each hierarchy helps improve the other and jointly refines both. The proposed algorithm is new, as are the theoretical convergence guarantees we establish and the utility of our methods as a preprocessing step for HGCNs. This alternating optimisation, along with the Haar wavelet filtering over the learned trees, constitutes a new approach that extends beyond prior uses of TWD or hierarchical representations. In addition, the iterative process is broadly applicable to various TWD-based methods and lays a foundation for future extensions to more general geometric representations beyond trees, while retaining the proposed iterative procedure based on the Wasserstein distance learned between samples and features. We will highlight more explicitly what is novel in our formulation, optimisation scheme, and applications in the revision. Thank you for bringing this up, and we will better clarify the novel contribution of this work in the revision.
>
> >**On the integration of TWD with neural architectures**
>
> We thank the reviewer for this insightful feedback. These are valuable suggestions that could extend the scope of our work.
>
> Indeed, due to the discrete nature of TWD, direct integration into end-to-end neural network training is non-trivial. In this work, we focus on using our method as a preprocessing step that produces hierarchical representations that can then be used as input to initialise hyperbolic graph convolutional networks, where we observe performance improvements. While this is the current scope of our work, the reviewer's comment actually motivates a promising direction: exploring a differentiable variant of TWD, such as the soft TWD proposed by Takezawa et al. (2021), could enable integration of our iterative process into neural architectures. In Takezawa's work, the TWD is differentiable with respect to the probability of the parent–child relationships of a tree.  In our setting, these relationships are derived from the diffusion operator introduced by Coifman and Lafon (2006), for which gradients can be computed. While a full integration would require further investigation, this connection suggests that a differentiable extension of our iterative framework may be feasible and could further extend the applicability of our method within neural architectures. We will add this direction for future work in the revised manuscript. Thank you again for motivating us to consider this line of research.
>
> >**Evaluating the preprocessing step with additional backbones**
>
>  In terms of evaluating the effectiveness of our method as a preprocessing step across different neural network backbones, we appreciate the suggestion to extend our experiments beyond the set of models originally included. In response, we expanded our experimental analysis to include GCN and GraphSAGE, in addition to the hyperbolic models originally considered. We report the link prediction and node classification performance in the table below. We find that initialising these models with our learned hierarchical representations  (denoted "-Alg.2") consistently improves performance in node classification and link prediction tasks. Improvements are marked with an upward arrow ($\uparrow$). These results support the general applicability and effectiveness of our method as a preprocessing step across different neural architectures. Notably, we see that HGCN-Alg.2, which incorporates our method as an initialisation within HGCNs, still achieves the best overall results. We attribute this to the unique advantages of hyperbolic representations for modelling hierarchical structures, which distinguishes HGCNs from other Euclidean-based models. We will include these results in the revised manuscript. Thank you again for this valuable suggestion.
>
>  | | DISEASE LP| DISEASE NC| AIRPORT LP| AIRPORT NC| PUBMED LP| PUBMED NC| CORA LP| CORA NC|
> |------------------|-------------------|-------------------|-------------------|-------------------|-------------------|-------------------|-------------------|-------------------|
> | GCN| 64.7±0.5| 69.7±0.4| 89.3±0.4| 81.4±0.6| 91.1±0.5| 78.1±0.2| 90.4±0.2| 81.3±0.3|
> | GCN-Alg.2| 68.1±0.4 (↑)| 71.2±0.5 (↑)| 90.6±0.2 (↑)| 83.0±0.9 (↑)| 90.5±0.3| 78.7±0.1 (↑)| 91.2±0.4 (↑)| 81.9±0.2(↑)|
> | GraphSAGE| 65.9±0.3| 69.1±0.6| 90.4±0.5| 82.1±0.5| 86.2±1.0| 77.4±2.2| 85.5±0.6| 77.9±2.4|
> | GraphSAGE-Alg.2  | 67.8±0.1 (↑)| 70.5±0.3 (↑)| 90.2±0.4| 83.7±0.5 (↑)| 90.3±1.1 (↑)| 78.8±1.7 (↑)| 89.6±0.8 (↑)| 80.5±1.6 (↑)|
> | HGCNs| 90.8±0.3| 74.5±0.9| 96.4±0.1| 90.6±0.2| 96.3±0.0| 80.3±0.3| 92.9±0.1| 79.9±0.2|
> | H2H-GCN| 97.0±0.3| 88.6±1.7| 96.4±0.1| 89.3±0.5| 96.9±0.0| 79.9±0.5| 95.0±0.0| 82.8±0.4|
> | HGCN-Alg.2| **98.4±0.4 (↑)**| **89.4±0.3 (↑)**| **97.2±0.1 (↑)**| **92.1±0.3 (↑)**| **97.2±0.2 (↑)**| **83.6±0.4 (↑)**| **96.9±0.3 (↑)**| **83.9±0.2 (↑)**|
>
> >**Discussion and comparison with other tree distance metrics, and replacing TWD with other graph-based distances while preserving hierarchical features**
>
> We thank the reviewer for the insightful comment and question.  In our framework, TWD plays a central role due to its ability to compare probability distributions supported on hierarchical structures. This is essential for our alternating, joint-learning process, where the hierarchical structure from one mode (e.g., samples) is used to inform and refine the hierarchy in the other mode (e.g., features), and vice versa. That is, the tree structure provides the foundation for expressing hierarchy, while the Wasserstein metric provides the mechanism for distributional comparison and transfer from one mode to the other. This cross-mode interaction is central to our method and relies on the Wasserstein nature of TWD.
>
> We agree that the paper would benefit from a more explicit discussion of how TWD differs from other tree distance metrics, e.g., minimum spanning trees, neighbour joining or low-stretch trees. These tree distance metrics typically operate on tree structures or discrete node sets, but do not directly support distribution-based comparison that allows for cross-mode refinement. To clarify this difference, we consider two alternative setups using MST: (1) using independently constructed MSTs for samples and features without cross-mode interaction, and (2) using an MST as the tree structure in the TWD framework for our iterative learning process (Co-MST-TWD). We compare these alternatives to our full method (Alg. 2) by measuring representation quality using the $L_1$ norm of Haar coefficients. Results are shown in the table below (format: sample / feature). We see that both alternatives yield substantially higher $L_1$ norms. These results suggest that the learned joint hierarchical representation from our iterative process better reflects the hierarchical information of the two data modes. We attribute this advantage to the specific design of our proposed method. In the revised manuscript, we will emphasise more how TWD supports iterative cross-mode refinement, which is not feasible with independent tree metrics, and report the empirical comparison to support our use of TWD. We thank the reviewer again for encouraging us to address this important point more thoroughly.
>
> ||BBCSPORT|TWITTER|CLASSIC|AMAZON|ZEISEL|CBMC|
> |------------------|---------------|---------------|---------------|---------------|----------------|----------------|
> | Independent MST| 37.9 / 42.8| 82.4 / 31.1| 83.6 / 131.4| 109.6 / 144.1 | 194.3 / 281.5| 1928.7 / 97.6  |
> | Co-MST-TWD| 30.4 / 34.1| 69.1 / 22.1| 77.5 / 109.3  | 97.0 / 126.4  | 179.1 / 254.0  | 1755.3 / 83.7  |
> | Alg.2| **10.1 / 4.8**| **22.4 / 3.4**| **37.2 / 19.4**| **46.6 / 26.6**| **50.9 / 93.7**| **489.4 / 45.6**|
>
> Regarding graph-based distances, from a conceptual standpoint, using other graph-based distances is possible, but there are important considerations:
>
> - Most graph-based distances are designed to compare node positions or global connectivity in general graphs, not to measure how distributions are supported on a tree or reveal explicit nested groupings
> - Gromov-Wasserstein distance could, in principle, be integrated into an iterative scheme as it compares distributions across different metric spaces or graphs. However, such approaches are much more computationally intensive
> - Another possible direction is to construct a graph from high-dimensional data and transform this general graph into unrolling trees, as explored in recent works such as Chuang \& Jegelka (2022), Rauchwerger et al. (2025), and Vasileiou et al. (2025). Further investigation would be needed to determine if and how such transformations can be integrated within our iterative learning process
>
> We will add this discussion as a potential extension for future work to the revision. We appreciate the reviewer’s suggestion to explore broader classes of hierarchical metrics and would be happy to consider specific alternatives if the reviewer has recommendations.
>
> ---
> We thank the reviewer again for their valuable comments and questions. We look forward to discussing any ongoing concerns and suggestions further.

---

### Official Review · Reviewer_pif9 · 2025-07-10

**Clarity:** 3
**Significance:** 3
**Originality:** 3
**Rating:** 5
**Confidence:** 3

**Summary:**

This paper proposes a novel representation learning method for data with hierarchical structure in both sample dimension and feature dimension. The method proceeds by refining Tree-Wasserstein Distance for both dimension in an alternative manner. The authors also propose to use Haar filter to reduce noises and even show that the proposed method can work well on graph data. Overall, this is a novel and interesting work with solid theoretical insights and extensive experiment results.

**Questions:**

Please see weaknesses

**Ethical Concerns:**

["NO or VERY MINOR ethics concerns only"]

**Final Justification:**

This is a novel paper and my concerns have been resolved as acknowledged. I will maintain the recommended score.

**Limitations:**

Yes.

**Paper Formatting Concerns:**

No.

**Quality:**

3

**Strengths And Weaknesses:**

**Strengths:**
- The paper is targeting an important and common representation learning scenario with a novel TWD-based method; the alternating refinement step with proved convergence guarantee is reasonable
- The experiments are extensive and solid - I especially like the one that shows proposed method can work well as an initialization step for hyperbolic graph data, which is an interesting application scenario.

**Weaknesses:**
 - I think some terms in this paper could be presented with more clarity. For example, it was not clear to me what the "two data modes" of "samples and features" mean until I read through to the second paragraph. It could be mentioned earlier that they correspond to rows and columns of an m by n data matrix.
-  The authors may consider further integrate the Haar wavelet filtering into Figure 1's illuatration

---

> ### Author Rebuttal · Authors · 2025-07-30
>
> We thank the reviewer for the thoughtful and encouraging review, and for recognising the relevance of addressing an important and common representation learning scenario, as well as its application to hyperbolic graph data. In terms of weaknesses:
>
> >**Clarifying the terminology regarding data modes**
>
> Thank you for this valuable suggestion. We agree that introducing "samples" and "features" as corresponding to the rows and columns of an $n \times m$ data matrix at the very beginning would improve the clarity and accessibility of our paper. In the revision, we will update the first paragraph of the introduction to clearly state early on that the "two data modes" refer to the rows ("samples") and columns ("features") of the data matrix. We hope this clarification will help readers immediately understand the problem setting. If there are any further suggestions for clarifying the terminology, we would be happy to incorporate them. Thank you!
>
>
> >**Including Haar wavelet filtering in the illustration in Figure 1**
>
> We thank the reviewer for the helpful suggestion. We agree that including the Haar wavelet filtering step into Figure 1 would provide a more complete visual summary of the overall method. In the revised version, we will add a filtering block below each learned tree to represent the application of wavelet filtering to the data. This filtered data is then used in the computation of the TWDs for the subsequent iteration. We hope this addition will enhance how the filtering is induced by the learned tree, how it modifies the data between iterations, and how it contributes to refining the joint hierarchical representation. Thank you again for this helpful recommendation.
>
> ---
>
> We thank the reviewer again for their constructive comments and hope to discuss any ongoing suggestions further.

---

> > ### Comment · Reviewer_pif9 · 2025-08-07
> >
> > Thanks for your response. I have read it and am happy to keep my current score. Good luck.

---

### Decision · Program_Chairs · 2025-09-17

**Decision:**

Accept (spotlight)

**Comment:**

This manuscript proposes a methodology for learning hierarchical representations of data along both features and samples. In addition to motivating and introducing the idea, the authors provide theoretical convergence results for their method and an empirical study showing improvement in a variety of learning tasks.

This work was received in extremely regard by the reviewers. In particular, the reviewers were impressed by the paper's problem setting -- learning hierarchical representations for both features and samples -- and equally impressed by the design and performance of the proposed methodology.

Due to the novelty of this setting, and the universally positive reception of this work, I believe this paper is appropriate for spotlight presentation or higher.